# Early Jurassic climate and atmospheric $CO_2$ concentration in the Sichuan paleobasin, Southwest China

Xianghui Li[1], Jingyu Wang[1], Troy Rasbury[2], Min Zhou[1], Zhen Wei[1], Chaokai Zhang[1]

[1]State Key Laboratory for Mineral Deposits Research, School of Earth Sciences and Engineering, Nanjing University, Nanjing 210023 China.

[2]Department of Geosciences, Stony Brook University, Stony Brook, NY 11794-2100, USA

*Correpondence to:* Xiangui Li (leeschhui@126.com)

**Abstract:** Climatic oscillations had been developed through the (Early) Jurassic from marine sedimentary archives, but remain unclear from terrestrial records. This work presents investigation of climate-sensitive sediments and carbon and oxygen isotope analyses of lacustrine and pedogenic carbonates for the Early Jurassic Ziliujing Formation from the grand Sichuan paleobasin (GSB), Southwest China. Sedimentary and stable isotope proxies manifest that an overall secular (semi-) arid climate dominated the GSB during the Early Jurassic except for the Hettangian. This climate pattern is similar to the arid climate in the Colorado Plateau region, western North America, but distinct from the relatively warm-humid climate in North China and high latitude in Southern Hemisphere. The estimated atmospheric $CO_2$ concentration ($pCO_2$) from carbon isotopes of pedogenic carbonates shows a range of 980-2610 ppmV (~ 3.5-10 times the pre-industrial value) with a mean of 1660 ppmV. Three phases of $pCO_2$ (the Sinemurian 1500-2000 ppmV, the Pliensbachian 1000-1500 ppmV, and the early Toarcian 1094-2610 ppmV) and two events of rapid falling $pCO_2$ by ~1000-1300 ppmV are observed, illustrating the $pCO_2$ perturbation in the Early Jurassic. The perturbation of $pCO_2$ is compatible with seawater temperature and carbon cycle from the coeval marine sediments, suggesting a positive feedback of climate to $pCO_2$ through the Early Jurassic.

# 1. Introduction

Global paleotemperatures were possibly 5-10°C higher than present druing the Jurassic period based on climate modelling results (e.g., Rees et al., 1999; Sellwood and Valdes, 2008). However, seawater temperature fluctuated by-5 °C to +5 °C, or even much higher magnitude (e.g., Suan et al., 2008; Littler et al., 2010), based on esitmates from the oxygen isotopes of the belemnite and bivalve fossils (Dera et al., 2011, and references therein). In the Sinemurian–Pliensbachian age, the mean sea surface temperatures of the North Atlantic were in excess of 28°C ($TEX_{86}$), comparable with similar palaeolatitudes during the Cretaceous and Early Cenozoic (Robinson et al., 2017); whereas in the late Pliensbachian age, the northern West Tethys Ocean (e.g., Paris basin, northern Spain basin) was ~12.7°C (e.g., Gómez et al., 2008; Gómez and Goy, 2011; Arabas et al., 2017), leading to a polar icesheet hypothesis (e.g., Sellwood and Valdes, 2008; Suan et al., 2010; Dera et al., 2011; Gómez et al., 2015). At ~183 Ma of the early Toarcian oceanic anoxia event (T-OAE), the surface seawater temperature was high to ~35°C (e.g., Bailey et al., 2003; Korte et al., 2015), and a high temperature (plateau) even continued in the whole Toarcian (Dera et al., 2011). Examples of seawater temperature transitions between cold and hot show the climate oscillation through the Early Jurassic.

Data from the terrestrial realm also provide important details of environmental and climatic change (e.g., Hesselbo et al., 2000; Suan et al., 2010; Jenkyns, 2010; Philippe et al., 2017), from which the oscillated climate could be observed and revealed too. Terrestrial proxies, such as flora (e.g., Riding et al., 2013; Deng et al., 2017; Philippe et al., 2017), vegetation (Pole, 2009), and geochemistry (e.g., Riding et al., 2013; Kenny, 2015; Tramoy et al., 2016) as well as the $p$CO$_2$ record (e.g., Retallack, 2001a; Beerling and Royer, 2002; McElwain et al., 2005; Berner, 2006; Steinthorsdottir and Vajda, 2015) provide an emerging record of the Early Jurassic terrestrial climate and environment changes. Correspondingly, the proxy application of terrestrial sedimentary archives could play a key role in the global Early Jurassic correlation of the marine and terrestrial climate.

Proxies for $p$CO$_2$ are the important linkage between the marine and terrestrial climatic condition. Studies of the terrestrial $p$CO$_2$ record have focused on the Triassic-Jurassic boundary (e.g., Tanner et al., 2001; Cleveland et al., 2008; Schaller et al., 2011; Steinthorsdottir and Vajda, 2015) and the Toarcian oceanic anoxic event (McElwain et al., 2005), where $p$CO$_2$ estimates range 1000 ppm to 4000 ppmV (e.g., Tanner et al., 2001; Cleveland et al., 2008; Schaller et al., 2011). Few relatively continuous $p$CO$_2$ records and coupled terrestrial climate changes have been documented for the Early Jurassic.

There are several large Triassic-Jurassic terrestrial basins in West China, in which the Sichuan Basin has a relatively complete and continuous continental sedimentary sequence of the Upper Triassic-Paleogene (e.g., SBGM, 1991, 1997; Wang et al., 2010). During the Early Jurassic, the Sichuan Basin was in a Boreotropical climate zone based on climate-sensitve sediments (Fig. 1a. Boucot et al., 2013), or a warm temperate climate is suggested based on clay mineralogy and phytogeography (e.g., Dera et al., 2009). In this work, we present a field investigation, including lithofacies and paleosol

interpretation, and carbon and oxygen isotope analyses of both lacustrine and pedogenic carbonates in Sichuan Basin. New results allow us to reconstruct the paleoclimate and relatively consecutiove $p$CO$_2$ record through the Early Juassic, for which we compare to stable isotopes of marine sediments and estimated sea water temperature.

**2. Geological setting and stratigraphy**

Southwest China, including the provinces of Yunnan, Sichuan, Chongqing, and Guizhou, had been the main part of the upper Yangtze Plate since the Proterozoic, possibly since the Neoarchean. With the amalgamation of the Cathaysia and Yangtze plates, it became the western South China plate or cratonic basin since the Neoproterozoic (Sinian),and continued to the late Middle Triassic. By the Indosinian orogeny, new foreland basins were formed since the Late Triassic (e.g., He and Liao, 1985; Li et al., 2003), recording the Mesozoic and Cenozoic evolution of tectonics, environment, and climate in Southwest China.

The Mesozoic Sichuan paleobasin was confined by the Longmenshan thrust belt in the northwest, the Micangshan-Dabashan arcuate thrust belt in the northeast (Fig. 1b), and the northern hilly topography boundary of the Yunnan-Guizhou plateau in the south and east. It was mainly developed during the Late Triassic-Jurassic and includes provincial areas of eastern Sichuan, entire Chongqing, northern Guizhou, western Hubei, and northwestern Hunan. This Triassic-Jurassic Sichuan foreland basin was much larger than the present Sichuan Basin in the eastern Sichuan province. We estimate the size of Sichuan paleobasin is roughly 480,000 km$^2$ based on lithofacies paleogeography (Fig. 1b. Ma et al., 2009; Li and He, 2014), and suggest naming this the grand Sichuan paleobasin (GSB).

The Mesozoic terrestrial sediments accumulated up to ~9 km (Guo et al., 1996) in the GSB; and the Jurassic part can be as much as 3-3.5 km thick (SBGM, 1991). Two types of Lower Jurassic deposits have been distinguished (Table 1): the Baitianba Formation (Fm) in the north (~10%) and the Ziliujing Fm (e.g., SBGM, 1991; Wang et al., 2010) in the south (over 90% of the basin).

The Baitianba Fm was deposited unconformably on the Upper Triassic Xujiahe Fm and is overlain conformably by the Middle Jurassic Xintiangou Fm / Qianfuyan Fm (Table 1). It is mainly composed of grayish shales and sandstones with coal layers and massive conglomerates. Abundant plant fossils, sporopollens, conchostracans, bivalves, and gastropods indicate it is of the Early Jurassic (SBGM, 1991, 1997). Sporopollen assemblages of the Hettangian-Sinemurian age were found in the lower part (Zhang and Meng, 1987) and the Pliensbachian-Toarcian assemblages were reported in the upper part (Wang et al., 2010).

The Ziliujing Fm is composed of variegated and reddish mudrocks (some shales) intercalated with sandstones, siltstones, and bioclastic limestones as well as dolomitic marlstones / limy dolomites, conformably or unconformably overlying the Xujiahe Fm or Luqiao Fm and conformably underlying the Xintiangou Fm (SBGM, 1997. Table 1). It has been dated as the Early

Jurassic by fossil assemblages of bivalves, ostracods, conchostracans, and plants. Dinosaur fauna can be well correlated to
the Lufeng Fauna in central Yunnan (e.g., Dong, 1984; SBGM, 1991, 1997; Peng, 2009). This formation is subdivided into
five parts in an ascending order: the Qijiang, Zhenzhuchong, Dongyuemiao, Ma'anshan, and Da'anzhai members (SBGM,
1997. Table 1). Of these, the former two are sometimes combined as the Zhenzhuchong Fm (e.g., SBGM, 1991; Wang et al.,

88  2010).

The Da'anzhai Member is characterized by dark gray to black shales and bioclastic limestones with a southward increase of
reddish mudrocks (SBGM, 1991, 1997; Wang et al., 2010), and is regarded the sediment in a grand Sichuan paleolake (e.g.,
Ma et al., 2009; Li and He, 2014). Ostracod assembleges indicate it is the late Early Jurassic (e.g., Wei, 1982; Wang et al.,
2010). A Re–Os isochron age of 180.3 ± 3.2 Ma associated with an organic carbon isotope excursion indicates that the lower
Da'anzhai Member corresponds to the T-OAE (Xu et al., 2017).
The Ma'anshan Member is comprised of violet-red mudrocks with a few greyish, greenish thin-bedded fine sandstones and
siltstones, in which floral fossils are common (Li and Meng, 2003). The Dongyuemiao Member consists of greenish and
reddish mudrocks and siltstones with greyish bioclastic limestone and marlstone, of which abundant bivalve and plant fossils
were reported from eastern Sichuan and Chongqing (Li and Meng, 2003; Meng et al., 2003; Wang et al., 2010). The
Zhenzhuchong Member is dominated by violet red mudrocks/shales intercalated with thin-bedded sandstones and / or
siltstones and numerous plant fossils of the Early Jurassic affinity (e.g., Duan and Chen, 1982; Ye et al., 1986). Taken
together, fossil associations suggest that the three members were deposited in the middle-late Early Jurassic. The age
limitation of the overlying Da'anzhai Member and the correlation to the Lufeng dinosaur fauna places these members in the
Sinemurian – Pliensbachian, and the Zhenzhuchong and Dongyuemiao Fms are suggested to be the Sinemurian (Table 1).
The Qijiang Member is composed of quartz arenite interbedded/intercalated with dark shales. Coal seams are often seen in
the middle of the Qijiang Member. This member mainly occurs in the central part of the GSB. It is likely the earliest Jurassic,
possibly Hettangian age, but plant fossils cannot precisely indicate the age (Wang et al., 2010).
**3. Materials and methods**
We have measured sections and made detatiled observations and descriptions of sedimentary characteristics for lithofacies
analysis at six outcrop sections (Locations A1 to A4, A6 and A7, Fig. 1). Published descriptions for other sections
(Locations A5, A8, and A9, Fig. 1) are integrated into our observations. Details of microscopic examination of sedimentary
rocks and analysis of sedimentary facies underpinning the climate analsysis are attached as the supplementary data Note S1.
Below we state climate-sensitive sediment observation, carbon and oxygen isotope analyses, and estimate of $p$CO$_2$.
**3.1. Observation of climate-sensitive sediments**
Climate-sensitive sediments are mainly dolomites, gypsum, and paleosols, which are used to analyze the climate in this work
(Table S1).
Dolomites and gypsum are relatively easy to recognize in both field and under microscope. We distinguish dolomites from
limestones following Tucker (2011) and Flügel (2004). As Flügel (2004) stated, field distinctions of limestone and dolomite
can also be made although detailed differentiation of carbonate rocks is best performed in the laboratory. In field, we
recognize gypsum by particular structures such as chicken-wire cage, gypsum pseudomorph, and cluster of (0.5-1 cm) pore.
There are multiple classifications of paleosols (e.g., Wright, 1992; Mack et al., 1993; Retallack, 2001b; Imbellone, 2011),
mostly based on the US Soil Taxonomy. We recognized paleosols in the field based on color, structures, horizonation, root
traces, and textures, and followed the general classification paleosols by Mack et al. (1993) and Retallack (2001b). In this
paper, paleosols are described following the procedures of the Soil Survey Manual and classified according to Soil Survey
Staff (1998).
Within the measured and observed sections, paleosol profiles were mainly identified from the two main locations/sections
A4 and A6 (Figs. S1 and S2, and Table S2). Horizonation, BK horizon thickness, boundaries, structures, trace fossils,
rootlets, carbonate accumulations (calcretes), etc. were recorded (Table S2). Paleosols interpreted in other cited sections (Fig.
1) rely on the description of lithology, strcucture, and calcrete in the original references. Based upon a modification of the
Retallack (1998) categorization of paleosol maturity, the relative paleosol development (maturity) was assigned.
**3.2. Analyses of carbon andoxygen isotopes**
Ten lacustrine carbonate samples were analysed for carbon and oxygen isotopes from the Da'anzhai Member at the Shaping
section, Ya'an (Location A4. Fig. S1 and Table S3). 26 pedogenic carbonate samples were analyzed for carbon and oxygen
isotopes from 32 paleosols of the Ziliujing Fm at the same section (Fig. S1 and Table S4). Two or three microdrilling
powder samples (columns 7 and 8 in Table S4) were taken from the same individual calcrete for stable isotope analysis, and
then a mean value for each calcrete sample was calculated (columns 9 and 10 in Table S4).
At the field scale, calcretes are ginger-like and sporadically spaced within the soil horizon. We observed no linear and planar
calcretes that would indicate precipitation at or below the water table. Before drilling, thin-sections were petrographically
studied using polorized light microscopy and cathodoluminescence imaging. Micritic calcite is predominant in both
lacustrine and pedogenic carbonate samples, with no evidence for carbonate detritus in calcretes (Fig. 2a and 2b). The
micritic calcites used for stable isotope analyses are chiefly null- to non-luminescent, with <10% light orange and brownish
luminescence, indicating genesis primarily in the vadose zone. While luminescent calcretes indicate a high possibility of
hydrological influence (e.g., Mintz, et al., 2016), we sampled to avoid this. Based on petrography and CL imaging together
with the field observations, the dense micritic zones sampled for the stable isotope composition should give pristine $\delta^{13}C$
values that can be used to estimate $pCO_2$.
Microsampling of lacustrine and pedogenic carbonates focused on only micrites, avoiding diagenetic spar from cracks, veins,
and vug spaces. Powder samples were obtained using a dental drill (aiguille diameter $\phi$=1-2 mm).
Isotopic analyses were conducted on 0.3 ~ 0.5 mg powder samples. Powder samples were dried in an oven at 60°C for 10
hours before being moved to the instrument. Carbon dioxide for isotopic analysis was released using orthophosphoric acid at
70°C and analysed on-line in a DELTA-Plus xp (CF-IRMS) mass spectrometer at the State Key Laboratory for Mineral
Deposits Research, Nanjing University. The precision of the measurements was regularly checked with a Chinese national
carbonate standard (GBW04405) and the international standard (NBS19) and the standard deviation of $\delta^{13}C$ was ±0.1‰ over
the period of analysis. Calibration to the international PeeDee Belemnite (PDB) scale was performed using NBS19 and
NBS18 standards.
**3.3. Calculation of atmospheric $CO_2$ concentration**
The Cerling (1991, 1999) equation was used to calculate the $p CO_2$ using the carbon isotope of pedogenic carbonates as
below:
$$C_a = S_{(z)}(\delta^{13}C_s - 1.0044\delta^{13}C_r - 4.4)/(\delta^{13}C_a - \delta^{13}C_s)$$
where $C_a$ is $p CO_2$; $\delta^{13}C_s$, $\delta^{13}C_r$, $\delta^{13}C_a$ are the isotopic compositions (‰) of soil $CO_2$, soil-respired $CO_2$, and atmospheric $CO_2$,
respectively; and $S_{(z)}$ is the $CO_2$ contributed by soil respiration (ppmV).
$\delta^{13}C_s$ is often calibrated by fractionation factor –8.98‰ with the formula -8.98‰+$\delta^{13}C_c$ (Ekart et al., 1999), with which $\delta^{13}C_c$
is the measured result of pedogenic calcrete. Alternatively, $\delta^{13}C_s$ can be replaced by $\delta^{13}C_{sc}$, which is calibrated by carbon
isotope ratio of pedogenic carbonate at 25°C based on latitude–temperature correlations (Besse and Courtillot, 1988; Ekart et
al., 1999) following the equation $\delta^{13}C_{sc} = (\delta^{13}C_c+1000)/((11.98-0.12*T)/1000+1) -1000$ (Romanek et al., 1992). We used
both $\delta^{13}C_s$ and $\delta^{13}C_{sc}$ to calculate the $p CO_2$ (Table S4).
$\delta^{13}C_r$ represents carbon isotope ratio of average bulk C3 vascular tissue (Arens et al., 2000), reflecting atmospheric $\delta^{13}C$
(Jahren et al., 2008). The $\delta^{13}C_{om}$ of organic matter within paleosols based on the range of modern C3 ecosystem
fractionations (Buchmann, et al., 1998; Ekart et al., 1999), is commonly used for $\delta^{13}C_r$. However, the $\delta^{13}C_r$ could be could be
compromised in fossil soils due to oxidation and metabolism of organic matter after burial (Nadelhofer and Fry, 1988). In
this paper, we use the $\delta^{13}C_{om}$ from the Paris Basin (Bougeault et al., 2017; Peti et al., 2017) for the Sinemurian-Pliensbachian
$\delta^{13}C_r$ and from Cardigan Bay, UK (Xu et al., 2018) for the Toarcian.
$\delta^{13}C_a$, the carbon isotopic composition of the atmosphere, was about -8‰ in the 1980s, being depleted relative to the
pre-industrial atmosphere which was around -6.5‰ (Friedli et al., 1986). The average value of –6.5‰ has been chosen as the
$\delta^{13}C_a$ for acquiring $\delta^{13}C_r$ and $S_{(z)}$ (e.g., Ekart et al., 1999; Robinson et al., 2002), and the $\delta^{13}C_a$ was generally calibrated as
$\delta^{13}C_{ac}$ from $\delta^{13}C_r$ using the formula $(\delta^{13}C_r+18.67)/1.1$ (Arens et al., 2000). Herein we used both calibrations to calculate the
$\delta^{13}C_a$ (Table S4).
$S_{(z)}$ is the largest source of uncertainty in $p$CO$_2$ estimates (Breecker, 2013) and the uncertainty arises primarily from their
sensitivity to soil-respired CO$_2$ ($S_{(z)}$ (Montañez, 2013). It is a function of depth and effectively constant below 50 cm (e.g.,
Cerling, 1991). $S_{(z)}$=2500 ppmV is suggested for the sub-humid temperate and tropical climates (Breecker et al., 2010),
2500-5000 ppmV for higher moisture and productivity soil (Montañez, 2013), 2000 ppmV for semi-arid areas (Breecker et
al., 2009), 1500-2000 ppmV for aridisols and alfisols (calcisol-argillisol) and 2000±1000 for paleo-vertisol (Montañez,
2013), and 1000 ppmV in desert areas (Breecker et al., 2010) or 400 ± 200 ppmV for immature soil (Montañez, 2013). In
this context, we chose the $S_{(z)}$=2000 ppmV for calculating $p$CO$_2$ at 25˚C as the calcisols are reddish-brownish aridisols, and
we also compared the results with that by $S_{(z)}$=2500 ppmV (Table S4). Additionally, we took samples at the middle and
lower Bk horizon (often > ~20-30 cm to the BK top). That means the depth of calcrete samples in the examined palaeosols
was generally deeper than 50 cm below the paleosol surface, meeting the requirement for a constant value of $S_{(z)}$.

## 4. Results

Based on the investigation of cross-sections (locations A1-A4, and A6-A7. Fig. 1), we have classified six sedimentary facies
units in the Ziliujing Fm. They are alluvial fan, fluvial river, flood plain, lake, lake-delta, and swamp facies. Details of
description and interpretation are in the supplementary data Note S1. Below are results of climate-sensitive sediment
observation, stable isotope analyses, and $p$CO$_2$ calculation.

### 4.1. Climate-sensitive sediments

Field observation combined with published calcrete materials shows that paleosols widely occur in the Lower Jurassic
Ziliujing Fm of the GSB (Figs. 1, 3, and 4). A total of 32 paleosols were observed and described at the Shaping section,
Ya'an, and five paleosols were found at the Tanba section, Hechuan (Table S2).
Most of paleosols are reddish (GSA Munsell Rock-Color 5R 2/2, 5R 3/4, 5R 4/2) and brownish (10R 3/4, 10R 5/4) (Fig. 3
and Table S2). Peds of paleosols are mainly angular and subangular, and a few are prismatic and platy. Slickensides are
common. Mottles (Fig. 3a), rootlets /rhizoliths (Fig. 3c), and burrows sometimes occur with strong leaching structures (Fig.
3a). Occasionally mudcracks are associated with the aforementioned structures (Fig. 3d).
All paleosols are calcic with more or less calcretes in Bk horizons. The thickness of Bk horizons mainly changes from 30 cm
and 100 cm, and partly up to 170 cm (Table S2). Calcretes are generally ginger-like, ellipsoid, subglobular, and irregular in
shape (Fig. 3b and 3e) and nodules are 1-3 cm even up to 8-15 cm (paleosols J1z-10-01 and J1z-12-01) in size (Fig. 3e).
Calcrete is often less than 0.5-1% in an individual paleosol, but a few can be up to 3-5% (paleosol J1z-3-01. Fig. 3b) even 10%
(paleosols J1z-5-02 and 18HC-10).
All above paleosols are defined as relatively mature calcisols (Mack et al., 1993), a kind of aridisol (Soil Survey Staff, 1998;
Retallack, 2001b). The original lithofacies were chiefly argillaceous and silty (split-fan) overbank, interchannel, and flood
plain deposits (Figs. S1 and S2). Some formed landward the paleo-lakeshore.
Dolomites were found at seven loactions in central and southern GSB (Figs. 1, 4, and Table S1). The dolomites chiefly occur
in the Toracian Da'anzhai Member and a few in the Sinemurian-Plienbachian Dongyuemiao and Ma'anshan members (Fig.
4). They are often massive whitish (Figs. 3f and S3e) and micritic (Figs. S4b and S4d), likely indicating an authigenic origin.
Gypsum is recorded in two loactions (Figs. 1, 4, and Table S1). One is located at Zigong (Location A5. SBG, 1980a). The
other lies at Hechuan (Location A6), which can be idientifed by chicken-wire cage structure and is associated with micriditic
dolomites (Fig. 3f).
**4.2. Carbon and oxygen isotope values**
$\delta^{13}C$ values of lacustrine carbonate samples range from -2.02‰ to -4.07‰ and $\delta^{18}O$ values range from -9.91‰ to -12.28‰
(Table S3 and Fig. 5). An increasing trend of both carbon and oxygen isotope ratios is observed from lower to upper
horizons across a 45 m stratal interval of the lower Da'anzhai Member (Fig. 6).
Pedogenic carbonate samples have $\delta^{13}C$ values from -3.52‰ to -8.10‰, which fall in the typical stable isotope range for
pedogenic carbonates. Values of -6‰ to -8.0‰ characterize the sequence of the Zhenzhuchong Member and main
Ma'anshan Member, with an abrupt increase to -5.5‰ to -3.5‰ at the top of Ma'anshan Member (samples J1z-16-01 and
J1z-18-01. Fig. 6). $\delta^{18}O$ values are mainly from -11.3‰ to -13.10‰ in the interval of the Zhenzhuchong Member and
Ma'anshan Member. $\delta^{18}O$ follows $\delta^{13}C$ with a sudden increase to -5.5‰ at the top of the Ma'anshan Member (Fig. 6). Large
and frequent variations of both carbon and oxygen isotope ratios can be observed in the lower Da'anzhai Member (Fig. 6 and
Table S4).
**4.3. CO$_2$ concentrations**
$p$CO$_2$ values based on paleobarometer modelling of paleosol calcite (Cerling, 1999) vary depending on the parameters used
for the calculation.
If $S_{(z)}$=2500 ppmV and $\delta^{13}C_a$=-6.5‰ (constant pre-industrial atmosphere), $p$CO$_2$ values range between ~1140 ppmV and
~3460 ppmV with a mean of 1870 ppmV (column 15 in Table S4); and when $S_{(z)}$=2500 ppmV and $\delta^{13}C_a$=($\delta^{13}C_r$ +18.67)/1.1,
$p$CO$_2$ values change between ~1230 ppmV and ~3260 ppmV with a mean of 2070 ppmV (column 16 in Table S4).
When $S_{(z)}$=2000 ppmV and $\delta^{13}C_s$=-8.98+$\delta^{13}C_c$ are used, $p$CO$_2$ values are ~ 940-2530 ppmV with the mean 1600 ppmV
(column 17 in Table S4); and if $S_{(z)}$=2000 ppmV and $\delta^{13}C_s$ = ($\delta^{13}C_c$+1000) / ((11.98-0.12*25) / 1000+1) -1000 are adopted,
$p$CO$_2$ values become ~980 ppmV to ~2610 ppmV with the mean 1660 ppmV (column 18 in Table S4). Details of the
different parameters and $p$CO$_2$ results can be seen in Table S4.
Results further show that $p$CO$_2$ values at $S_{(z)}$=2500 ppmV are larger than at $S_{(z)}$=2000 ppmV.  The highestdifference is ~
1000 (3640-2610) ppmV, while the lowest is ~300 (1230-930) ppmV and the mean is ~ 370 (2070-1600) ppmV. In addition,
when $S_{(z)}$ is the same, the $p$CO$_2$ values are close even if other parameters are different (comp. between columns 15 and 16,
17 and 18 in Table S4, and Fig. 6).
Whatever parameters used, the trend of $p$CO$_2$ over the epoch is quite similar (Fig. 6). We chose $S_{(z)}$=2000 ppmV (column 18
in Table S4) to illustrate the nature of the Early Jurassic $p$CO$_2$ in the GSB.
$p$CO$_2$ values mostly range between 980 ppmV and 2610 ppmV, and the mean 1660 ppmV is ~6 times the pre-indutrial 275
ppmV. Most of the $p$CO$_2$ values are 1000-2000 ppmV with the mean 1580 ppmV in the Zhenzhuchong and Ma'anshan
members, ~3.5-7.5 times the pre-industrial $p$CO$_2$ value.
It is noted that the errors of $p$CO$_2$ range from 384 ppmV to 1017 ppmV with a mean 647 ppmV (Table S5), leading to a large
uncertainty of the mean ~39%. The largest source of the uncertainty is the standard error (766 ppmV) of modern soil carbonate
(Breecker and Retallack, 2014). The $p$CO$_2$ uncertainty decreases by ~ 20% if half (383 ppmv) of the standard error of soil
carbonate is selected, and decreases to ~12% if 1/4 (~191 ppmV) standard error is used. The second largest source of error in
the $p$CO$_2$ is the $S_{(z)}$ estimate. The uncertainty of $p$CO$_2$ becomes much smaller when the $S_{(z)}$ is larger, e.g., it will fall from ~39%
to ~17% if $S_{(z)}$=5000 ppmV instead of 2000 ppmV. Other parameters such as temperature, $\delta^{13}C_r$, $\delta^{13}C_a$, $\delta^{13}C_s$, contribute very
little to the calculated $p$CO$_2$ uncertainty.
**5. Discussion**
The Jurassic marine record shows climatic and environmental oscillations (e.g., van de Schootbrugge et al., 2005; Dera et al.,
2011; Gómez et al., 2015; Arabas et al., 2017), including sea water temperature fluctuation and carbon cycle reorganization
recorded in both carbonate and organic matter. The climate changes and events recorded in the the marine realm have been
mainly attributed to Karoo-Ferrar volcanism (e.g., Hesselbo et al., 2000; Caruthers et a., 2013), sea-level change (e.g.,
Hesselbo and Jenkyns, 1998; Hallam and Wignall, 1999), orbital forcing (e.g., Kemp et al., 2005; Huang and Hesselbo, 2014,
Storm et al., 2020), and / or the opening of the Hispanic corridor (e.g., van de Schootbrugge et al., 2005; Arias, 2009).
Eruption of the Karoo-Ferrar and Central Atlantic mgama is thought to have released large amounts of CO$_2$ into the
atmosphere in a short amount of time, resulting in rising temperatures of both marine and continental realms. The nearly
continuous record of Jurassic strata in the GSB provides an excellent test of this hypothesis in the terrestrial realm. We
compare the climate and $p$CO$_2$ record from the GSB in relationship to the marine temperature records.
**5.1. Paleoclimate variation**
During the Late Triassic, Southwest China was warm-hot and humid and occupied a tropical and / or subtropical zone, as
demonstrted by palynoflora, coals, and perennial riverine and lacustrine lithofacies in the Xujiahe Fm (e.g., Huang, 1995; Li
et al., 2016). However, the climate became dry through the Early Jurassic manifested by climate-senstive sediments and
stable isotopes albeit there are two lithofacies packages reflecting two major lake stages (for details refer to supplementary
data Note S1) in the GSB.

**5.1.1 The Hettangian Age**

In the Hettangian, the climate was warm-humid like the Late Triassic in the GSB. The Qijiang Member is comprised of mainly mature quartz arenites and siltstones with coals (Fig. 7) as well as siderite concretions, indicating a stable tectonic setting and warm-humid climate in the eastern and southern GSB. Climate was similar across the whole region, because multiple coal layers occur in the lower Baitianba Fm. The alluvial fan system of the lower Baitianba Fm. (Figs. 7 and S6) is characterized by moderate-good roundness and sorting of gravels with sandy matrix (Fig. S3a. e.g., Liu et al., 2016; Qian et al., 2016; and this work). In the Newark basin of eastern North America, climate-sensitive sediments such as nodules of carbonate and gypsum (pseudomorph) as well as mudcrack in mudflat facies indicate an arid climate in the fifth cycle of the Hettangian (>199 Ma) Passaic Fm (Kent et al., 2017). More widespread, the eolian Navajo Sandstone, dated as Hettangian-Sinemurian (200-195 Ma. Parrish et al., 2019), indicates arid in Colorado Plateau (Fig. 1a. Boucot et al., 2013).

**5.1.2 The Sinemurian Age**

The early Sinemurian Zhenzhuchong Member is a combination of riverine flood plain and lacustrine facies (supplementary Note S1). The lithology is dominated by violet-red mudrocks with few thin greyish, greenish fine sandstones and siltstones. The reddish color of rocks may indicate a change of climate. Differences in the color appearance show that the reddish color started in the middle member in the central basin (Location A6. Fig. S2) but almost developed through the whole member in the western basin (Location A4. Fig. 6).

Within reddish mudrocks of the flood plain facies, multiple calcisols were observed at the Shaping section, Ya'an (Location A4. Figs. 1, 4, and 7), including a strongly leached calcisol horizon (Fig. S3c). We also interpret the reddish muddy sediments with abundant calcretes as the calcisol at sections of Dafang (Location A8. Zhang et al., 2016), Tianzhu (Location A9. Li and Chen, 2010), and Weiyuan (Location A10. SBG, 1980a). The calcisols indicate a (semi-) arid climate in the Sinemurian.

This climate change, interpreted from reddish mudrocks and paleosols, is consistent with the floral fossils (e.g., Huang, 2001; Wang et al., 2010), suggesting the decreasing humidity and increasing temperature from the Late Triassic epoch and the Hettangian age into the Sinenmurian age in the southern GSB. However, in the northern GSB there are few proxies for climate change, and alluvial fan and lacustrine delta facies common in the middle Baitianba Fm (Fig. S6. e.g., Qian et al., 2016) do not give us information on climte.

The late Sinemurian Dongyuemiao Member also has reddish mudrocks and calcisols, similar to the Zhenzhuchong Member. Pedogenic calcretes were reported at Dafang (Location A8. Zhang et al., 2016), Tianzhu (Location A9. Li and Chen, 2010), and Yunyang (Location A15. Meng et al., 2005) and in the central and southern GSB (Figs 4 and 7 and Table S2), indicating

continued arid climate conditions at the time.
The interpreted Sinemurian (semi-) arid climate from reddish mudrocks and calcisols is supported by the flora (Li and Meng,
2003) and the mudrock geochemistry (Guo et al., 2017). Few records of coeval terrestrial climate are known from other
continents or regions in the literature. The Whitmore Point Member of the Moenave Fm deposited in dryland lakes (Tanner
and Lucas, 2008) and the upper part of eolian Navajo Sandstone (Blakey et al., 1988) could represent the coevally similar
climate in Colorado Plateau although relatively cool (~9 to 18 ºC) continental climate was inferred from oxygen and
hydrogen isotope composition of chert precipitated in interdune, freshwater lakes in the Navajo Sandstone (Kenny, 2015).
With a difference, in eastern England, the co-occurrence of the acmes of thermophilic pollens *Classopollis classoides* and
*Liasidium variabile* indicates the warm-humid climate in the late Sinemurian (Riding et al., 2013).

### 5.1.3 The Pliensbachian Age

The Ma'anshan Member of the Pliensbachian displays a prominent change in the distribution and extent of red color
sediment and pedogenesis. The reddish sediments extend through the entire member (comp. Figs. 6 and S2) and can be
observed across most of the GSB. Calcisols are documented in both the western and central GSB (Figs. 6, 7, S1, and S2).
Ten calcisol horizons were recognized at the Shaping section, Ya'an (Figs. 6 and S1). Strongly leached pedogenic structures
and mudcracks are seen in Bed H8 of the Tanba section, Hechuan (Fig. 3a and 3d). Abundant calcretes within terrestrial red
mudrocks are widely described at Gaoxian of Dafang (Location A8. Zhang et al., 2016), Hulukou of Weiyuan (Location A10.
SBG, 1980a), Geyaoguan of Gulin (Location A13. SBG, 1976), Taiyuan of Fengdu (Location A16. SBG, 1975), and Yaxi of
Zunyi (Location A17. Yang, 2015). The widespread distribution of redbeds and calcisols (Figs. 4 and 7) denotes an
intensification of the (semi-) arid climate.
Plant and sporopollen fossils also show a change to drier climate in the Pliensbachian. Compared to the Sinemurian members,
more plant fossils are reported in this member (e.g., Meng and Chen, 1997; Wang et al, 2010). The Pliensbachian-Toarcian
sporopollen assemblages are dominated by sporomorph genera assemblage *Dictyophyllidites-Cyathidites-Classopollis*, in
which the dry-type gymnosperm spore *Classopollis* is more prevalent than in the Hettangian-Sinemurian (Zhang and Meng,

318  1987).

Similar dry temperate / subtropical climate is interpreted for the upland coniferous forest in Qaidam Basin, Northwest China
(Wang et al., 2005) and by interdune playa mudstones of the Kayenta Fm in Colorado Plateau (e.g., Bromley, 1992) albeit it
was a cool-humid climate in South Kazakhstan, central Asia (Tramoy et al., 2016).

### 5.1.4 The Toarcian Age

In spite the fact that the Da'anzhai Member was deposited in the largest lacustrine transgression period (Fig. 7. details see
supplementary data Note S1), abundant evidence for arid conditions, including backshore reddish mudrocks with calcisols,

lacustrine micritic dolomites and / or gypsum, and stable isotopic geochemistry of lacustrine carbonate, indicate that the Toarcian aridification could be the most intensive of the late Early Jurassic in the GSB.

Redbeds with abundant calcretes are well developed in this member (Figs. 4 and 7). Four calcisols in the Shaping section (Figs. 6 and S1) and the leaching/illuvial structure (Bed H13) in the Tanba section (Fig. 3c) were observed. Calcisols with calcretes also occur at sections of Dafang (Location A8. Zhang et al., 2016), Nanxi (Location A11. SBG 1980a), Gongxian (Location A12. Liang et al., 2006), and Yunyang (Location A15. Meng et al., 2005). The widespread occurance of calcisols within the lacustrine facies reveals that subaerial exposure of sediments often interrupted the lake environment, illustrating dynamic lake level fluctuations and an arid climate.

Gypsum and micritic dolomites are reported in the western and southern GSB (SBG, 1980a; Mo and Yu, 1987; Peng, 2009; and this work) (Figs. 1, 4, and 7). Though there are a number of hypothesies on the dolomite formation in deep time, such as authigenic origin, diagenetic replacement, microbial mediation (e.g., Vasconcelos et al., 1995; Mckenzie et al., 2009; Petrash et al., 2017), a high abundance of dolomite was interpreted to form during greenhouse periods, characterized by warm climates, probably reflecting favourable conditions for evaporite deposition and dolomitization via hypersaline reflux (Warren, 2000). Dolomites are aslo thought the results of interplay of climate and sea-level / base-level change (e.g., Newport et al., 2017) or are interacted with climatic regimes (Vandeginste et al., 2012). The widespread micritic dolomites in the Da'anzhai Member, which are associated with gypsum (Fig. 3f), likely indicate an arid climate in the central and western GSB (Fig. 1b). Gypsum occasionally occurs at Maliuping of Hechuan (Fig. 3f) and Wujiaba of Zigong (SBG, 1980a), showing a possible evaporitic climate in the early Toarcian in the central GSB.

Carbon and oxygen isotopes of lacustrine carbonates further support the interpretation of an aird climate in the Toarcian age in the GSB. The mainly positive $\delta^{13}C$ values 0 to 2 ‰ (Fig. 5) from Hechuan (Wang et al., 2006) indicate the lakes were brackish or even saline. The relatively heavy negative $\delta^{13}C$ values -1‰ to -3.5 ‰ (Fig. 5) from Zigong (Wang et al. 2006) and Ya'an (this work) denote low depletions of $^{13}C$ during calcite/aragonite precipitation and mean that the lakes were possibly brackish. Lightly negative $\delta^{18}O$ values -5‰ to -12 ‰ (Fig. 5) of the lacustrine carbonates, suggest closed lacustrine, palustrine and pond systems formed in a regional arid-semiarid climate with evaporation exceeding precipitation.

The covariance of $\delta^{13}C$ and $\delta^{18}O$ is a criterion to distinguish closed or open lakes (e.g., Talbot, 1990; Li and Ku, 1997). Pronounced positive covariances ($R^2$=0.44-0.96) between carbon and oxygen isotopes (Fig. 5) indicate a typical arid-semiarid pattern of lakes in the central and western GSB.

The Da'anzhai Member has the same palynofloral assemblage with the Ma'anshan Member, in which the dry-type gymnosperm spore *Classopollis* is more abundant than in underlying strata (e.g., Zhang and Meng, 1987; Wang et al., 2010), supporting the aridification indicated by climate–sensitive sediments and stable isotope ratios of lacustrine carbonates aforementioned.

Coastal Cheirolepidiacean (gymnosperm) forests indicate (temperate to subtropical) warm-humid climate punctuated by

locally dry and/or arid events in the Toarcian in Qaidam Basin, Northwest China (Wang et al., 2005). In Inner Mongolia of North China, the thermophilous plants such as the dipteridaceous fern *Hausmannia*, bennettitales *Ptilophyllum*, display similar warm and humid climate interrupted by hot and even arid conditions in a short intervals of the Toarcian (Deng et al., 2017). The warm-wet climate was also indicated by assemblages of sporomorph and vegetation in the late Early Jurassic in Jurong of Jiangsu, Lower Yangtze area (Huang et al., 2000). In South Kazakhstan, central Asia, paleoflora and $\delta^2H$ values suggest slightly less humid and warmer conditions starting from the early Toarcian (Tramoy et al., 2016).

Climate–sensitive sediments, carbon and oxygen isotope values and covariance, and palynoflora, together indicate that an overall (semi-) arid climate dominated the GSB during the Early Jurassic, possibly accompanied by occasional evaporitic climate. Relatively abundant calcisols suggest that the GSB was in a subtropical arid zone based on the paleoclimatic zonation model of paleosols (Mack and James, 1994) during the middle-late Early Jurassic. Through the Early Jurassic, this (semi-) arid climate in GSB is thoroughly comparable with the simultaneous arid climate recorded in dryland lacustrine and eolian facies in Colorado Plateau (e.g., Blakey et al., 1988; Bromley, 1992; Tanner and Lucas, 2008; Parrish et al., 2017), but distinct from the relatively warm-humid climate indicated by sedimentological and floral characteristics in North China (e.g., Wang et al., 2005, Deng et al., 2017) and in the relatively high latitudes of Southern Hemisphere (Pole, 2009).

In summary, the increasing aridity and warming in the GSB and arid climate in the Corlorado Plateau could have been consecutive through the Early Jurassic, and seems not harmonizaed with the global fluactueated climate that could be imprinted by two large volcanic eruptions of the Central Atlantic magmatic province and Karro-Ferrar Large Igneous Province. The secular arid climate in the two areas might be more possibly constrained by paleotopography, where both were laid in the relatively low latitudes 15-30˚N (Fig. 1a).

## 5.2. *p*CO$_2$ perturbations and events

Pedogenic carbonates found in various continental settings precipitate in direct contact with soil atmosphere and bed rock and hold a meaningful signature of past climate (Alonso-Zarza and Tanner, 2006). There are few high age resolution *p*CO$_2$ reconstructions for the Early Jurassic. The focus on *p*CO$_2$ estimates has on the event horizons, such as the transition of the Triassic to Jurassic (e.g., Tanner et al., 2001; Schaller et al., 2011). Herein we present a *p*CO$_2$ estimate based on data from the GSB at ~1.0 Myr age resolution for ~20 Myr (199-179 Ma) interval of the Early Jurassic (Figs. 6 and 8a).

### 5.2.1. *p*CO$_2$ perturbation

Results of model estimates show that the *p*CO$_2$ values range 980-2610 ppmV with a mean 1660 ppmV in the Early Jurassic post the Hettangian and can be divided into three intervals (Figs. 6 and 8a): phase I, stable 1500-2000 (mean ~1700) ppmV in the Zhenzhuchong and Dongyuemiao members (Sinemurian age); phase II, main 1000-1500 (mean ~ 1300) ppmV in the Ma'anshan Member (Pliensbachian age); and phase III, great fluctuation 1094-2610 (mean ~1980) ppmV in the lower

Da'anzhai Member (early Toarcian age).
The evolution and level of $pCO_2$ estimated by carbon isotope ratios of the pedogenic carbonates from the GSB compare
favorably with the global composite based on the plant stomata method (data of the composite curve see Table S6), but show
significant differences relative to the global composite $pCO_2$ based on paleosols (Fig. 8a. Suchecki et al., 1988; Cerling,
1991; Ekart et al., 1999), which may be attributed to the shortage (<4 samples) of global data and large age uncertainties (Fig.
8a and Table S5 and S6).
The changes in $pCO_2$ from the GSB, has a similar pattern to coeval seawater temperature estimates through the Early
Jurassic although there are some discrepancies in pace and in detail (comp. Fig. 8a and 8b). That is, the relatively high $pCO_2$
1500-2000 ppmV approximately corresponds to the relatively high seawater mean temperature -2ºC to +2ºC in the
Sinemurian, low $pCO_2$ 1000-1500 ppmV corresponds to low seawater mean temperature -5ºC to -2ºC in the Pliensbachian,
and quick rising $pCO_2$ of 1200 ppmV to ~2500 ppmV corresponds to the rapidly increased seawater temperature of -4ºC to
+4ºC in the late Pliensbachian-early Toarcian.
The $pCO_2$ record roughly trends with the carbon isotope records of marine carbonates and oganic matter (comp. Fig. 8a to
8d), suggesting a possible linkage of the $pCO_2$ record in the GSB to the global carbon cycle (see section 5.2.2). Nevertheless,
it is difficult for the proxies to compare in a higher detail, making it difficult to relate the record to orbital forcing of the
global carbon cycle in the Sinemurian-Pliensbachian (Storm et al., 2020).
As a greenhouse gas, atmospheric $CO_2$ has a strong control over global temperatures for much of the Phanerozoic (e.g.,
Crowley and Berner, 2001; Royer, 2006; Price et al., 2013), but a decoupling of $CO_2$ and temperature has also been
suggested (e.g., Veizer et al., 2000; Dera et al., 2011; Schaller et al., 2011). The pattern of the Early Jurassic $pCO_2$
reconstructed from the carbon isotope of pedogenic carbonates in GSB, Southwest China, supports the coupled relationship
of $CO_2$-temperature. Models of the coupling and decoupling of $CO_2$-temperature and $CO_2$-carbon cycle have to consider: 1),
age order of $CO_2$-temperature/carbon cycle relevance, i.e. they should be related in the same age (long term or short term)
hierarchy; 2) precise age constraints of individual $CO_2$ and temperature data; 3) methods of $CO_2$ and temperature estimates,
depending on precondition, presumptions, parameters, uncertainty, sample diagenesis, etc.; 4) controls or influences of key
factors such ice sheet, tectonic, paleogeography, cosmic ray flux, biota, volcanic eruption, and so on.
**5.2.2. Rapid $pCO_2$ falling events**
The GSB Early Jurassic $pCO_2$ curve reveals two rapid falling events (Fig. 6 and 8a). The first event ($1E_{CO2}$) shows a quick
drop from ~2370 ppmV (sample J1z-08-01 at depth 84.7 m) to 1350 ppmV (sample J1z-10-02 at depth 94.4 m) near the
boundary of the Dongyuemiao and Ma'anshan Members (Fig. 6), or to 1075 ppmV (sample J1z-11-02 at depth 111.7 m),
which took place in the early Pliensbachian (~190.4-189.9/189.1 Ma. Fig. 8c). The extent of the rapid falling $pCO_2$ is
~1000-1300 ppmV in 9.7-17.0 m. In other words, ~1000 ppmV drop could be accomplished within ~0.5-1.0 Myr based on
the estimate of sedimentation rate (Table S4).
While the corresponding early Pliensbachian climatic and isotopic-shifting events are not observed in the smoothed curves of
the Early Jurassic seawater temperature and carbon cycle (Dera et al. 2011), the rapid falling event $1E_{CO_2}$ is well correlated
to the nearly coeval excursion events of carbon and oxygen isotopes recorded in western Tethys and North Atlantic (Fig. 8).
The $1E_{CO_2}$ compares well to: 1) the rapid carbon isotope negative excursion of (oysters, belemnites, and brachiopods) shells
from the Cleveland Basin, UK (Korte and Hesselbo, 2011) and northwest Algeria (Baghli et al., 2020), 2) that of organic
matter and marine carbonates from southern Pairs Basin (Bougeault et al., 2017; Peti, et al., 2017) and Cardigan Bay Basin,
UK (Storm et al., 2020), and 3) rapid oxygen isotope negative excursion (seawater warming) of belemnites from northern
Spain (van de Schootbrugge et al., 2005). The rapid change of the stable isotope record had been called the
Sinemurian-Pliensbachian boundary event (SPBE) and dated in the ammonite of the upper *Raricostatum* - lower *Jamesoni*
zones (Bougeault et al., 2017).
The second event $2E_{CO_2}$ displays a large drop of 2574 ppmV (sample J1z-18-01 at depth 252.7 m) to 1094 ppmV (sample
J1z-19-01 at depth 272.3 m), ~1500 ppmV decrease within 19.6 m (estimated age interval ~0.8 Myr. Table S4 and Fig. 8a).
Following the second drop, $p$CO$_2$ rises rapidly by ~1300 ppmV of 1094 ppmV to 2386 ppmV (sample J1Z-20-01 at depth
294.3 m) although only a few samples support the this cycle of $p$CO$_2$ falling-rising.
Strata in western Sichuan (Xu et al., 2017), may correlate to the time interval of the T-OAE, during which $p$CO$_2$ doubled
over background values, from ~1000 ppmV to ~2000 ppmV (e.g., Beerling and Royer, 2002; McElwain et al., 2005; Berner,
2006). Given that chronostratigraphical correlation is challenging, the $p$CO$_2$ falling-rising cycle might correspond to the
quick shifting cycle of stable isotopes during the T-OAE (Fig. 8a and 8c-8d). In detail, the rapid falling-rising of $p$CO$_2$ is
consistent with: 1) the quick negative-positive carbon isotope excursion of marine carbonates from Italy (Jenkyns and
Clayton, 1986; Sabatino et al., 2009), England and Wales (Jenkyns and Clayton, 1997), north Spain (van de Schootbrugge et
al., 2005), the Lusitanian Basin of Portugal (Hesselbo et al., 2007), Paris Basin (Hermoso et al., 2009), and Morocco (Bodin
et al., 2016); 2) that of invertebrate calcareous shells from the Cleveland Basin of UK (Korte and Hesselbo, 2011) and
northwest Algeria (Baghli et al., 2020); 3) that of marine organic matter from Morocco (Bodin et al., 2016), Yorkshire of
England (Cohen et al., 2004; Kemp et al, 2005), Cardigan Bay Basin of UK (Xu et al., 2018), northern Germany (van de
Schootbrugge et al., 2013), Alberta and British Columbia of Canada (Them II et al., 2017), northern Tibet of China (Fu et al.,
2016), and Japan (Izumi et al., 2018); 4) that of terrestrial organic matter from Sichuan Basin, China (Xu et al., 2017); and 5)
quick oxygen isotope negative-positive shifting (seawater warming) of brachiopods (Suan et al., 2008) and fossil wood
(Hesselbo et al., 2007) from the Lusitanian Basin, Portugal.
Multiple hypotheses have been proposed to interpret the 5°–6 °C decrease of sea surface temperatures in the late
Pliensbachian (Bailey et al., 2003; van de Schootbrugge et al., 2005; Suan et al., 2010) and warming ~8 °C in the early
Toarcian (Bailey et al., 2003; Suan et al., 2010), such as the sea level falling and rising, methane release, Karoo–Ferrar
eruption, Hispanic corridor opening, etc. Perhaps, these hypotheses somewhat explain the rapid change of sea surface
temperatures, but might not link to drastic falling of $pCO_2$. As we know, atmospheric $CO_2$ is controlled by volcanism,
weathering, vegetation on land and phytoplankton in ocean, and orbiting forcing. The Sr isotope curve shows a rapid change
in the early Toarcian but does not in the early Pliensachian (e.g., Jones et al., 1999), indicating a distinct transfer of
weathering took place on the land only at the T-OAE time. No robust evidence shows the rapid changes of terrestrial
vegetation and marine primary productivy for the two intervals excepet for the floral change in western Tethys during the
T-OAE (Slater et al. 2019). The Karoo–Ferrar eruption could be responsible for the rapid rising of $pCO_2$ but not for the
falling. Then the orbital forcing might be an alternative.
To sum up, the rapid falling events of the Early Jurassic $pCO_2$ values in the GSB, are compatible with the response of stable
isotopes (carbon cycle) and seawater temperature from coeval marine sediments in a total tendency and eventful change, but
not harmonized at a high-resolution time scale. Whatever caused the rapid variations of sea surface temperatures, stable
isotopes, and $pCO_2$, their near concordance suggests that it is a positive feedback of the sea surface temperature and carbon
cycle to the $pCO_2$ in trend and event through the Early Jurassic; whereas the higher frequency changes in the
Sinemurian-Pliensbachian might may support other causal driving of the climate, such as orbital forcing (Storm et al., 2020).
**6. Conclusions**
Based on analyses of climate-sensitive sediments and stable isotopes and the reconstruction of paleoclimate and $pCO_2$, we
conclude:
1) An overall warm-hot and (semi-) arid climate dominated the GSB during the Early Jurassic, possibly accompanied by
occasional evaporitic climate in the Toarcian. This (semi-) arid climate in GSB is comparable with that in Colorado Plateau,
western America, but distinct from the relatively warm-humid terrestrial climate recognized in other places of Chinese
mainland (e.g., Qaidam, Inner Mongolia, and Lower Yangtze) and the high latitudes of Southern Hemisphere.
2) The Early Jurassic $pCO_2$ values show that a range between 980 ppmV and 2610 ppmV is ~3.5-10 times the pre-industrial
value 275 ppmV and the mean 1720 ppmV is ~6 times the pre-industrial value. Three phases of $pCO_2$ values were
distinguished: 1500-2000 (mean ~1700) ppmV in the Sinemurian age, 1000-1500 (mean ~ 1300) ppmV in the Pliensbachian
age, and 1094-2610 (mean ~1980) ppmV in the early Toarcian. Two events of rapidly falling $pCO_2$ were also recognized:
~1000-1300 ppmV drop at the Sinemurian-Pliensbachian boundary and quick falling (-rising) by ~1500 ppmV in the early
Toarcian. The phases and events manifest the perturbation of $pCO_2$ in the Early Jurassic.
3) The perturbation and rapid falling events of the Early Jurassic $pCO_2$ from the GSB are compatible with the carbon cycle
and seawater temperature from coeval marine sediments in the North Atlantic and western Tethys in a total tendency and
eventful change. The compatibility suggests that it is a positive linkage of the sea surface temperature and carbon cycle to
the $p$CO$_2$ through the Early Jurassic. On the contrary, differences at a high-resolution time sacle implies additional climate
drivers, such as orbital forcing are important in the Sinemurian-Pliensbachian record.
**Acknowledgements**
We thank Professors Helmut Weissert and Dan Breecker for careful scrutiny, constructive comments and suggestions. It is
acknowledged this research was supported by Natural Science Foundation of China (NSFC) project 41672097.

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

**Figure 1** **A, Global Early-Middle Jurassic climate zones (Boucot et al., 2013) laid on the Early Jurassc (~193 Ma, Sinemurian)**
**paleogeographic map (Scotese, 2014). B, Lithofacies paleogeographic sketch of the grand Sichuan paleobasin (GSB) in the early**
**Early Jurassic (Zhenzhuchong and Dongyuemiao members) showing locations of the observed and analysed sections and**
**climate-sensitve sediments. Lithofacies paleogeographic map was composed and modified from Ma et al. (2009) and Li and He**
**(2014). Blue area is the extent of paleolake, estemted as ~380,000 km$^2$; blue + gray region is the basin shape, eitmated ~480,000**
**km$^2$. Dot red line confines the deeper lake area in the late Early Jurassic (Ma'anshan and Da'anzhai members). Bold dashed line is**

the northern edge of calcisol occurrence, which may separate the climate of the GSB as the northern and southern types. Triangles with numbers are locations of observed and analysed sections: A1, Xiasi section, Jian'ge; A2, Puji section, Wangcang; A3, Shiguansi section, Wanyuan; A4, Shaping section, Ya'an (bed and thickness from Wen and Zhao, 2010); A6, Tanba and Maliping section, Hechuan (bed and thickness from Wang et al., 2010); A7, Wenquan section, Kaixian (thickness from Wang et al., 2010). Location and source data of sections A5 and A8-A17 (climate-sensitive sediments) refer to supplementary data Table S1.

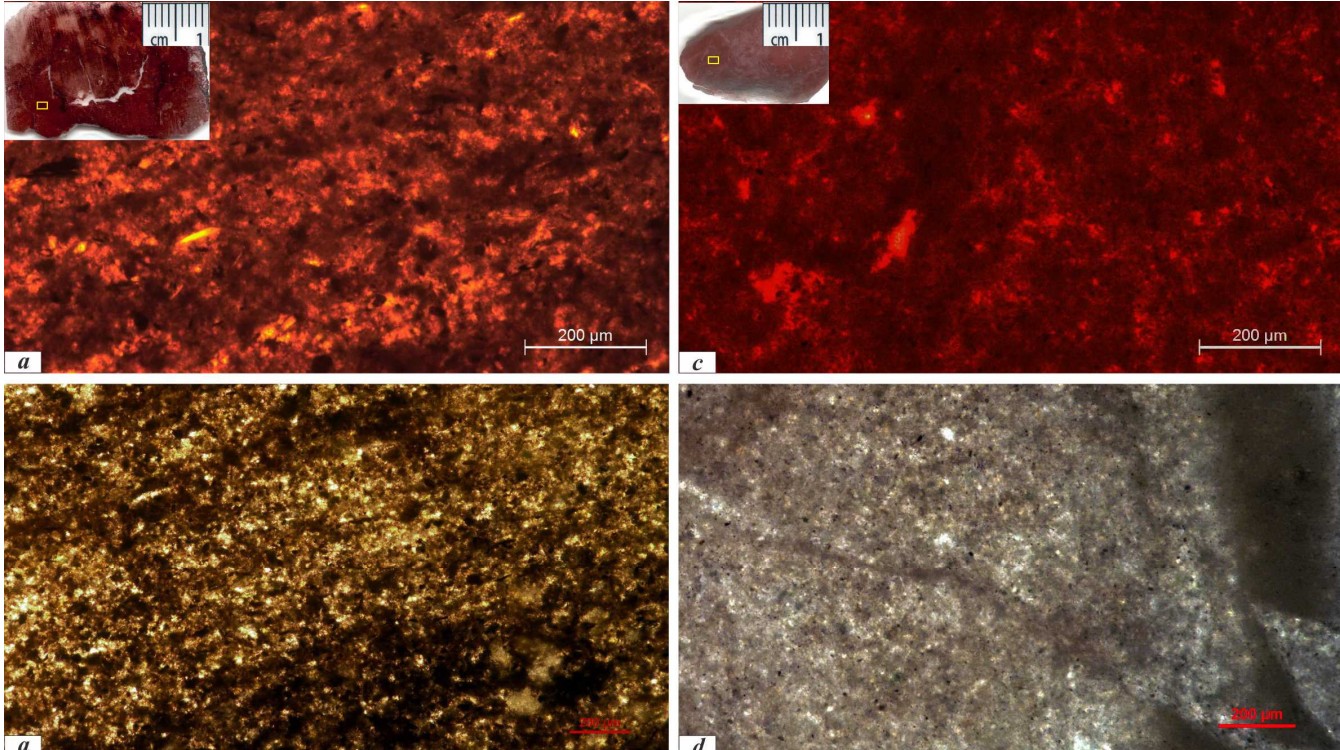

Figure 2    Microscopic cathodoluminescence photos of representative calcrete samples from the Ziliujing Fm at the Shaping section, Ya'an. *a*, Sample $J_1z$-12-01, Bed B12, Ma'anshan Member; *b*, Sample $J_1z$-22-01, Bed B22, Da'anzhai Member. Pedogenic calcites are mainly null to non-luminescent, minor are orange/red luminescence. Inserts are the scanned photos of thin-section, and rectangles are the area under cathodoluminescence and drilling.

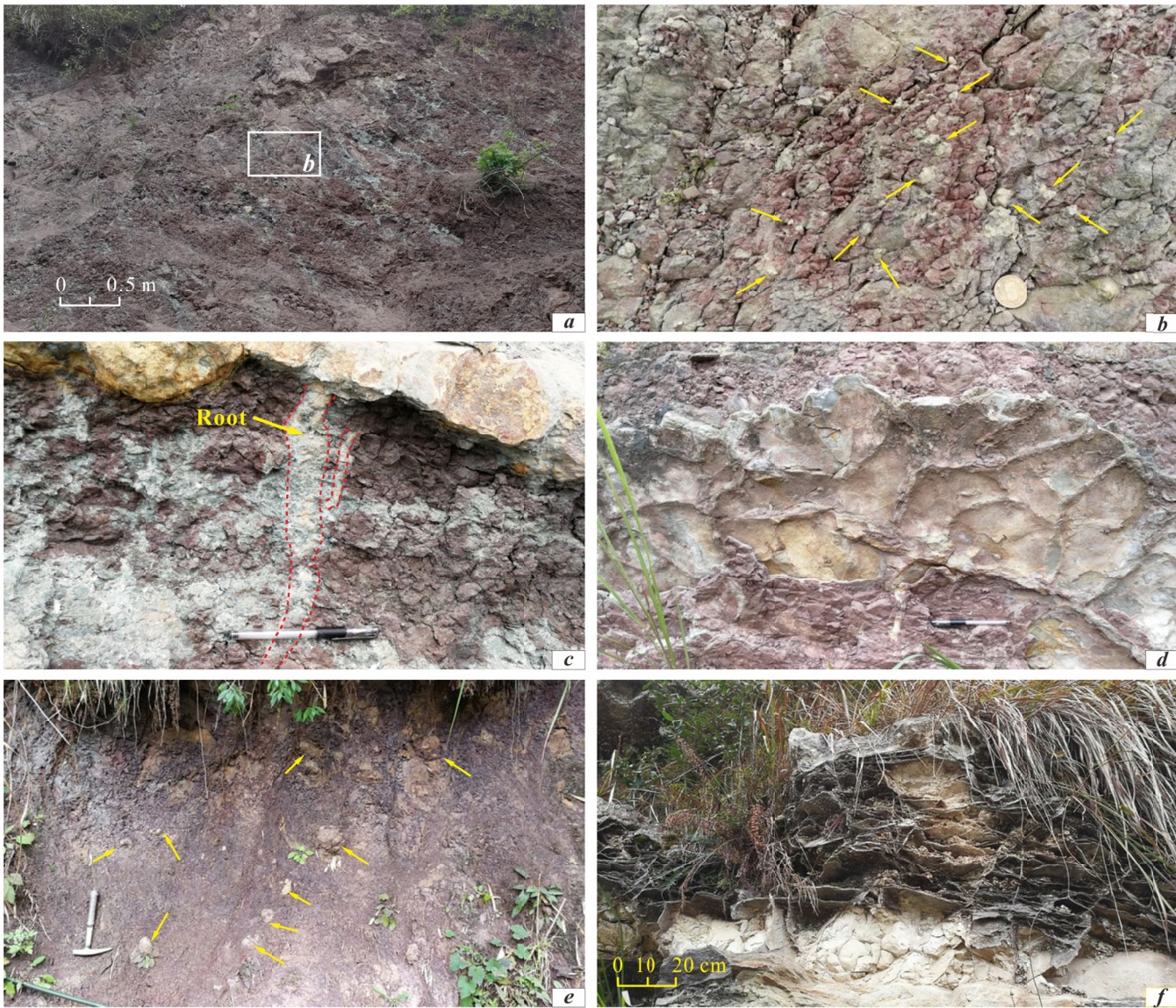

**Figure 3  Field photographs of climate-sensitive sediments from the Lower Jurassic Ziliujing Fm in the GSB.** *a*, Reddish purple calcisol with strong leaching structure. Lower Bed H8 of the upper Ma'anshan Member at Tanba, Hechuan. *b*, Reddish purple calcisol showing the density and size of calcretes. The horizon and location same as *a*. Arrows point to calcretes. Coin 2.0 cm in diameter. *c*, Reddish purple calcisol with strong leaching structure and rhizoliths. Bed H13 of the top Ma'anshan Member at Maliuping, Hechuna. Pen 15 cm long. *d*, Mudcracks. Lower Bed H8 of the upper Ma'anshan Member at Maliuping, Hechuan. Pen 15 cm long. *e*, Brownish red calcisol with big calcretes (calcareous concretions). Arrows point to big calcretes. Calcisol horizon J₁z-10-01, Bed B10 of Ma'anshan Member at Shaping, Ya'an. Hammer 34 cm long. *f*, Chicken-wire structure. Bed H12 of the Da'anzhai Member at Maliuping, Hechuan.

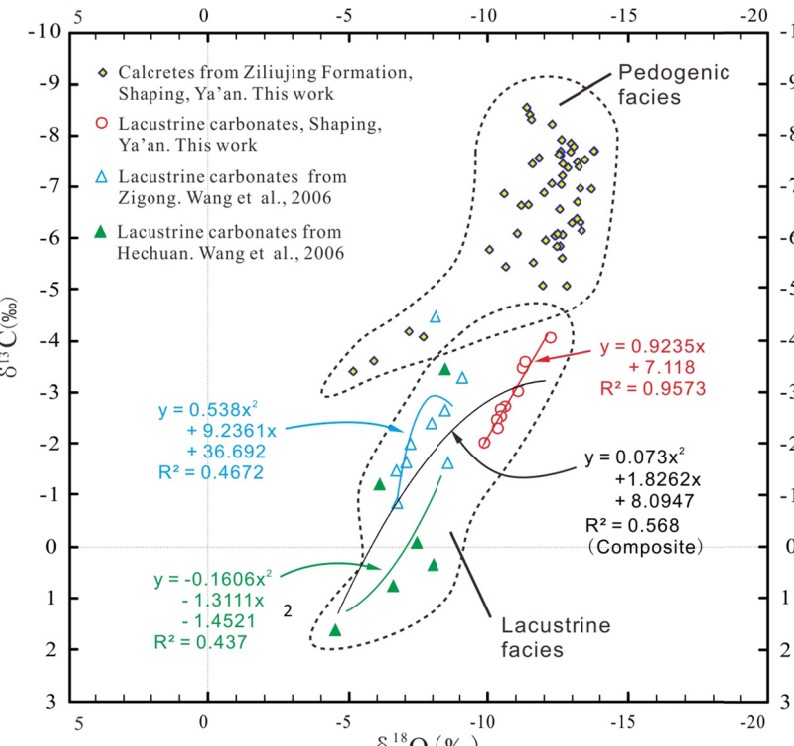

**Figure 4** Diagram showing the temporal and spatial variation of climate-sensitive sediments in GSB. Section loactions and data
sources refer to Table S1.


**Figure 5** Cross-plot and covariance of carbon and oxygen isotopic values of the Lower Jurassic pedogenic and lacustrine
carbonates from the GSB. Note, the pronounced covariance ($R^2$=0.957) between $\delta^{13}C$ and $\delta^{18}O$ from Shaping section, Ya'an,
indicating a compositional arid-evaporate and closed pattern lake; the moderate covariance ($R^2$=0.47 and 0.44) between $\delta^{13}C$ and
$\delta^{18}O$ from Zigong and Hechuan, indicating a (semi-) arid and semi-closed pattern lake.



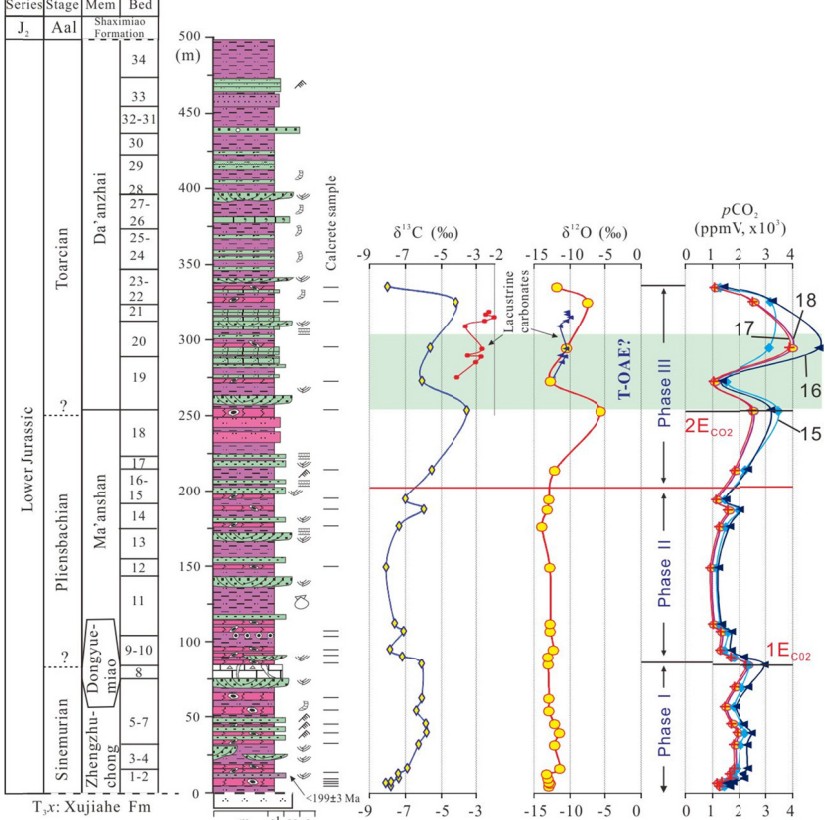

Figure 6 Diagram of the Lower Jurassic strata and litholigcal log at the Shaping section, Ya'an with carbon and oxygen isotope values of pedogenic and lacustrine carbonates and $p$CO$_2$ cruve. Three phases and two events can be observed for both stable isotope values of pedogenic carbonates and $p$CO$_2$ estimate. Legend of lithology in log refers to supplementary Figs. S1 and S2. T-OAE, Toarcian oceanic anoxic event. 1E$_{CO2}$ and 2E$_{CO2}$, rapid falling event of $p$CO$_2$. Numbers 15 to 18 are the curves of $p$CO$_2$ in different parameters, and details refer to supplementary Table S4.

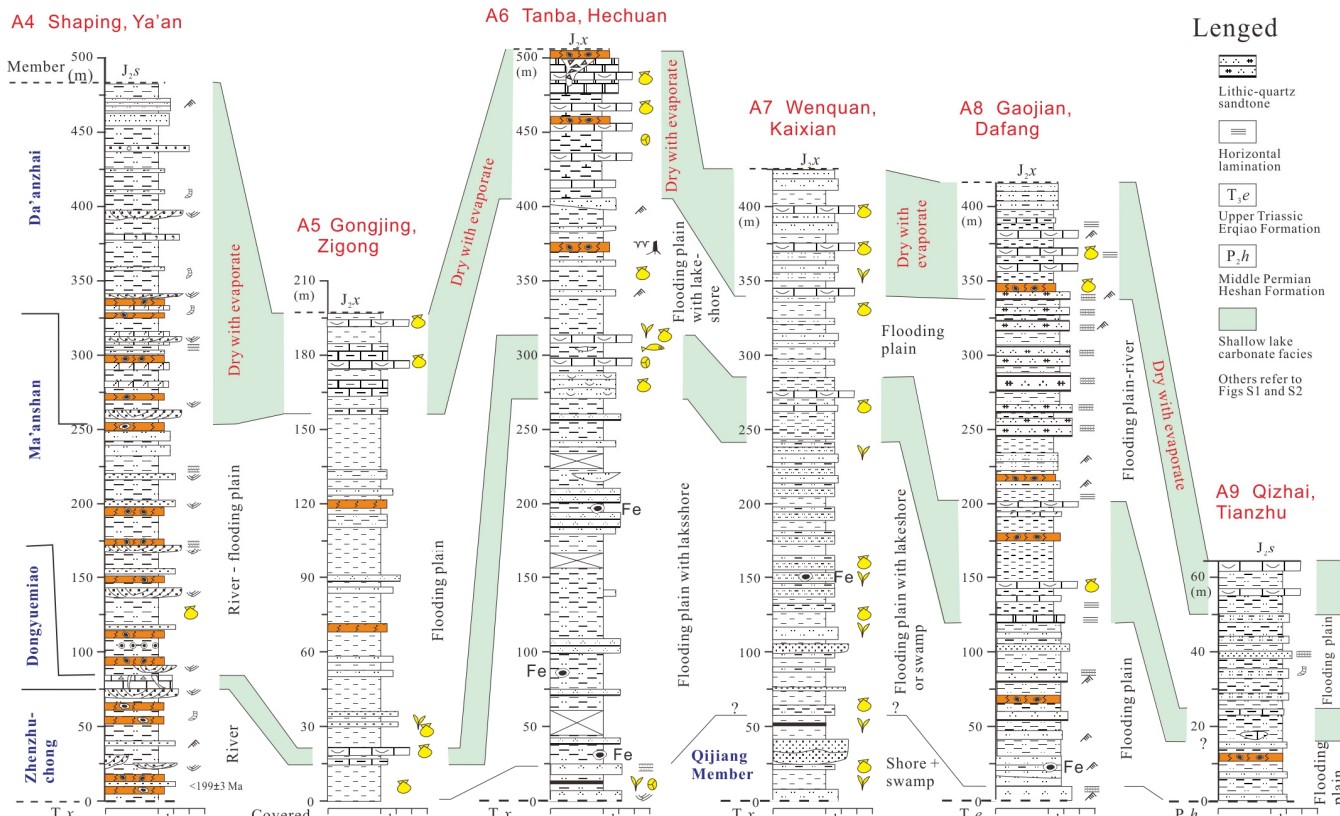

**Figure 7  Stratigraphic correlation and depositional environment interpretation of the Lower Jurassic in the GSB. Data of sections refer to Fig. 1. Note, two lacustrine transgressive cycles are marked by correlative pale green areas.**

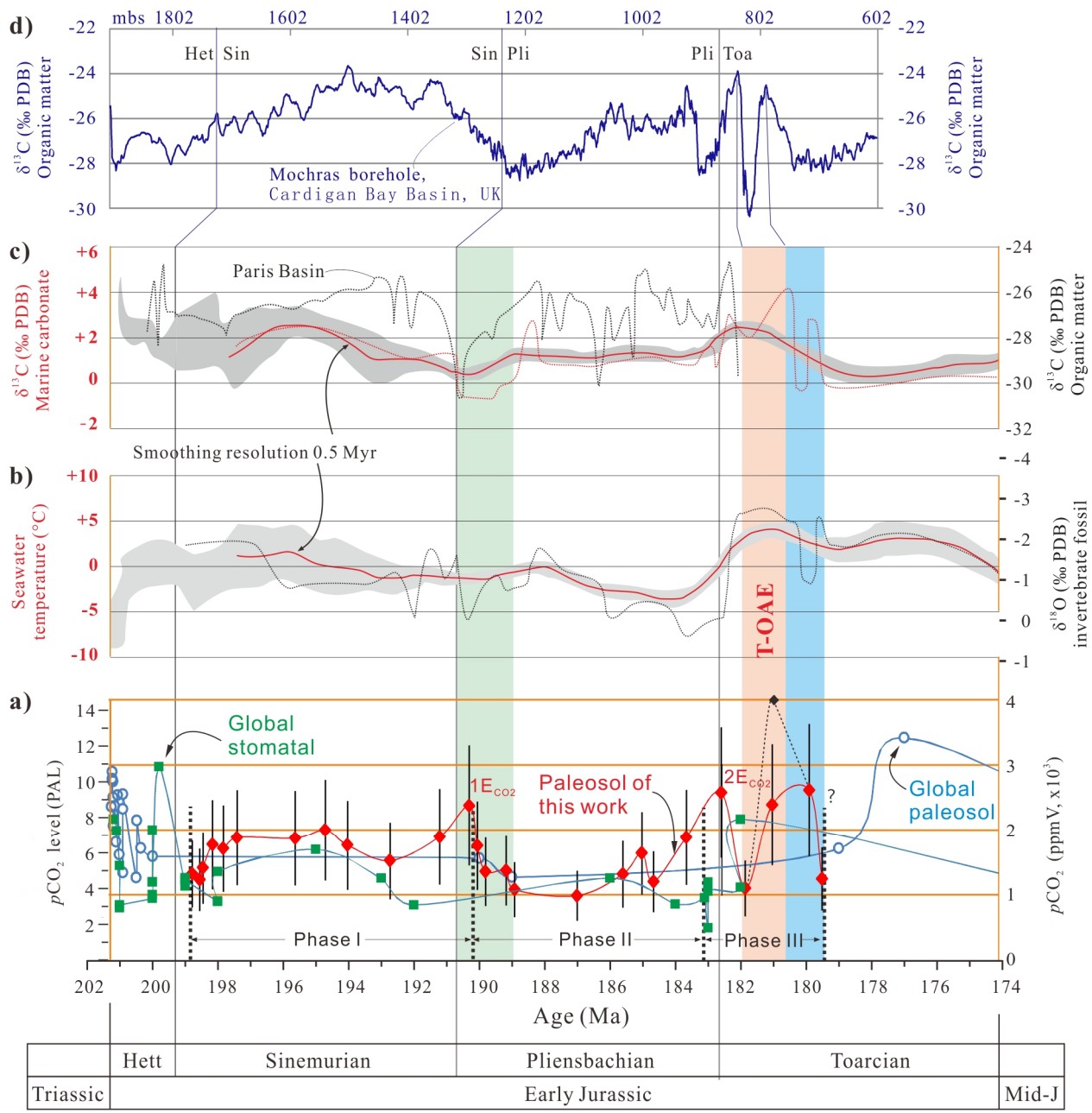

Figure 8  Comparison among the Early Jurassic $pCO_2$, $\delta^{13}C$ of marine carbonates and organic matters, $\delta^{18}O$ of invertebrate fossils, and seawater temperature. Age model is from Cohen et al. (2013). a), $pCO_2$ values of this work and the composite $pCO_2$ by paleosol and stomatal index (supplementary Table S6 and S7). Vertical bars are errors (1σ) of $pCO_2$ (Table S5). Errors are propagated using the Gaussian approach (Breecker and Retallack, 2014). Note: 1) $pCO_2$ = 4027 ppmV (black solid diamond, sample J1z-20-01) if the $\delta^{13}C_r$ = -29.0 ‰ at 181 Ma from Xu et al. (2018) in case of other constant parameters; 2) the early published $pCO_2$ values from both carbon isotope of pedogenic carbonates and stomatal index of fossil plants (data refer to Table S6 and S7) were awfully rough dated with the average age of a lithostratigraphic formation or group, with which the uncertainty can be upto 10 Myr, leading to the difficulty of precise and accurate $pCO_2$ correlation in pace, frequency, and event. b), $\delta^{18}O$ and seawater temperature (black dot line) of marine invertebrate fossils compiled from Rosales et al. (2001, 2004), Jenkyns et al. (2002), Bailey et al. (2003), van de Schootbrugge et al. (2005), Gómez et al. (2008), Metodiev and Koleva-Rekalova (2008), Suan et al. (2008), Korte et al. (2009), Dera et al. (2011), Gómez et al. (2015). c), red dot line $\delta^{13}C$ of marine carbonates and organic matters in western Tethys, composed from Jenkyns and Clayton (1986, 1997), Hesselbo et al. (2000), Dera et al. (2011), Arabas et al., 2017; black dot and solid line $\delta^{13}C$ of organic matters from Paris Basin, France (Peti et al., 2017). Smoothed $\delta^{18}O$ and seawater temperature (red curves) in b) and c) are after Dera et al. (2011). d), $\delta^{13}C$ of organic matters from North Atlantic. Composed from the Mochras borehole, Cardigan Bay Basin, UK (Xu et al., 2018; Storm et al., 2020), seven-point average smoothing against depth (mbs).

890

**Table**

**Table 1    Stratigraphic framework of the Lower Jurassic Ziliujing Fm in Sichuan and adjacent area (GSB), Southwest China**

| Epoch | Age | Formation | W Sichuan (Ya'an) | E Sichuan and Chongqing | S Sichuan and N Guizhou | N Sichuan |
|---|---|---|---|---|---|---|
| Middle Jurassic | Aalenian | Xintiangou Fm | Xintiangou Fm | Xintiangou Fm | Xintiangou Fm | Qianfuyan / Xintiangou Fm |
| Early Jurassic | Toarcian | Ziliujing Fm | Da'anzhai Mem (Bed 20-34) | Da'anzhai Mem | Da'anzhai Mem | Baitianba Fm |
| | Pliensbachian | | Ma'anshan Mem (Bed 9-18) | Ma'anshan Mem | Ma'anshan Mem | |
| | Sinemurian | | Dongyuemiao Mem (Bed 8) | Dongyuemiao Mem | Dongyuemiao Mem | |
| | | | Zhenzhuchong Mem (Bed 1-7) | Zhenzhuchong Mem | Zhenzhuchong Mem | |
| | Hettangian | | Hiatus | Qijiang Mem | Qijiang Mem | ? |
| Late Triassic | Rhaetian | Xujiahe Fm | Xujiahe Fm | Xujiahe Fm | Xujiahe Fm | Xujiahe Fm |

Notes:    Straigraphic classification and correlation were composed from Dong (1984); SBGM (1997), Wang et al. (2010), Wen and Zhao (2010), Xu et al (2017). Re-Os isotope age of the lower Da'anzhai Member is $180.3 \pm 3.2$ Ma in western Sichuan (Xu et al., 2017). Fm, Formation; Mem, Member.

893

**Supplementary data**

**Captions of supplemenatary figures**

**Figure S1    Lithological log of the Lower Jurassic Ziliujing Fm with depositional environment interpretations and sample positions at the Shaping section, Ya'an of Sichuan. Bed number and thickness are partly referred to Wen and Zhao (2010).**

**Figure S2    Lithological log of the Lower Jurassic Ziliujing Fm at the Tanba-Maliuping section, Hechuan of Chongqing with depositional environment interpretations and sample positions. Bed number and thickness are partly referred to Wang et al (2010).**

**Figure S3    Field photographs of the Lower Jurassic Ziliujing Fm lithofacies in the GSB. *a*, Well roundness and sorting gravels in the alluvial fan conglomerate. Basal and lower Baitianba Fm. Puji, Wangcang. Hammer 30 cm long. *b*, Large trough cross-bedding with scours in the point bar and channel sandstones. Upper Baitianba Fm; Puji, Wangcang. *c*, Calcisol developed within strong leaching overbank mudrocks on channelized sandstones. Middle of Bed B2, the Zhenzhuchong Member, Shaping section, Ya'an. *d*, Purple red mudrocks intercalated with thin siltstones in flood plain facies. Bed H7 of the Ma'anshan Member, Tanba section, Hechuan. *e*, Whitish medium-thick micritic dolomites in lacustrine facies. Bed H12 of the Da'anzhai Member, Maliuping section, Hechuan. Hammer 34 cm long. *f*, Greeinsh gray lacustrine muddy dolomites and dolomitic mudrocks associated with brownish / reddish purple mudrocks. Bed B21 of the Da'anzhai Member, Shaping section, Ya'an.**

**Figure S4    Microscopic photos showing lithological microfacies of the Lower Jurassic Ziliujing Fm. *a*, Fine lithic (quartz) sandstone. Lithic-dominant fragments are mudrock. Sample $J_1z$-02-01b, Zhenzhuchong Member, Shaping section, Ya'an. Plain-polarised light. *b*, Laminated muddy dolomite and dolomitic mudrocks. Sample $J_1z$-21S2B, Da'anzhai Member, Shaping**

section, Ya'an. Plain-polarised light. *c*, Fine quartz arenite. Sample 18HC-02b3, Bed H2, Qijiang Member, Tanba section, Hechuan. Cross-polarised light. *d*, Micritic dolomite. Sample 18HC-06b, Bed H12, Da'anzhai Member, Maliuping section, Hechuan. Plain-polarised light. *e*, Coquina. Shell wall of bivalves were micritized. Mud and recrystalline calcites filled inter-shells and intra-shells. Sample 18HC-04b, Base of Bed H12, Da'anzhai Member, Maliuping section, Hechuan. Cross-polarised light. *f*, Relict of coquina. Shell wall of bivalves were parly micritized. Strongly recrystalline calcites replaced the fills and shells. Sample 18HC-05b, Bed H12, Da'anzhai Member, Maliuping section, Hechuan. Cross-polarised light.

**Figure S5**   Field photographs of the Lower Jurassic Ziliujing Fm lithofacies in the GSB. *a*, Lithofacies and stratigraphic sequence. Beds B8 to B10 of the lower Ma'anshan and Dongyuemiao members at Shaping, Ya'an. *b*, Karstified gravels within the limestone. The horizon and location is same as *a*. Pen 15 cm long. *c*, Layered dolomites with Karstified cave gravels. Bed H12 of the Da'anzhai Member at Maliuping, Hechuan. *d*, Karstified cave gravels. The horizon and location is same as *c*. Hammer 34 cm long.

**Figure S6**   Stratigraphic correlation of the Lower Jurassic Baitianba Fm in northern GSB. Locations and sources refer to Figure 1. Plant fossils and stratal thickness in the Shiguansi section, Wanyuan are cited from SBG (1980b).

## Captions of supplemenatary tables

**Table S1**   Occurrence list of the Early Jurassic climate-sensitive sediments in the GSB

**Table S2**   Early Jurassic paleosols in Ya'an of Sichuan and Hechuan of Chongqing, Southwest China

**Table S3**   Carbon-oxygen isotope composition of lacustrine carbonates from the Lower Jurassic Ziliujing Fm (Da'anzhai Mem) in the GSB

**Table S4**   $p$CO$_2$ estimate by carbon isotope of pedogenic carbonates from the Lower Jurassic Ziliujing Fm at Shapingion, Ya'an of Sichuan

**Table S5**   Calculation of Gaussian error propagation for the Early Jurassic $p$CO$_2$ estimate in the Sichuan paleobasin

**Table S6**   Global $p$CO$_2$ data of the Latest Triassic - Early Jurassic by stomatal method

**Table S7**   Global $p$CO$_2$ data of the Latest Triassic - Early Jurassic estimated by carbon isotope of pedogenic carbonates

## Captions of supplemenatary notes

**Note S1, Description and interpretation of sedimentary facies and its evolution**

# Early Jurassic climate and atmospheric CO$_2$ concentration in the Sichuan paleobasin, Southwest China

Xianghui Li[1], Jingyu Wang[1], Troy Rasbury[2], Min Zhou[1], Zhen Wei[1], Chaokai Zhang[1]

[1]State Key Laboratory for Mineral Deposits Research, School of Earth Sciences and Engineering, Nanjing University, Nanjing 210023 China.

[2]Department of Geosciences, Stony Brook University, Stony Brook, NY 11794-2100, USA

*Correpondence to:* Xiangui Li (leeschhui@126.com)

**Abstract:**

Climatic oscillations had been developed through the (Early) Jurassic from marine sedimentary archives, but remain unclear from terrestrial records. ~~Climatic oscillations took place through the (Early) Jurassic from marine sedimentary archives, but were not clear from terrestrial records~~~~Unlike marine archives, terrestrial sediments show more complicated and dynamic environment and climate.~~ This work presents investigation of climate-sensitive sediments and ~~new results of climate sensitive sediment observation and~~ carbon and ~~-~~oxygen isotope analyses of lacustrine and pedogenic carbonates for the Early Jurassic Ziliujing Formation from the grand Sichuan paleobasin (GSB), Southwest China. ~~Lithofacies analysis indicates calcisols were widespread in riverine and flood plain facies. Climate sensitive s~~Sedimentar~~y~~s and ~~carbon-oxygen~~stable isotope proxies ~~with and palynofloral assemblages~~ manifest that an overall secular (semi-) arid climate dominated the GSB during the Early Jurassic~~; and that it became drier through the time~~ except for the Hettangian~~, accompanied by occasional evaporites in the Toarcian~~. This climate pattern is similar ~~with~~ to the arid climate in the Colorado Plateau region, western North America, but distinct from the relatively warm-humid climate in North China and ~~northern Gondwanaland~~ high latitude in Southern Hemisphere. The estimated ~~Early Jurassic~~ atmospheric $CO_2$ concentration ($pCO_2$) from carbon isotopes of pedogenic carbonates shows a range of 980-2610 ppmV (~ 3.5-10 times the pre-industrial value) with a mean of 1660 ppmV. Three phases of $pCO_2$ (the Sinemurian 1500-2000 ppmV, the Pliensbachian 1000-1500 ppmV, and the early Toarcian 1094-2610 ppmV) and two events of rapid falling $pCO_2$ by ~1000-1300 ppmV are observed, illustrating the $pCO_2$ perturbation in the Early Jurassic. The ~~pattern~~ perturbation ~~and associated rapid falling events~~ of $pCO_2$ ~~are~~ is compatible with ~~the excursions of stable isotopes and~~ seawater temperature and carbon cycle from the coeval marine sediments, suggest~~consist~~ing~~nt with~~ a positive feedback of climate to $pCO_2$ ~~in a total tendency and eventful change~~through the Early Jurassic~~; but the~~. However, a lack of synchroneity.

# 1. Introduction

Global paleotemperatures were possibly 5-10 °C higher than present druing the Jurassic period based on climate modelling results (e.g., Rees et al., 1999; Sellwood and Valdes, 2008). However, seawater temperature fluctuated by -5 °C to +5 °C, or even much higher magnitude (e.g., Suan et al., 2008; Littler et al., 2010), based on esitmates from the oxygen isotopes of the belemnite and bivalve fossils (Dera et al., 2011, and references therein). The Early Jurassic epoch was an interval of extreme environmental change. In the Sinemurian–Pliensbachian age, the mean sea surface temperatures of the North Atlantic were in excess of 28°C ($TEX_{86}$), comparable with similar palaeolatitudes during the Cretaceous and Early Cenozoic (Robinson et al., 2017); whereas in the late Pliensbachian age, the northern West Tethys Ocean (e.g., Paris basin, northern Spain basin) was ~12.7°C (e.g., Gómez et al., 2008; Gómez and Goy, 2011; Arabas et al., 2017), leading to a polar icesheet hypothesis (e.g., Sellwood and Valdes, 2008; Suan et al., 2010; Dera et al., 2011; Gómez et al., 2015). At ~183 Ma of the early Toarcian oceanic anoxia event (T-OAE) record of highly enhanced organic carbon burial, the surface seawater temperature was high to ~35°C (e.g., Bailey et al., 2003; Korte et al., 2015), and a high temperature (plateau) even continued in the whole Toarcian (Dera et al., 2011). Examples of seawater temperature transitions between cold and hot show in the climate oscillation through the Early Jurassic. Data from the terrestrial realm also provide important details of environmental and climatic change (e.g., Hesselbo et al., 2000; Suan et al., 2010; Jenkyns, 2010; Philippe et al., 2017), from which the oscillated climate could be observed and revealed too. Terrestrial proxies, such as flora (e.g., Riding et al., 2013; Deng et al., 2017; Philippe et al., 2017), vegetation (Pole, 2009), and geochemistry (e.g., Riding et al., 2013; Kenny, 2015; Tramoy et al., 2016) as well as the $pCO_2$ record (e.g., Retallack, 2001a; Beerling and Royer, 2002; McElwain et al., 2005; Berner, 2006; Steinthorsdottir and Vajda, 2015) provide an emerging

record of the ~~–~~Early Jurassic terrestrial climate and environmental change~~s on continents~~. ~~. Particularly, a negative feedback~~ ~~in the global exogenic carbon cycle, from carbon isotopes of lacustrine organic matter, has been hypothesized to account for~~ ~~the Toarcian oceanic anoxic event (Xu et al., 2017), opening a new avenue to link marine and terrestrial climate in the Early~~ ~~Jurassic.~~ Correspondingly, the proxy application of terrestrial sedimentary archives could play a key role in the global Early Jurassic correlation of the marine and terrestrial climate.

~~Obviously, of the p~~Proxies for ~~, ~~$pCO_2$ ~~is the most probableprovide~~are the important linkage between the marine and terrestrial climatic condition. ~~Up to date, the reconstruction of~~Studies of the terrestrial $pCO_2$ record ~~was ~~have focused ~~at~~on the Triassic-Jurassic boundary (e.g., Tanner et al., 2001; Cleveland et al., 2008; Schaller et al., 2011; Steinthorsdottir and Vajda, 2015) and ~~at ~~the Toarcian oceanic anoxic event (McElwain et al., 2005), where~~and~~ $pCO_2$ estimates range 1000 ppm to 4000 ppmV (e.g., Tanner et al., 2001; Cleveland et al., 2008; Schaller et al., 2011). ~~Inconsistent $pCO_2$ results occur in~~ ~~different proxies, however, and~~ ~~However, f~~Few relatively continuous $pCO_2$~~terrestrial climate records~~ records and coupled terrestrial climate ~~environmental ~~changes have been documented for the Early Jurassic. ~~.~~

There are several large Triassic-Jurassic terrestrial basins in West China~~,~~. ~~providing a great opportunity to recover the coeval~~ ~~terrestrial environment and climate, in which.~~ in which ~~T~~the Sichuan Basin has a relatively complete and continuous continental sedimentary sequence of the Upper Triassic-Paleogene (e.g., SBGM, 1991, 1997; Wang et al., 2010). During the Early Jurassic, ~~it had been laid~~the Sichuan Basin was in a Boreotropical climate zone ~~by~~based on climate-sensitve sediments (Fig. 1a. Boucot et al., 2013), or~~,or and~~ a warm temperate climate is suggested based~~y~~ on clay mineralog~~y~~s and phytogeography (e.g., Dera et al., 2009). ~~Correspondingly, the sedimentary archive could play a key role in the global Early~~ ~~Jurassic correlation of the marine and terrestrial climate.~~ In this work, we present ~~new results of~~a field investigation, including lithofacies and paleosol ~~recognition~~interpretation, and ~~carbon-oxygen~~carbon and oxygen isotope analyses of both lacustrine and pedogenic carbonates~~, and $pCO_2$ estimates~~ in ~~the Early Juassic terrestrial ~~Sichuan ~~paleob~~Basin. ~~These~~New results allow us to~~,~~ reconstruct the paleoclimate and relatively consecutiove $pCO_2$ ~~change~~record through the Early Juassic, ~~for ~~~~and we discuss the relationship of $pCO_2$terrestrial climatic change to~~which we compare to stable isotopes of marine sediments and ~~that of the ~~estimated sea water temperature~~marine counterpart~~.

## 2. Geological setting and stratigraphy

Southwest China, including the provinces of Yunnan, Sichuan, Chongqing, and Guizhou, had been the main part of the upper Yangtze Plate since the Proterozoic, possibly since the Neoarchean. With the amalgamation of the Cathaysia and Yangtze plates, it became the western South China plate or cratonic basin ~~since~~in the Neoproterozoic (Sinian),~~-~~ ~~and marine~~ ~~Neoproterozoic through ~~continued to the late Middle Triassic ~~strata is well preserved~~. ~~With~~By the Indosinian orogeny, new

foreland basins were formed since the Late Triassic (e.g., He and Liao, 1985; Li et al., 2003), ~~which~~ recording the Mesozoic
and Cenozoic evolution of tectonics, environment, and climate in Southwest China.
The Mesozoic Sichuan paleobasin was confined by the Longmenshan thrust belt in the northwest, the Micangshan-Dabashan
arcuate thrust belt in the northeast (Fig. 1b), and the northern hilly topography boundary of the Yunnan-Guizhou plateau in
the south and east. It was mainly developed during the Late Triassic-Jurassic and includes provincial areas of eastern
Sichuan, entire Chongqing, northern Guizhou, western Hubei, and northwestern Hunan. This Triassic-Jurassic Sichuan
foreland basin was much larger than the present Sichuan Basin in the eastern Sichuan province. We estimate the size of
Sichuan paleobasin is roughly 480,000 km$^2$ ~~by the~~based on lithofacies paleogeography (Fig. 1b. Ma et al., 2009; Li and He,
2014), and suggest naming this the grand Sichuan paleobasin (GSB).
The Mesozoic terrestrial sediments accumulated up to ~9 km (Guo et al., 1996) in the GSB; and the Jurassic part can be as
much as 3-3.5 km thick (SBGM, 1991). Two types of Lower Jurassic deposits have been distinguished (Table 1): the
Baitianba Formation (Fm) in the north (~10%) and the Ziliujing Fm (e.g., SBGM, 1991; Wang et al., 2010) in the south
(over 90% of the basin).
The Baitianba Fm was deposited unconformably on the Upper Triassic Xujiahe Fm and is overlain conformably by the
Middle Jurassic Xintiangou Fm / Qianfuyan Fm (Table 1). It is mainly composed of grayish shales and sandstones with coal
layers and massive conglomerates. Abundant plant fossils, sporopollens, conchostracans, bivalves, and gastropods indicate it
is of the Early Jurassic (SBGM, 1991, 1997). Sporopollen assemblages of the Hettangian-Sinemurian age were found in the
lower part (Zhang and Meng, 1987) and the Pliensbachian-Toarcian assemblages were reported in the upper part (Wang et
al., 2010).
The Ziliujing Fm is composed of variegated and reddish mudrocks (some shales) intercalated with sandstones, siltstones, and
bioclastic limestones as well as dolomitic marlstones / limy dolomites, conformably or unconformably overlying the Xujiahe
Fm or Luqiao Fm and conformably underlying the Xintiangou Fm (SBGM, 1997. Table 1). It has been dated as the Early
Jurassic by fossil assemblages of ~~dinosaurs,~~ bivalves, ostracods, conchostracans, and plants~~, within which the~~. ~~dinosaur~~
Dinosaur fauna can be well correlated to the Lufeng Fauna in central Yunnan (e.g., Dong, 1984; SBGM, 1991, 1997; Peng,
2009). This formation is subdivided ~~as~~into five parts in an ascending order: the Qijiang, Zhenzhuchong, Dongyuemiao,
Ma'anshan, and Da'anzhai members (SBGM, 1997. Table 1). Of ~~them~~these, the former two are sometimes combined as the
Zhenzhuchong Fm (e.g., SBGM, 1991; Wang et al., 2010).
The Da'anzhai Member is characterized by dark gray to black shales and bioclastic limestones with a southward increase of
reddish mudrocks (SBGM, 1991, 1997; Wang et al., 2010), ~~which has been mainly~~and is regarded the sediment in a grand
Sichuan paleolake (e.g., Ma et al., 2009; Li and He, 2014). Ostracod ~~assemblagse~~assembleges indicate it is the late Early
Jurassic (e.g., Wei, 1982; Wang et al., 2010). A Re–Os isochron age of 180.3 ± 3.2 Ma associated~~combined~~ with ~~the~~an

organic carbon isotope excursion indicates that the lower Da'anzhai Member corresponds to the ~~Toarcian Oceanic Anoxic event~~ (T-OAE (~~.~~ Xu et al., 2017)~~, consistent with the assigned Toarcian age~~.

The Ma'anshan Member is comprised of violet-red mudrocks with a few greyish, greenish thin-bedded fine sandstones and siltstones, in which floral fossils are common (Li and Meng, 2003). The Dongyuemiao Member consists of greenish and reddish mudrocks and siltstones with greyish bioclastic limestone and marlstone, of which abundant bivalve and plant fossils were reported from eastern Sichuan and Chongqing (Li and Meng, 2003; Meng et al., 2003; Wang et al., 2010). The Zhenzhuchong Member is dominated by violet red mudrocks/shales intercalated with thin-~~-~~bedded ~~ned~~ sandstones and / or siltstones and numerous plant fossils of the Early Jurassic affinity (e.g., Duan and Chen, 1982; Ye et al., 1986). Taken together, fossil associations suggest that the three members were deposited in the middle-late Early Jurassic. The age limitation of the overlying Da'anzhai Member and the correlation to the Lufeng dinosaur fauna places these members in the Sinemurian – Pliensbachian, and the Zhengzhuchong and Dongyuemiao Fms are ~~temporally~~ suggested ~~the~~ to be the Sinemurian ~~age~~ (Table 1).

The Qijiang Member is composed of quartz arenite interbedded/intercalated with dark shales. Coal seams ~~can be~~are often seen in the middle of the Qijiang Member. This member mainly occurs in the central part of the GSB. It is likely the earliest Jurassic, possibly Hettangian age, but plant fossils cannot precisely indicate the age (Wang et al., 2010).

## 3. Materials and methods

We have measured sections and made detatiled ~~O~~observations and descriptions of sedimentary characteristics for ~~sedimentary~~ lithofacies analysis ~~were executed on~~at six outcrop sections (Locations A1 to A4, A6 and A7, Fig. 1). Published descriptions for other sections (Locations A5, A8, and A9, Fig. 1) ~~is~~ are integrated into our observations. Details of microscopic examination of sedimentary rocks and analysis of sedimentary facies ~~analysis which are the~~ underpinning the ~~of~~ climate analsysis are attached as the supplementary data Note S1. Below ~~are chiefly introduced materials and~~we ~~discuss~~state ~~methods of~~ climate-sensitive sediment observation, ~~carbon-oxygen~~carbon and oxygen isotope analyses, and estimate of $p$CO$_2$.

### 3.1. Observation of climate-sensitive sediments

Climate-sensitive sediments are mainly ~~the~~ dolomites, ~~gupsum~~gypsum, and paleosols, which are used to analyze the climate in this work (Table S1).

Dolomites and gy~~u~~psum are relatively easy to recognize in both field and under microscope. We distinguish dolomites from limetstones following Tucker (20~~03~~11) and Flügel (2004). As Flügel (2004) stated, field distinctions of limestone and dolomite can also be made although detailed differentiation of carbonate rocks is best performed in the laboratory. ~~The basic~~

~~method that we use to examine dolomites is: limestone will fizz strongly and dolomite will show little or no reaction when~~
~~add dilute 10 % hydrochloric acid on carbonate (Flügel, 2004); and limestone will stain pink to mauve but dolomite will be~~
~~unstained (e.g., Tucker, 2003; Flügel, 2004) when Alizarin red S in weak HCl is added on freash outcrop or coverslip free~~
~~thin section. Gypsum is recognizable by properties of low Mohs hardness (2) and transparence to translucence.~~ In field, we
~~also~~ recognize gypsum by particular structures such as chic~~k~~hen-wire cage, gypsum pseudomorph, and cluster of (0.5-1 cm)
pore.
There are multiple classifications of paleosols (e.g., Wright, 1992; Mack et al., 1993; Retallack, 2001b; Imbellone, 2011),
mostly based on the US Soil Taxonomy. We recognized paleosols in the field based on color, structures, horizonation, root
traces, and textures, and followed the general classification paleosols by Mack et al. (1993) and Retallack (2001b). In this
paper, paleosols ~~were~~ are described following the procedures of the Soil Survey Manual and classified according to Soil
Survey Staff (1998).
Within the measured and observed sections, paleosol profiles were mainly identified from the two main locations/sections
A4 and A6 (Figs. S1 and S2, and Table ~~S1~~S2). Horizonation, BK horizon thickness, boundar~~ies~~~~y condition~~, structures, trace
fossils, rootlets, carbonate accumulations (calcretes), etc. were ~~observed and described~~recorded (Table ~~S1~~S2). Paleosols
interpreted in other cited sections (Fig. 1) rely on the ~~diescription~~ description of lithology, strcucture, and calcrete in the
original references.
Based upon a modification of the Retallack (1998) categorization of paleosol maturity, the relative paleosol development
(maturity) was assigned.
**3.2. Analyses of carbon and ~~oxygen~~ oxygen isotopes**
Ten lacustrine carbonate samples were ~~collected to~~ analysed for ~~carbon oxygen~~carbon and oxygen isotopes from the
Da'anzhai Member ~~of the Ziliujing Fm~~ at the Shaping section, Ya'an (Location A4. Fig. S1 and Table S3). ~~Twenty six~~26
pedogenic carbonate samples were ~~selected to measure~~analyzed for ~~carbon oxygen~~carbon and oxygen isotopes from
~~thirty-one~~32 paleosol ~~horizon~~s of the Ziliujing Fm at the same section (Fig. S1 and Table S4). Two or three microdrilling
powder samples (columns 7 and 8 in Table S4) were taken from the same individual calcrete for stable isotope analysis, and
then a mean value for each calcrete sample was calculated (columns 9 and 10 in Table S4).
At the field scale, calcretes are ginger-like and sporadically spaced within the soil horizon. We observed no linear and planar
calcretes that would indicate precipitation at or below the water table. Before drilling, ~~the diagenetic fabrics of~~ thin-sections
~~of the samples~~ were petrographically studied ~~under a microscope~~petrographically. ~~Each sample was cut and prepared as thin~~
~~sections for diagenetic diagnosis,~~ using polorized light microscopy and cathodoluminescence ~~(CL) images~~ imaging.
Micritic calcite is predominant in both lacustrine and pedogenic carbonate samples, with no evidence for carbonate detritus
in calcretes (Fig. 2a and 2b). The micritic calcites used for stable isotope analyses are chiefly null- to non-luminescent, with
<10% light orange and brownish luminescence, indicating genesis primarily in the vadose zone. While luminescent calcretes
indicate a high possibility of hydrological influence (e.g., Mintz, et al., 2016), we sampled to avoid this. Based on
petrography and CL imaging together with the field observations, the dense micritic zones sampled for the stable isotope
composition should give pristine $\delta^{13}C$ values that can be used to estimate $pCO_2$.
(Fig. 2) were used to examine if the calcites were evenly precipitated. Only the areas that were a uniform (often orange)
luminescene (Fig. 2) were microsampled for isotope analyses. SDiagenetic sfromCracks, veins, and vug spaces in concretion
samples were found to be filled by multidirectional growth of spar crystals. These crack spar fills were was avoided when
microsampling as they were interpreted as recrystallization and replacement diagenetic phases. Microsampling of lacustrine
and pedogenic carbonate samples focused on avoiding spar and sampling only micrites, and avoiding spar.diagenetic spar
from cracks, veins, and vug spaces. Powder samples were obtained by dentist drilling machineusing a dental drill (aiguille
diameter φ=1-2 mm).
Isotopic analyses were conducted on 0.3 ~ 0.5 mg powder samples. Powder samples were dried in an oven at 60˚C for 10
hours before being moved to the instrument. Carbon dioxide for isotopic analysis was released using orthophosphoric acid at
70˚C and analysed on-line in a DELTA-Plus xp (CF-IRMS) mass spectrometer at the State Key Laboratory for Mineral
Deposits Research, Nanjing University. The precision of the measurements was regularly checked with a Chinese national
carbonate standard (GBW04405) and the international standard (NBS19) and the standard deviation of $\delta^{13}C$ was ±0.1‰ over
the period of analysis. Calibration to the international PeeDee Belemnite (PDB) scale was performed using NBS19 and
NBS18 standards.

## 3.3. Calculation of atmospheric $CO_2$ concentration

There are multiple methods to reconstruct the concentration of atmospheric carbon dioxide (, i.e., $pCO_2$,) in deep time. It can
be determined from the $\delta^{13}C$ values of pedogenic carbonate using a paleobarometer model (Cerling, 1999), and the
reconstruction of $pCO_2$ has been applied in the climate case study of the Mesozoic climate time (e.g., Ekart et al., 1999;
Nordt et al., 2003; Myers et al., 2012; Li et al., 2014; Zhang et al., 2018).
The Cerling (1991, 1999) equation was used to calculate the $pCO_2$ using the carbon isotope of pedogenic carbonates as
below:

$$C_a = S_{(z)}(\delta^{13}C_s - 1.0044\delta^{13}C_r - 4.4)/(\delta^{13}C_a - \delta^{13}C_s)$$

where $C_a$ is $pCO_2$; $\delta^{13}C_s$, $\delta^{13}C_r$, $\delta^{13}C_a$ are the isotopic compositions (‰) of soil $CO_2$, soil-respired $CO_2$, and atmospheric $CO_2$,
respectively; and $S_{(z)}$ is the $CO_2$ contributed by soil respiration (ppmV). Details of parameter usage and selection for the
$pCO_2$ calculation are in the supplememtary data Note S2.
−$\delta^{13}C_s$ is often calibrated by fractionation factor −8.98‰ with the formula -8.98‰+$\delta^{13}C_c$ (Ekart et al., 1999), with which
$\delta^{13}C_c$ is the measured result of pedogenic calcrete. OrAlternatively, $\delta^{13}C_s$ can be replaced by $\delta^{13}C_{sc}$, which is calibrated by
carbon isotope ratio of pedogenic carbonate at 25°C based on latitude–temperature correlations (Besse and Courtillot, 1988;
Ekart et al., 1999) following the equation $\delta^{13}C_{sc} = (\delta^{13}C_c+1000)/((11.98-0.12*T)/1000+1) -1000$ (Romanek et al., 1992). We
used both $\delta^{13}C_s$ and $\delta^{13}C_{sc}$ to calculate the $p$CO$_2$, ~~respectively~~ (Table S4).
$\delta^{13}C_r$ represents carbon isotope ratio of average bulk C3 vascular tissue (Arens et al., 2000), reflecting atmospheric $\delta^{13}C$
(Jahren et al., 2008). ~~So, the~~The $\delta^{13}C_{om}$ of organic matter within paleosols based on the range of modern C3 ecosystem
fractionations (Buchmann, et al., 1998; Ekart et al., 1999), is commonly ~~the representative of~~used for $\delta^{13}C_r$ ~~in the above~~
~~model equation~~. However, the $\delta^{13}C_r$ could be ~~not almost applied in the measurement of~~could be compromised in ~~the~~fossil
soils due to oxidation and metabolism of organic matter after burial (Nadelhofer and Fry, 1988). In this paper, we use~~d~~ the
$\delta^{13}C_{om}$ from the Paris Basin (Bougeault et al., 2017; Peti et al., 2017) for the Sinemurian-Pliensbachian $\delta^{13}C_r$ and from
Cardigan Bay, UK (Xu et al., 2018) for the Toarcian ~~one which was not oxidized, metabolized and well-dated~~.
$\delta^{13}C_a$, the carbon isotopic composition of the atmosphere, was about -8‰ in the 1980s, being depleted relative to the
pre-industrial atmosphere which was around -6.5‰ (Friedli et al., 1986). The average value of –6.5‰ has been chosen as the
$\delta^{13}C_a$ for acquiring $\delta^{13}C_r$ and $S_{(z)}$ (e.g., Ekart et al., 1999; Robinson et al., 2002), and the $\delta^{13}C_a$ was generally calibrated as
$\delta^{13}C_{ac}$ from $\delta^{13}C_r$ using the formula $(\delta^{13}C_r+18.67)/1.1$ (Arens et al., 2000). Herein we used both calibrations to calculate the
$\delta^{13}C_a$ (Table S4).
$S_{(z)}$ is the largest source of uncertainty in $p$CO$_2$ estimates (Breecker, 2013) and the uncertainty arises primarily from their
sensitivity to soil-respired CO$_2$ ($S_{(z)}$ (Montañez, 2013). It is a function of depth and effectively constant below 50 cm (e.g.,
Cerling, 1991). ~~In earlier publications, $S_{(z)}$=5000 ppmV was often adopted. Large discrepancy of $S_{(z)}$ was interpreted and~~
$S_{(z)}$=2500 ppmV is suggested for the sub-humid temperate and tropical climates (Breecker et al., 2010), 2500-5000 ppmV for
higher moisture and productivity soil (Montañez, 2013), 2000 ppmV for semi-arid areas (Breecker et al., 2009), 1500-2000
ppmV for aridisols and alfisols (calcisol-argillisol) and 2000±1000 for paleo-vertisol (Montañez, 2013), and 1000 ppmV in
desert areas (Breecker et al., 2010) or 400 ± 200 ppmV for immature soil (Montañez, 2013). In this context, we chose the
$S_{(z)}$=2000 ppmV for calculating $p$CO$_2$ at 25°C as the calcisols are reddish-brownish aridisols, and we also compared the
results with that by $S_{(z)}$=2500 ppmV (Table S4). Additionally, we took samples at the middle and lower Bk horizon (often >
~20-30 cm to the BK top). That means the depth of calcrete samples in the examined palaeosols was generally deeper than
50 cm below the paleosol surface, meeting the requirement for a constant value of $S_{(z)}$.

**4. Results**
Based on the investigation of cross-sections (locations A1-A4, and A6-A7. Fig. 1), we have classified six sedimentary facies
units in the Ziliujing Fm. They are alluvial fan, fluvial river, flood plain, lake, lake-delta, and swamp facies. Details of
description and interpretation are in the supplementary data Note S1. Below are results of climate-sensitive sediment
observation, stable isotope analyses, and $p$CO$_2$ calculation.

## 4.1. Climate-sensitive sediments

Field observation combined with published calcrete materials shows that paleosols widely occur in the Lower Jurassic
Ziliujing Fm of the GSB (Figs. 1, 3, and 4). A total of 32 paleosols were observed and described at the Shaping section,
Ya'an, and five paleosols were found at the Tanba section, Hechuan (Table ~~S1~~S2).
Most of paleosols are reddish (GSA Munsell Rock-Color 5R 2/2, 5R 3/4, 5R 4/2) and brownish (10R 3/4, 10R 5/4) (Fig. 3
and Table ~~S1~~S2). Peds of paleosols are mainly angular and subangular, and a few are prismatic and platy. Slickensides are
common. Mottles (Fig. 3a), rootlets /rhizoliths (Fig. 3c), and burrows sometimes occur with strong leaching structures (Fig.
3a). Occasionally mudcracks are associated with the aforementioned structures (Fig. 3d).
All paleosols are calcic with more or less calcretes in Bk horizons. The thickness of Bk horizons mainly changes ~~is~~
~~mainly~~from −30-~~50~~ cm and ~~50~~-100 cm, and partly up to ~~100~~-170 cm (Table S~~1~~2). Calcretes are generally ginger-like,
ellipsoid, subglobular, and irregular in shape (Fig. 3b and 3e) and nodules are 1-3 cm even up to 8-15 cm (paleosols
J1z-10-01 and J1z-12-01) in size (Fig. 3e). Calcrete is often less than 0.5-1% in an individual paleosol ~~horizon~~, but a few can
be up to 3-5% (paleosol J1z-3-01. Fig. 3b) even 10% (paleosols J1z-5-02 and 18HC-10).
~~Based on the description of the paleosols described above, a~~All above paleosols are defined as relatively mature calcisols
(Mack et al., 1993), a kind of aridisol (Soil Survey Staff, 1998; Retallack, 2001b). The original lithofacies were chiefly
argillaceous and silty (split-fan) overbank, interchannel, and flood plain deposits (Figs. S1 and S2). Some formed
land~~share~~ward ~~of~~ the paleo-lakeshore.
Dolomites were found at seven loactions in central and southern GSB (Figs. 1, 4, and Table ~~S2~~S1)~~), which are to some~~
~~degree an indicative of arid/evapoatre climate~~. The dolomites chiefly occur in the Toracian Da'anzhai Member and a few in
the Sinemurian-Plienbachian Dongyuemiao and Ma'anshan members (Fig. 4). They are often massive whitish (Figs. 3f and
S3e) and micritic (Figs. S4b and S4d), likely indicating an authigenic ~~syndepositional~~ origin.
Gypsum is ~~only~~ recorded in two loactions (Figs. 1, 4, and Table ~~S2~~S1). One is located at Zigong (Location A5. SBG, 1980a).
The other lies at Hechuan (Location A6), which can be idientifed by chicken-wire cage structure and is associated with
micriditic dolomites (Fig. 3f).

## 4.2. ~~Carbon-oxygen~~Carbon and oxygen isotope values

δ$^{13}$C values of lacustrine carbonate samples range from -2.02‰ to -4.07‰ and δ$^{18}$O values ~~dorange~~range from -9.91‰ to
-12.28‰ (Table S3 and Fig. 5). A~~n~~ ~~distinct~~increasing trend of both carbon and oxygen isotope ratios ~~can be detected~~is
observed from lower to upper horizons across a 4~~0~~5 m stratal interval of the lower Da'anzhai Member (Fig. 6).

Pedogenic carbonate samples have $\delta^{13}C$ values from -3.52‰ to -8.10‰, which fall in the typical stable isotope range for pedogenic carbonates. Values of -6‰ to -8.0‰ characterize the sequence of the Zhenzhuchong Member and main Ma'anshan Member, with an abrupt increase to -5.5‰ to -3.5‰ at the top of Ma'anshan Member (samples J1z-16-01 and J1z-18-01. Fig. 6). $\delta^{18}O$ values are mainly from -11.3‰ to -13.10‰ in the interval of the Zhenzhuchong Member and Ma'anshan Member. $\delta^{18}O$ follows $\delta^{13}C$ with a sudden increase to -5.5‰ at the top of the Ma'anshan Member (Fig. 6). Large and frequent variations of both carbon and oxygen isotope ratios can be observed in the lower Da'anzhai Member (Fig. 6 and Table S4).

**4.3. CO$_2$ concentrations**

$p$CO$_2$ values based on paleobarometer modelling of paleosol calcite (Cerling, 1999) of the Early Jurassic paleosols vary in the Early Jurassic when different depending on the parameters are selectedused for the calculation.

If S$_{(z)}$=2500 ppmV and $\delta^{13}C_a$=-6.5‰ (constant pre-industrial atmosphere), $p$CO$_2$ values range between ~1140 ppmV and ~3460 ppmV with a mean of 1870 ppmV (column 15 in Table S4); and when S$_{(z)}$=2500 ppmV and $\delta^{13}C_a$=($\delta^{13}C_r$ +18.67)/1.1, $p$CO$_2$ values change between ~1230 ppmV and ~3260 ppmV with a mean of 2070 ppmV (column 16 in Table S4).

IfWhen S$_{(z)}$=2000 ppmV and $\delta^{13}C_s$=-8.98+$\delta^{13}C_c$ are used, $p$CO$_2$ values are ~ 940-2530 ppmV with the mean 1600 ppmV (column 17 in Table S4); and if S$_{(z)}$=2000 ppmV and $\delta^{13}C_s$ = ($\delta^{13}C_c$+1000) / ((11.98-0.12*25) / 1000+1) -1000 are adopted, $p$CO$_2$ values become ~980 ppmV to ~2610 ppmV with the mean 1660 ppmV (column 18 in Table S4). Details of the different parameters and $p$CO$_2$ results can be seen in Table S4.

Results further show that $p$CO$_2$ values at S$_{(z)}$=2500 ppmV are larger than at S$_{(z)}$=2000 ppmV., and the discrepancy of thedifference between the The highest difference calculated $p$CO$_2$ is ~ 1000 (3640-2610) ppmV, but while that the difference of the lowest value is ~300 (1230-930) ppmV and that of the mean value is ~ 370 (2070-1600) ppmV. In addition, when S$_{(z)}$ is the same, the $p$CO$_2$ values are close even if other parameters are different (comp. between columns 15 and 16, 17 and 18 in Table S4, and Fig. 6).

HoweverWhichatever parameters used, the trend of $p$CO$_2$ over the epoch is quite similar using different values of S$_{(z)}$ and other parameters (Fig. 6). We chose S$_{(z)}$=2000 ppmV (column 18 in Table S4) to illustrate the nature of the Early Jurassic $p$CO$_2$ estimated from calcisols in the GSB.

$p$CO$_2$ values mostly range between 980 ppmV and 2610 ppmV, and the mean 1660 ppmV is ~6 times the pre-indutrial 275 ppmV. Most of the $p$CO$_2$ values are 1000-2000 ppmV with the mean 1580 ppmV in the Zhenzhuchong and Ma'anshan members, ~3.5-7.5 times the pre-industrial $p$CO$_2$ value.

It is noted that the errors of $p$CO$_2$ range from 384 ppmV to 1017 ppmV with a mean 647 ppmV (Table S5), leading to a large uncertainty of the mean ~39%. The largest source of the uncertainty is the standard error (766 ppmV) of modern soil carbonate (Breecker and Retallack, 2014). The $p$CO$_2$ uncertainty decreases by ~ 20% if half (383 ppmv) of the standard error of soil

carbonate is selected, and decreases to ~12% if 1/4 (~191 ppmV) standard error is used. The second largest source of error in
the $p$CO$_2$ is the S$_{(z)}$ estimate. The uncertainty of $p$CO$_2$ becomes much smaller when the S$_{(z)}$ is larger, e.g., it will fall from ~39%
to ~17% if S$_{(z)}$=5000 ppmV instead of 2000 ppmV. Other parameters such as temperature, $\delta^{13}C_r$, $\delta^{13}C_a$, $\delta^{13}C_s$, contribute very
little to the calculated $p$CO$_2$ uncertainty.

## 5. Discussion

~~Similar with the entire Jurassic period, the Early Jurassic epoch had experienced~~The Jurassic marine record shows climatic
and environmental oscillations (e.g., van de Schootbrugge et al., 2005; Dera et al., 2011; Gómez et al., 2015; Arabas et al.,
2017), ~~expressing the~~including sea water temperature fluctuation and carbon cycle reorganization recorded in~~s~~ both
carbonate and organic matter~~s~~. The climate~~ic~~ change~~s~~ and events ~~of~~ recorded in the the marine realm have been mainly
attributed to ~~the~~ Karoo-Ferrar volcanism (e.g., Hesselbo et al., 2000; Caruthers et a., 2013), ~~and alternatively~~ sea-level
change (e.g., Hesselbo and Jenkyns, 1998; Hallam and Wignall, 1999), orbital forcing (e.g., Kemp et al., 2005; Huang and
Hesselbo, 2014, Storm et al., 2020), and / or the opening of the Hispanic corridor (e.g., van de Schootbrugge et al., 2005;
Arias, 2009). ~~The volcanic forcing hypothesis is that the volcanism had triggered the degassing, increasing~~E~~E~~ruption of the
Karoo-Ferrar and Central Atlantic ~~basalts~~ mgama is thought to have released large amounts of CO$_2$ ~~concentration in~~into the
atmosphere in a short amount of time, ~~and~~ resulting in rising temperatures of both marine and continental realms. The
nearly continuous record of Jurassic strata in the GSB provides an excellent test of this hypothesis in the terrestrial realm. ~~To~~
~~test this hypothesis, we analyzed~~We compare the climate and $p$CO$_2$ record ~~of~~from the GSB ~~and discussed the $p$CO$_2$ change~~
~~with~~in relationship to the marine temperature records.

### 5.1. Paleoclimate variation

~~Results show that the depositional environment and paleoclimate in the Early Jurassic were distinctly different from those in~~
~~the Late Triassic in Southwest China. As a whole, the climate became dry and $p$CO$_2$ varied in three phases through the Early~~
~~Jurassic.~~
~~Sedimentary facies analysis indicates two lithofacies cycles were developed and calcisols were largely spread in the Lower~~
~~Jurassic Ziliujing Fm in the GSB, Southwest China. The first cycle is the riverine and flood plain lithofacies of the Qijiang~~
~~Membe and Zhenzhuchong Member~~ which is ~~succeeded by the lacustrine facies of the Dongyuemiao Member, and the~~
~~second is the flood plain and river facies with swamp lithofacies of the Ma'anshan Member followed by the lacustrine facies~~
~~of the Da'anzhai Member. We interpret the two packages to reflect two major lake stages (for details refer to supplementary~~
~~data Note S1).~~
~~Results of climate sensitive sediment analyses show that the depositional environment and climate in the Early Jurassic were~~

distinctly different from those in the Late Triassic in Southwest China. With the change of depositional environments, paleoclimate and pCO2 changed, as reflected by climate-sensitive facies and stable isotope analyses.

5.1. Paleoclimate variation

During the Late Triassic, Southwest China was warm-hot and humid in and occupied a tropical and / or subtropical zone, as demonstrted by palynoflora, coals, and perennial riverine and lacustrine lithofacies in the Xujiahe Fm (e.g., Huang, 1995; Xu et al., 2015; Li et al., 2016; Yang et al., 2019), ). However, the whereas and a distinct transfer of climate took palce in theCclimate became dry through the Early Jurassic manifested by climate-senstive sediments and stable isotopes of the Ziliujing Fm in GSB albeit there are two lithofacies packages reflecting two major lake stages (for details refer to supplementary data Note S1) in the GSB. As a whole, the climate became dry through the Early Jurassic. Below are illustrations of climate by age.

### 1)5.1.1 The Hettangian Age

By In the Hettangian time (the Qijiang Member), the climate was a warm-humid climate followedlike the Late Triassic in the GSB. The limited sedimentary recordsQijiang Member is comprised of are mainly mature quartz sandstonarenites and siltstones with coals (Fig. 7) and as well as siderite concretions (Fig. 7), indicating a stable tectonic setting and warm-humid climate in the eastern and southern GSB. In the northern margin, the cClimate was similar across the whole region, because multiple coal layers occur in the lower Baitianba Fm. and the hosted aThe alluvial fan system of the lower Baitianba Fm. (Figs. 7 and S6) is characterized by moderate-good roundness and sorting of gravels with sandy fillings matrix (Fig. S3a. e.g., Liu et al., 2016; Qian et al., 2016; and this work). In the Newark basin of eastern North America, climate-sensitive sediments such as nodules of carbonate and gypsum (pseudomorph) as well as mudcrack in mudflat facies indicate an arid climate in the fifth cycle of the Hettangian (>199 Ma). Kent et al., 2017) Passaic Fm (Kent et al., 2017) (Smoot and Olsen, 1994). More widespread, the eolian Navajo Sandstone, dated as Hettangian-Sinemurian (200-195 Ma. Parrish et al., 2019), indicates an arid climate in Colorado Plateau (Fig. 1a. Boucot et al., 2013). Obvieously, the arid climate in western America was different from that in the GSB at the time.

### 5.1.22) The Sinemurian Age

The early Sinemurian Zhenzhuchong Member is a combination of riverine and flood plain facies withand lacustrine facies (supplementary Note S1). , in which theThe lithology is dominated by violet-red mudrocks with few thin greyish, greenish fine sandstones and siltstones. The reddish color of rocks may indicate a change of climate even if there is a little bit difference in the color appearance of reddish color sediments in the western and central basin. The differenceDifferences in the red color appearance That show thatis, the reddish color the reddish rocksstarted in the middle member in the central basin (Location A6. Fig. S2) but almost developed through the whole member in the western margin basin (Location A4.

Fig. 6)~~, but it started in the middle member in the central basin (Location A6. Fig. S2)~~.
With~~in~~ ~~the~~ red~~-color~~dish mudrocks of ~~, a kind of climate-sensitive pedogenesis is recognized from~~ the flood plain facies
~~demonstrates an arid climate,~~. Mmultiple calcisols ~~horizons~~ were observed at the Shaping section, Ya'an (Location A4. Figs.
1, 4, and 7), ~~within which~~here including a strongly leach~~ed~~ing calcisol horizon ~~can be found~~ (Fig. S3c). We~~Calcisols were~~ ~~also~~
also interpret~~ed~~ the reddish muddy sediments with ~~the description of~~ abundant calcretes as the calcisol at sections of Dafang
(Location A8. Zhang et al., 2016), Tianzhu (Location A9. Li and Chen, 2010), and Weiyuan (Location A10. SBG, 1980a)~~,~~
~~respectively.~~ The Ccalcisols indicate ~~a transition from the humid climate of the Late-Triassic and Hettangian to that a~~ (semi-)
arid climate ~~at least began to replace the previous humid climate in western and southern margins of the basin~~in the
Sinemurian ~~(Figs.1, 4, and 7 and Table S2)~~.
This climate change, ~~indicated~~ interpreted from reddish mudrocks and paleosols, is consistent with the ~~climatic signal from~~
floral fossils (e.g., Huang, 2001; Wang et al., 2010)~~,~~ ~~that,~~ suggest~~ing~~ting ~~the~~ ~~a decrease in~~ decreasing humidity and ~~an~~
~~increase in~~increasing temperature ~~across the interval, compared to that in~~from the Late Triassic epoch and the Hettangian
~~age~~into the Sinenmurian~~Qijiang Member and Late Triassic Xujiahe Fm~~ into the Sinenmurian age in the southern GSB.
However, ~~the climate was not distinct in humidity and temperatue~~ in the northern GSB ~~without~~there are few proxies ~~for~~ of
~~sediments and flora~~climate change, ~~even though~~ and alluvial fan and lacustrine delta facies ~~are~~ common in the middle~~of~~
~~the~~ Baitianba Fm (Fig. S6. e.g., Qian et al., 2016) do not give us information on climte.
~~No climate sensitive sediments are documented in the late Sinemurian Dongyuemiao Member from previous studies, in~~
~~which it is characterized by lacustrine limestones. However,~~The late Sinemurian Dongyuemiao Member also has ~~similar to~~
~~the Zhenzhuchong Member,~~ reddish mudrocks and calcisols, ~~with~~ similar to the Zhenzhuchong Member.~~,~~ Pedogenic
c~~newly interpreted calcisols indicate drier climate (Figs. 4 and 7 and Table S2).~~ Calcretes ~~within reddish mudrocks~~ were
reported at Dafang (Location A8. Zhang et al., 2016), Tianzhu (Location A9. Li and Chen, 2010), and Yunyang (Location
A15. Meng et al., 2005) and in the central and southern GSB (Figs 4 and 7 and Table S2), ~~.~~ ~~newly interpreted calcisols~~
~~indicate~~displayingindicating ~~a drier~~ continued arid climate ~~conditions~~(Figs. 4 and 7 and Table S2) at the time.
~~nThe probable calcisols indicate the (semi-) arid climate may have interrupted the long-term warm and (semi-) humid~~
~~climate interpreted based on flora in the Early Jurassic (e.g., Meng et al., 1997; Li and Meng, 2003). This interpretation of~~
~~(semi-) arid.~~ The interpreted Sinemurian (semi-) arid climate from reddish mudrocks and calcisols ~~and it  punctuation~~ is
also supported by the flora~~l changes~~ (~~Meng et al., 1997;~~ Li and Meng, 2003) ~~and as well as~~and the mudrock geochemistry ~~of~~
~~mudrocks~~ (Guo et al., 2017).
Few records of coeval terrestrial climate are known from other continents or regions in the literature. ~~A report occurs in~~
~~eastern England, where the co-occurrence of the acmes of thermophilic pollens Classopollis classoides and Liasidium~~
~~variabile indicates the warm humid climate in the late Sinemurian (Riding et al., 2013).~~ TThe Whitmore Point Member of
the Moenave Fm deposited in dryland lakes (Tanner and Lucas, 2008) and the upper part of eolian Navajo Sandstone
(Blakey et al., 1988) could represent the coevally similar climate in Colorado Plateau although relatively cool (~9 to 18 ºC)
continental climate was inferred from oxygen and hydrogen isotope composition of chert precipitated in interdune,
freshwater lakes in the Navajo Sandstone (Kenny, 2015). ~~.~~ With a difference, in eastern England, the co-occurrence of the
acmes of thermophilic pollens *Classopollis classoides* and *Liasidium variabile* indicates the warm-humid climate in the late
Sinemurian (Riding et al., 2013).

**5.1.~~3)~~ The Pliensbachian Age**

~~The Ma'anshan Member is likely the Pliensbachian, though age information is lacking. In comparison to the previous~~
~~member, t~~The Ma'anshan Member of the Pliensbachian ~~age, diaplays~~ displays a prominent change in the distribution and
extent of red color sediment and pedogenesis. The reddish sediments extend through the entire member (comp. Figs. 6 and
S2) and can be observed across most of the GSB. Calcisols are documented in both the western and central GSB (Figs. 6, 7,
S1, and S2). Ten calcisol horizons were recognized at the Shaping section, Ya'an (Figs. 6 and S1).~~, and s~~Strongly leach~~ed~~~~ing~~
pedogenic structure~~s~~ and mudcrack~~s~~ are seen in Bed H8 of the Tanba section, Hechuan (Fig. 3a and 3d). ~~Other more~~
~~a~~Abundant calcretes within terrestrial red mudrocks ~~were~~ are widely described at ~~the~~ Gaoxian ~~section~~ of Dafang (Location
A8. Zhang et al., 2016), ~~the~~ Hulukou ~~section~~ of Weiyuan (Location A10. SBG, 1980a), ~~the~~ Geyaoguan ~~section~~ of Gulin
(Location A13. SBG, 1976), ~~the~~ Taiyuan ~~section~~ of Fengdu (Location A16. SBG, 1975), and ~~the~~ Yaxi ~~section~~ of Zunyi
(Location A17. Yang, 2015). ~~We interpret these calcretes were formed by the pedogenesiscalcisol origin.~~ The widespread
distribution of redbeds and calcisols (Figs. 4 and 7) denotes~~implies~~ an ~~a~~ intensification of the (semi-) arid climate ~~had been~~
~~intensified in the GSB during the Pliensbachian age~~.
Plant and sporopollen fossils also show~~indicate~~ a change to drier climate in the Pliensbachian. ~~With comparison~~Compared
to the ~~Zhenzhuchong and Dongyuemiao Members~~Sinemurian members, ~~much many fewer~~more plant fossils ~~were~~ are
reported in this member (e.g., Meng and Chen, 1997; Wang et al, 2010)~~, likely implying a rapid climatic change~~. The
Pliensbachian-Toarcian sporopollen assemblages are dominated by ~~classical~~ sporomorph genera assemblage
(*Dictyophyllidites--Cyathidites--Classopollis*), in which the dry-type gymnosperm spore *Classopollis* is more prevalent than
in the Hettangian-Sinemurian (Zhang and Meng, 1987)~~, also indicating~~consistent with ~~the interpretation of the intensification~~
~~of arid climate~~.
Similar dry temperate / subtropical climate ~~was verified by~~is interpreted for the upland coniferous forest in Qaidam Basin,
Northwest China (Wang et al., 2005)~~. In other hand~~ and~~,~~ by interdune playa mudstones of the Kayenta Fm ~~(e.g., Bromley,~~
~~1992) indicate similar arid climate~~ in Colorado Plateau ~~, western America~~(e.g., Bromley, 1992).~~However, at the same time~~
albeit~~,~~ it was ~~a~~the ~~probably~~ cool~~est / most~~ humid climate in South Kazakhstan, central Asia (Tramoy et al., 2016). ~~These~~
~~discrepancies might corroborate the unstable and heterogeneous climate in the mid-latitude area of North Hemisphere in the~~
~~Pliensbachian. In other hand, interdune playa mudstones of the Kayenta Fm (e.g., Bromley, 1992) indicate similar arid~~
~~climate in Colorado Plateau, western America.~~
**5.1.~~4)~~ The Toarcian Age**
In spite the fact that the Da'anzhai Member was deposited in the largest lacustrine transgression period (Fig. 7. details see
~~Appedix~~ supplementary data Note S1), abundant evidence for arid conditions, including backshore reddish mudro~~ucks~~cks with
calcisols, lacustrine ~~climate sensitive facies~~ micritic dolomites and / or gypsum, and stable isotopic geochemistry of
lacustrine carbonate, ~~together~~ indicate that the Toarcian ~~the~~ aridification could be the most intensive ~~in~~ of the late Early
Jurassic in the GSB.
Redbeds with abundant calcretes are well developed in the ~~Da'anzhai~~is ~~Member~~ member (Figs. 4 and 7). Four calcisols
~~horizons~~ in the Shaping section ~~, Ya'an~~ (Figs. 6 and S1) and the leaching/illuvial structure (Bed H13) in the Tanba section~~,~~
~~Hechuan~~ (Fig. 3c)~~,~~ were observed. Calcisols with ~~C~~calcretes also occur at sections of Dafang (Location A8. Zhang et al.,
2016), Nanxi (Location A11. SBG 1980a), Gongxian (Location A12. Liang et al., 2006), and Yunyang (Location A15. Meng
et al., 2005)~~, also record the occurrence of~~contain calcisols. The widespread occurance of calcisols within thi~~se~~s lacustrine
facies reveals that subaerial exposure of sediments often interrupted the lake environment, illustrating dynamic lake level
fluctuations ~~and aridification~~and an arid climate.
~~In addition to redbeds and calcisols, g~~Gypsum and micritic dolomites ~~(SBG, 1980a; Mo and Yu, 1987; Peng, 2009; and this~~
~~work) were~~are reported in the western and southern GSB (SBG, 1980a; Mo and Yu, 1987; Peng, 2009; and this work) (Figs.
1, 4, and 7). ~~It is plausible that gypsum and dolomites indicate arid climate type.~~ Tho~~u~~gh there are a number of hypothesies
on the dolomite formation in deep time,~~s have been in dispute for the significance of climate due to great deal~~such as
authigenic origin, diagenetic replacement, ~~of diagenetic dolomites~~ microbial mediation (e.g., Vasconcelos et al., 1995;
Mckenzie et al., 2009; Petrash et al., 2017), ~~in deep time,~~ a high abundance of dolomite was interpreted to form during
greenhouse periods, characterized by warm climates, probably reflecting favourable conditions for evaporite deposition and
dolomitization via hypersaline reflux (Warren, 2000). Dolomites are aslo thought the results of interplay of climate and
sea-level / base-level change (e.g., Newport et al., 2017) or are interacted with climatic regimes (Vandeginste et al., 2012).
~~Therefore, I~~it is plausible that ~~micritic~~ gypsum and dolomites ~~may indicate~~represent deposits of ~~arid/evaporate climate~~
~~when are associated with other climate sensitive sediments~~type. ~~That ,is So,, the~~The widespread micritic dolomites in the
Da'anzhai ~~Fm~~Member, which are associated with gypsum (Fig. 3f), ~~probably~~likely indicate ~~can serve the determination of~~
~~climate and suggest~~an arid climate in the central and western GSB (Fig. 1b). Gypsum occasionally occurs at Maliuping of
Hechuan (Fig. 3f) and Wujiaba of Zigong (SBG, 1980a)~~,~~ ~~implying~~showing a ~~short-term~~possible evaporitic climate in the
early Toarcian in the central GSB.
Carbon and oxygen isotopes of lacustrine carbonates further support the interpretation of an aird climate in the Toarcian age
in the GSB. ~~In general, -9.0‰ to -3.0‰ of $\delta^{13}C$ and $\delta^{18}O$ values represent a range of normal river lake and groundwater~~
~~carbonates (Alonso-Zarza, 2003). Therefore, the~~The mainly positive δ$^{13}$C values 0 to 2 ‰ (Fig. 5) from Hechuan (Wang et
al., 2006) indicate the lakes were brackish or even saline~~, and the~~. The relatively heavy negative δ$^{13}$C values -1‰ to -3.5 ‰
(Fig. 5) from Zigong (Wang et al. 2006) and Ya'an (this work) denote low depletions of $^{13}$C during calcite/aragonite
precipitation and mean that the lakes were possibly brackish. ~~In other hand, l~~Lightly negative δ$^{18}$O values -5‰ to -12 ‰ (Fig.
5) ~~dominate~~of the lacustrine carbonates, suggest~~ing~~ ~~that~~ closed lacustrine, palustrine and pond systems formed in a regional
arid-semiarid climate with ~~significant~~ evaporation ~~relative to~~exceeding precipitation.
The covariance of δ$^{13}$C and δ$^{18}$O is ~~also~~ a criterion to distinguish closed or open lakes (e.g., Talbot, 1990; Li and Ku, 1997).
~~That is, high δ$^{18}$O and low δ$^{13}$C values will be produced in relatively low temperature lake water when the covariation is~~
~~negative; high values of both δ$^{18}$O and δ$^{13}$C will be produced in high-temperature meteoric water and indicate increased~~
~~evaporation when the covariation is positive.~~ Pronounced positive covariances (R$^2$=0.44-0.96) between carbon and oxygen
isotopes (Fig. 5) indicate a typical arid-semiarid pattern of lakes in the central and western GSB.
The Da'anzhai Member has the same palynofloral assemblage with the Ma'anshan Member, in which the dry-type
gymnosperm spore *Classopollis* is ~~much more~~more abundant than in ~~previous~~underlying strata (e.g., Zhang and Meng, 1987;
Wang et al., 2010), supporting the aridification indicated by climate–sensitive sediments and stable isotope ratios of
lacustrine carbonates aforementioned.
Coastal Cheirolepidiacean (gymnosperm) forests indicate (temperate to subtropical) warm-humid climate punctuated by
locally dry and/or arid events in the Toarcian in Qaidam Basin, Northwest China (Wang et al., 2005). In Inner Mongolia of
North China, the thermophilous plants such as the dipteridaceous fern *Hausmannia*, bennettitales *Ptilophyllum*, display
similar warm and humid climate interrupted by hot and even arid conditions in a short intervals of the Toarcian (Deng et al.,
2017). The warm-wet climate was also indicated by assemblages of sporomorph and vegetation in the late Early Jurassic in
Jurong of Jiangsu, Lower Yangtze area (Huang et al., 2000). In South Kazakhstan, central Asia, paleoflora and δ$^2$H values
suggest slightly less humid and warmer conditions starting from the early Toarcian (Tramoy et al., 2016).
~~In summary, c~~Climate–sensitive sediments, ~~carbon oxygen~~carbon and oxygen isotope values and covariance, and
palynoflora, together indicate that an overall (semi-) arid climate dominated the GSB during the Early Jurassic, possibly
accompanied by occasional evaporitic climate. Relatively abundant calcisols suggest that the GSB was in a subtropical arid
zone based on the paleoclimatic zonation model of paleosols (Mack and James, 1994) during the middle-late Early Jurassic.
Through the Early Jurassic, this (semi-) arid climate in GSB is thoroughly comparable with the simultaneous arid climate
recorded in dryland lacustrine and eolian facies in Colorado Plateau~~, western America~~ (e.g., Blakey et al., 1988; Bromley,
1992; Tanner and Lucas, 2008; Parrish et al., 2017), but distinct from the relatively warm-humid climate indicated by
sedimentological and floral characteristics in North China (e.g., Wang et al., 2005, Deng et al., 2017) and in the ~~northern~~
~~margin of Gondwanaland,~~relatively high latitudes of Southern Hemisphere (~~Jansson et al., 2008;~~Pole, 2009).
In summary, the increasing aridity and warming in the GSB and arid climate in the Corlorado Plateau could have been

consecutive through the Early Jurassic, and seems not harmonizaed with the global fluactueated climate that could be imprinted by two large volcanic eruptions of the Central Atlantic magmatic province and Karro-Ferrar Large Igneous Province. The secular arid climate in the two areas might be more possibly constrained by paleotopography, where both were laid in the relatively low latitudes 15-30˚N (Fig. 1a).

## 5.2. $p\text{CO}_2$ perturbations and events

Pedogenic carbonates found in various continental settings precipitate in direct contact with soil atmosphere and bed rock and hold a meaningful signature of past climate (Alonso-Zarza and Tanner, 2006). ~~Ancient $p\text{CO}_2$ has been estimated by carbon isotope of pedogenic carbonates using the empirical (Cerling, 1991) and optimized (Ekart et al., 1999) formula. This paleosol method has roughly been applying in the Phanerozoic $p\text{CO}_2$ estimate (e.g., Cerling, 1991; Ekart et al., 1999; Retallack, 2001a) with >10 Myr interval of age resolution.~~ There are few high age resolution $p\text{CO}_2$ reconstructions for the Early Jurassic. The focus on $p\text{CO}_2$ estimates has on the event horizons, such as the transition of the Triassic to Jurassic (e.g., Tanner et al., 2001; Schaller et al., 2011). Herein we present ~~the~~ a $p\text{CO}_2$ estimate based on data from the GSB ~~in~~ at ~1.0 Myr age resolution for ~~a >~~ 120 Myr~~a~~ (199-179 Ma) interval of the Early Jurassic (Figs. 6 and ~~8c~~8a).

### 5.2.1. $p\text{CO}_2$ perturbation

Results of model estimates show that the $p\text{CO}_2$ values range 980-2610 ppmV with a mean 1660 ppmV in the Early Jurassic ~~except for the~~post the Hettangian and can be divided into three intervals (Figs. 6 and 8~~c~~a): phase I, stable 1500-2000 (mean ~1700) ppmV in the Zhenzhuchong and Dongyuemiao ~~m~~Members (Sinemurian age); phase II, main 1000-1500 (mean ~ 1300) ppmV in the Ma'anshan Member~~s~~ (Pliensbachian age); and phase III, great fluctuation 1094-2610 (mean ~1980) ppmV in the lower Da'anzhai Member (early Toarcian age).

The evolution and level of $p\text{CO}_2$ estimated by carbon isotope ratios of the pedogenic carbonates from the GSB ~~are roughly comparable~~compare favorably with the global composite based on the plant stomata method (data of the composite curve see Table S6~~6~~), but ~~difficult to compare~~show significant differences relative to the global composite $p\text{CO}_2$ based on paleosols (Fig. 8a~~c~~. Suchecki et al., 1988; Cerling, 1991; Ekart et al., 1999), which may be attributed to the shortage (<4 samples) of global data and large age uncertainties (Fig. 8a and Table S5~~5~~ and S6~~6~~). ~~On the other hand, the swing of~~The changes ~~the~~ in $p\text{CO}_2$ from the GSB, has a similar pattern to coeval seawater temperature estimates through the Early Jurassic although there are some discrepancies in pace ~~and at a high time resolution~~and in detail (comp. Fig. 8a~~b~~ and 8e~~b~~). That is, the relatively high $p\text{CO}_2$ 1500-2000 ppmV approximately corresponds to the relatively high seawater mean temperature -2ºC to +2ºC in the Sinemurian ~~age (Fig. 8b)~~, low $p\text{CO}_2$ 1000-1500 ppmV corresponds to low seawater mean temperature -5ºC to -2ºC in the Pliensbachian ~~age (Fig. 8b)~~, and quick rising $p\text{CO}_2$ of 1200 ppmV to ~2500 ppmV corresponds to the rapidly increased seawater temperature of -4ºC to +4ºC in the late Pliensbachian-early

Toarcian ~~(Fig. 8b)~~.
The $p$CO$_2$ record ~~and the carbon isotope of the marine carbonates are also somewhat~~ roughly ~~comparable~~ trends with the
carbon isotope records of marine carbonates and oganic matter ~~s in tendency~~ (comp. Fig. 8~~a~~a to 8d~~and 8e~~),
~~implying~~suggesting a possible linkage of the $p$CO$_2$ record in the GSB to the global carbon cycle ~~in total trend and rapid~~
~~change~~ (see section 5.2.2). Nevertheless, it is difficult for the proxies to compare ~~.~~ in a higher ~~time resolution~~detail, making it
difficult to relate the record to ~~even if it could be attributed to the low resolution of paleosol sample intervals and to the~~
orbital forcing of the global carbon cycle in the Sinemurian-Pliensbachian (Storm et al., 2020).
~~It has been disputed whether climate change was resulted from $p$CO$_2$ perturbation in the Phanerozoic (e.g., Veizer et al.,~~
~~2000; Crowley and Berner, 2001; Royer, 2006). For instance, the~~As a greenhouse gas, ~~a~~atmospheric CO$_2$ has a strong
control over global temperatures for much of the Phanerozoic (e.g., Crowley and Berner, 2001; Royer, 2006; Price et al.,
2013~~; Mills et al., 2019~~), but a decoupling of CO$_2$ and temperature has also been suggested (e.g., Veizer et al., 2000; Dera et
al., 2011; Schaller et al., 2011~~; Kashiwagi, 2016~~). The pattern of the Early Jurassic $p$CO$_2$ reconstructed from the carbon
isotope of pedogenic carbonates in GSB, Southwest China, supports the ~~coupling~~coupled relationship of CO$_2$-temperature ~~at~~
~~a ~1.0 Myr resolution scale. Even so, m~~Models of the coupling and decoupling of CO$_2$-temperature and CO$_2$-carbon cycle
have to consider: 1), age order of CO$_2$-temperature/carbon cycle relevance, i.e. they should be related in the same age (long
term or short term) hierarchy; 2) precise age constrain$_{ts}$ of individual CO$_2$ and temperature data; 3) methods of CO$_2$ and
temperature estimates, depending on precondition, presumptions, parameters, uncertainty, sample diagenesis, etc.; 4)
controls or influences of key factors such ice sheet, tectonic, paleogeography, cosmic ray flux, biota, volcanic eruption, and
so on.
**5.2.2. Rapid $p$CO$_2$ falling events**
The ~~recovered~~ GSB Early Jurassic $p$CO$_2$ curve reveals two rapid falling events (Fig. 6 and ~~8c~~8a). The first event (1E$_{CO2}$)
shows a quick drop from ~2370 ppmV (sample J1z-08-01 at depth 84.7 m) to 1350 ppmV (sample J1z-10-02 at depth 94.4 m)
near the boundary of the Dongyuemiao and Ma'anshan Members (Fig. 6), or to 1075 ppmV (sample J1z-11-02 at depth
111.7 m), which took place in the early Pliensbachian (~190.4-189.9/189.1 Ma. Fig. ~~8e~~8c). The extent of the rapid falling
$p$CO$_2$ is ~1000-1300 ppmV in 9.7-17.0 m. In other words, ~1000 ppmV drop could be accomplished within ~0.5-1.0 Myr
based on the estimate of ~~the rate of~~ sediment~~ation rate~~ ~~deposition~~ (Table S4).
While the corresponding early Pliensbachian climatic and isotopic-shifting events ~~cannot be~~are not observed in the smoothed
curves of the Early Jurassic seawater temperature and carbon cycle (Dera et al. 2011), the rapid falling event 1E$_{CO2}$ is well
correlated to the nearly coeval excursion events of ~~carbon oxygen~~carbon and oxygen isotopes recorded in western Tethys
and North Atlantic (Fig. 8). The 1E$_{CO2}$ compares well to: 1) the rapid carbon isotope negative excursion ~~–~~of (oysters,
belemnites, and brachiopods) shells from the Cleveland Basin, UK (Korte and Hesselbo, 2011) and northwest Algeria
(Baghli et al., 2020), 2) that of organic matter and marine carbonates from southern Pairs Basin (Bougeault et al., 2017; Peti,
et al., 2017) and Cardigan Bay Basin, UK (Storm et al., 2020), and 3) rapid oxygen isotope negative excursion (seawater
warming) of belemnites from northern Spain (van de Schootbrugge et al., 2005). The rapid change of the stable isotope
record had been called the Sinemurian-Pliensbachian boundary event (SPBE) and dated in the ammonite of the upper
*Raricostatum* - lower *Jamesoni* zones (Bougeault et al., 2017).
The second event 2E$_{CO_2}$ displays a large drop of 2574 ppmV (sample J1z-18-01 at depth 252.7 m) to 1094 ppmV (sample
J1z-19-01 at depth 272.3 m), ~1500 ppmV decrease within 19.6 m (estimated age interval ~0.8 Myr. Table S4 and Fig. 8~~e~~a).
Following the second drop, $p$CO$_2$ rises rapidly by ~1300 ppmV of 1094 ppmV to 2386 ppmV (sample J1Z-20-01 at depth
294.3 m) although only a few samples support the this cycle of $p$CO$_2$ falling-rising.
Strata in western Sichuan (Xu et al., 2017), may correlate to the time interval of the T-OAE, during which $p$CO$_2$ doubled
over background values, from ~1000 ppmV to ~2000 ppmV (e.g., Beerling and Royer, 2002; McElwain et al., 2005; Berner,
2006). Given ~~the~~ that chronostratigraphical correlation is challenging, the $p$CO$_2$ falling-rising cycle might correspond to the
quick shifting cycle of stable isotopes during the T-OAE (Fig. ~~8b~~ 8a and ~~8c~~8c-8d). In detail, the rapid falling-rising of $p$CO$_2$
is consistent with: 1) the quick negative-positive carbon isotope excursion of marine carbonates from Italy (Jenkyns and
Clayton, 1986; Sabatino et al., 2009), England and Wales (Jenkyns and Clayton, 1997), north Spain (van de Schootbrugge et
al., 2005), the Lusitanian Basin of Portugal (Hesselbo et al., 2007), Paris Basin (Hermoso et al., 2009), and Morocco (Bodin
et al., 2016); 2) that of invertebrate calcareous shells from the Cleveland Basin of UK (Korte and Hesselbo, 2011) and
northwest Algeria (Baghli et al., 2020); 3) that of marine organic matter from Morocco (Bodin et al., 2016), Yorkshire of
England (Cohen et al., 2004; Kemp et al, 2005), Cardigan Bay Basin of UK (Xu et al., 2018), northern Germany (van de
Schootbrugge et al., 2013), Alberta and British Columbia of Canada (Them II et al., 2017), northern Tibet of China (Fu et al.,
2016), and Japan (Izumi et al., 2018); 4) that of terrestrial organic matter from Sichuan Basin, China (Xu et al., 2017); and 5)
quick oxygen isotope negative-positive shifting (seawater warming) of brachiopods (Suan et al., 2008) and fossil wood
(Hesselbo et al., 2007) from the Lusitanian Basin, Portugal.
Multiple hypotheses have been proposed to interpret the 5°–6 °C decrease of sea surface temperatures in the late
Pliensbachian (Bailey et al., 2003; van de Schootbrugge et al., 2005; Suan et al., 2010) and warming ~8 °C in the early
Toarcian (Bailey et al., 2003; Suan et al., 2010), such as the sea level falling and rising ~~(Hallam, 1978; Hesselbo and Jenkyns,~~
~~1998)~~, methane release ~~(e.g., Hesselbo et al., 2000; Kemp et al, 2005; Hermoso et al., 2009; Them II et al., 2017)~~., ~~and the~~
Karoo–Ferrar eruption~~s (e.g., Hesselbo et al., 2000; Beerling and Brentnall, 2007; Bodin et al., 2016).~~, Hispanic corridor
opening, etc. Perhaps, these hypotheses somewhat explain the rapid change of sea surface temperatures, but ~~it remains~~
~~unclear how~~might not link ~~the hypotheses~~ to drastic falling of $p$CO$_2$ ~~in a high age resolution~~. As we know, atmospheric CO$_2$
is controlled by volcanism, weathering, vegetation on land and phytoplankton in ocean, and orbiting forcing. The Sr isotope
curve shows a rapid change in the early Toarcian but does not in the early Pliensachian (e.g., Jones et al., 1999), indicating a
distinct transfer of weathering took place on the land only at the T-OAE time. No robust evidence shows the rapid changes
of terrestrial vegetation and marine primary productivy for the two intervals excepet for the floral change in western Tethys
during the T-OAE (Slater et al. 2019). The Karoo–Ferrar eruption could be responsible for the rapid rising of $p$CO$_2$ but not
for the falling. Then the orbital forcing might be an alternative.
To sum up, the ~~perturbation and~~ rapid falling events of the Early Jurassic $p$CO$_2$ values ~~estimated from the carbon isotope of~~
~~pedogenic carbonates~~ in the GSB, are compatible with the response of stable isotopes (carbon cycle) and seawater
temperature from coeval marine sediments in a total tendency and eventful change, but not harmonized at a high-resolution
time scale. Whatever caused the rapid variations of sea surface temperatures, stable isotopes, and $p$CO$_2$, their near
concordance ~~implies~~ suggests that it is a positive feedback of the sea surface temperature and carbon cycle to the $p$CO$_2$ in
trend and event through the Early Jurassic; ~~accordingly, positive linkage could have taken place between the Early Jurassic~~
~~climate and $p$CO$_2$.~~ whereas the ~~uncomparibility at a high resolution time sacle~~higher frequency changes in the
Sinemurian-Pliensbachian ~~might~~ may support other causal driving of the climate, such as orbital forcing (Storm et al.,
2020)~~in the Sinemurian Pliensbachian of the Early Jurassic. Other more, as concluded in section 5.1, theThe~~in North from
~~suggested byf~~

**6. Conclusions**

Based on analyses of climate-sensitive sediments and stable isotopes ~~of the GSB, leading to a~~ and the reconstruction of
paleoclimate and $p$CO$_2$, we conclude:
1) ~~Climate-sensitive sediments and carbon-oxygen isotope values and covariances with palynofloral reference indicate that~~
~~a~~An overall warm-hot and (semi-) arid climate dominated the GSB during the Early Jurassic, possibly accompanied by
occasional evaporitic climate in the Toarcian. This (semi-) arid climate in GSB is comparable with that in Colorado Plateau,
western America, but distinct from the relatively warm-humid terrestrial climate recognized in other places of Chinese
mainland (e.g., Qaidam, Inner Mongolia, and Lower Yangtze) and the ~~northern Gondwanaland, relatively~~ high latitudes of
Southern Hemisphere~~;~~.
2) The Early Jurassic $p$CO$_2$ values ~~estimated from the carbon isotope of pedogenic carbonates in GSB~~ show that a range
between 980 ppmV and 2610 ppmV is ~3.5-10 times the pre-industrial value 275 ppmV and the mean 1720 ppmV is ~6
times the pre-industrial value.
~~3)~~ Three phases of $p$CO$_2$ values were distinguished: 1500-2000 (mean ~1700) ppmV in the Sinemurian age, 1000-1500
(mean ~ 1300) ppmV in the Pliensbachian age, and 1094-2610 (mean ~1980) ppmV in the early Toarcian. ~~The phases~~
~~manifest the perturbation of $p$CO$_2$ in the Early Jurassic.~~
~~4)~~ Two events of rapidly falling $p$CO$_2$ were also recognized: ~1000-1300 ppmV drop at the Sinemurian-Pliensbachian
boundary and quick falling (-rising) by ~1500 ppmV in the early Toarcian. The phases and events manifest the perturbation
of $p$CO$_2$ in the Early Jurassic.
3) The rapid falling events of $p$CO$_2$ are compatible with the response of stable isotopes and seawater temperature from the
coeval marine sediments, implying a positive feedback of climate to $p$CO$_2$ during the Early Jurassic. The perturbation and
rapid falling events of the Early Jurassic $p$CO$_2$ from the GSB are compatible with the carbon cycle and seawater temperature
from coeval marine sediments in the North Atlantic and western Tethys in a total tendency and eventful change, but not
consistent in between at a high time resolution. The compatibility suggests that it is a positive linkage of the sea surface
temperature and carbon cycle to the $p$CO$_2$ in whole trend and event through the Early Jurassic;. oOn the contrary, the
uncomparibilitydifferences at a high-resolution time sacle implies the differentadditional climate drivers, such as orbital
forcing are important in the Sinemurian-Pliensbachian record.

## Acknowledgements

We thank reviewerProfessors Helmut Weisserts and Dan Breecker xxx xxx for  careful scrutiny, constructive comments
and suggestions. It is acknowledged this research was supported by Natural Science Foundation of China (NSFC) project

629  41672097.

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

Biostratigraphy in Yangtze Gorges (4), Geol. Publ. House, Beijing, 408, 1987 (in Chinese with English summary).

**Figures**

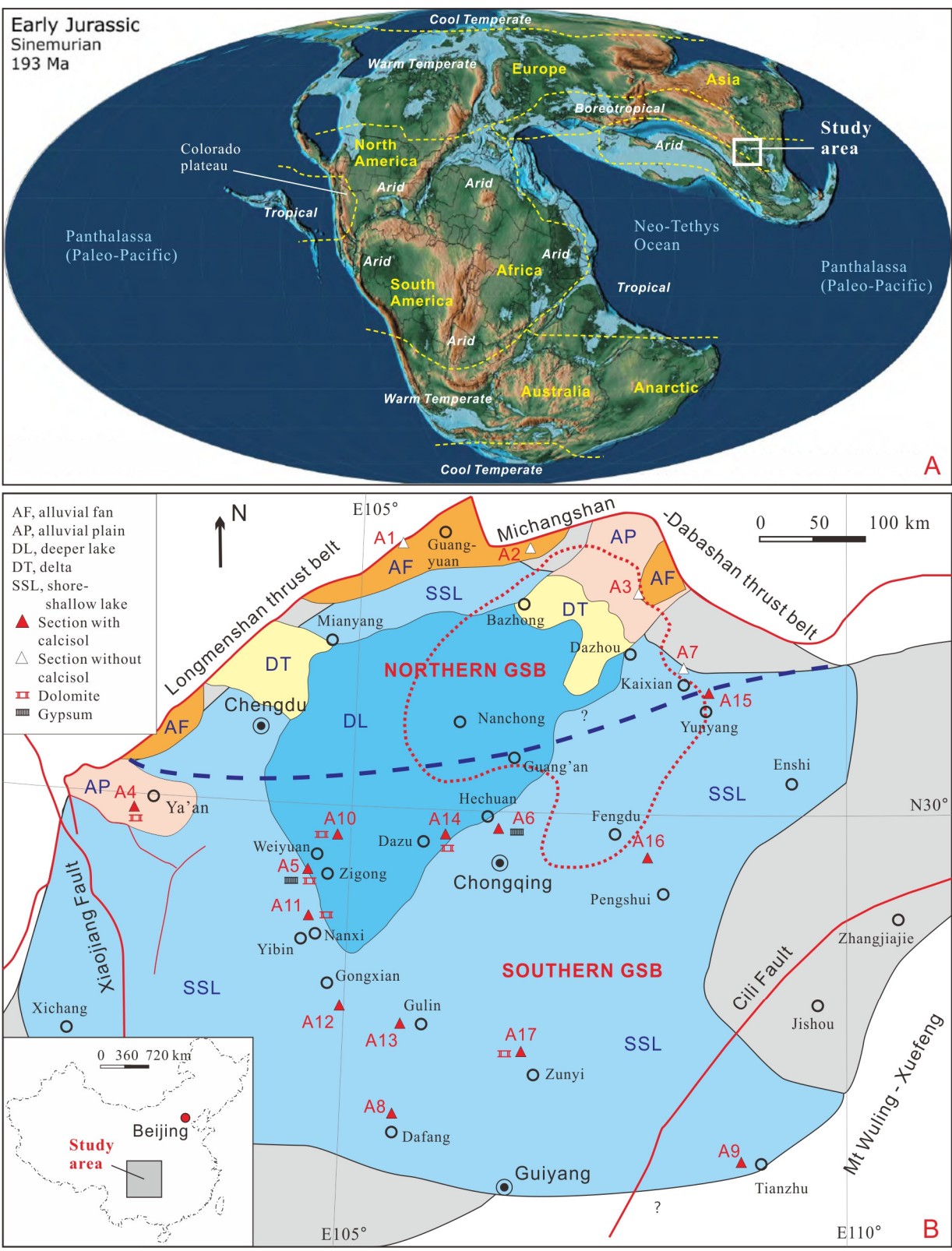

**Figure 1** A, Global Early-Middle Jurassic climate zones (Boucot et al., 2013) laid on the Early Jurassc (~193 Ma, Sinemurian) paleogeographic map (Scotese, 2014). B, Lithofacies paleogeographic sketch of the grand Sichuan paleobasin (GSB) in the early Early Jurassic (Zhenzhuchong and Dongyuemiao members) showing locations of the observed and analysed sections and climate-sensitve sediments. Lithofacies paleogeographic map was composed and modified from Ma et al. (2009) and Li and He (2014). Blue area is the extent of paleolake, estemted as ~380,000 km$^2$; blue + gray region is the basin shape, eitmated ~480,000 km$^2$. Dot red line confines the deeper lake area in the late Early Jurassic (Ma'anshan and Da'anzhai members). Bold dashed line is

the northern edge of calcisol occurrence, which may separate the climate of the GSB as the northern and southern types. Triangles with numbers are locations of observed and analysed sections: A1, Xiasi section, Jian'ge; A2, Puji section, Wangcang; A3, Shiguansi section, Wanyuan; A4, Shaping section, Ya'an (bed and thickness from Wen and Zhao, 2010); A6, Tanba and Maliping section, Hechuan (bed and thickness from Wang et al., 2010); A7, Wenquan section, Kaixian (thickness from Wang et al., 2010). Location and source data-s of sections A5 and A8-A17 (climate-sensitive sediments) refer to supplementary data Table S2S1.

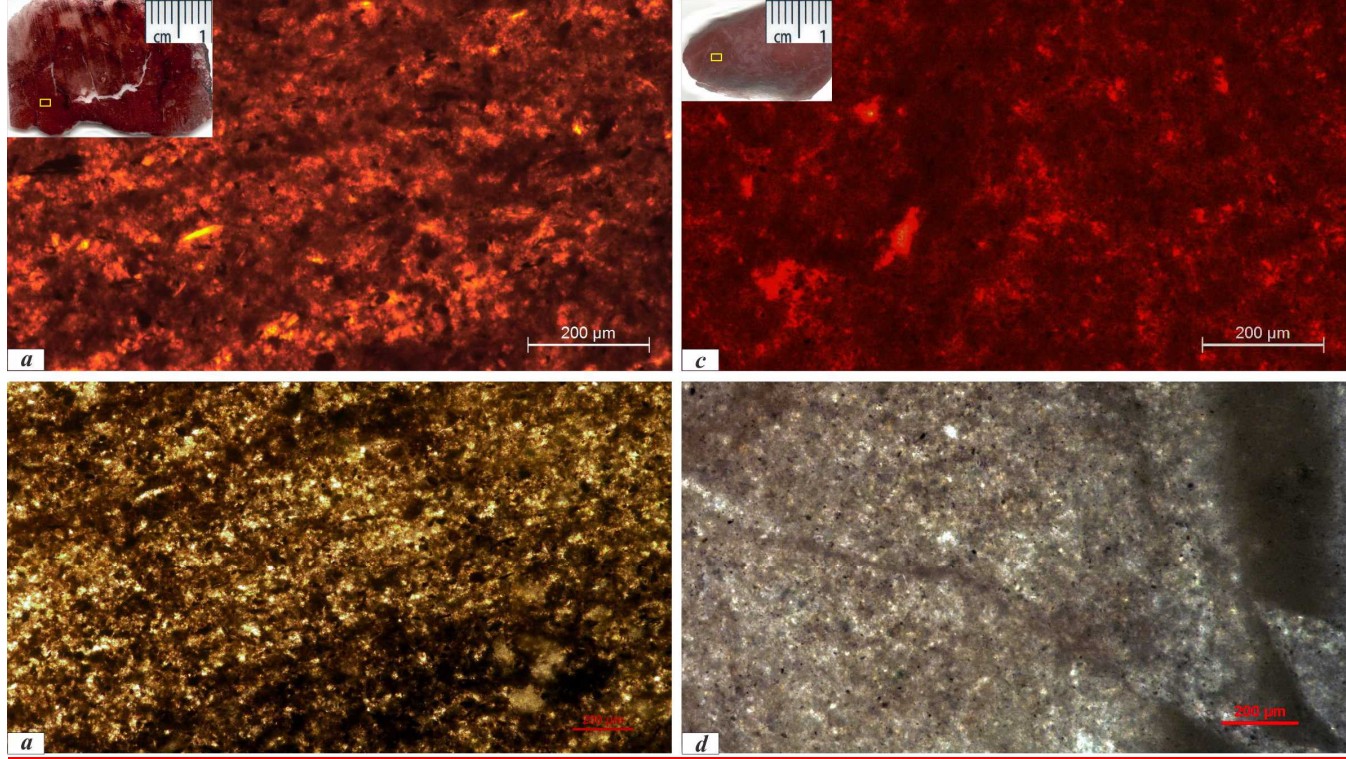

**Figure 2  Microscopic cathodoluminescence photos of representative calcrete samples from the Ziliujing Fm at the Shaping section, Ya'an. *a*, Sample J$_1$z-12-01, Bed B12, Ma'anshan Member; *b*, Sample J$_1$z-22-01, Bed B22, Da'anzhai Member. Pedogenic calcites are mainly nulllight orange and muds are not to non-luminescent, minor are orange/red luminescence. Pedogenic calcites of both samples are evenly luminescent light orange. Inserts are the scanned photos of thin-section, and rectangles are the area under cathodoluminescence and drilling.**

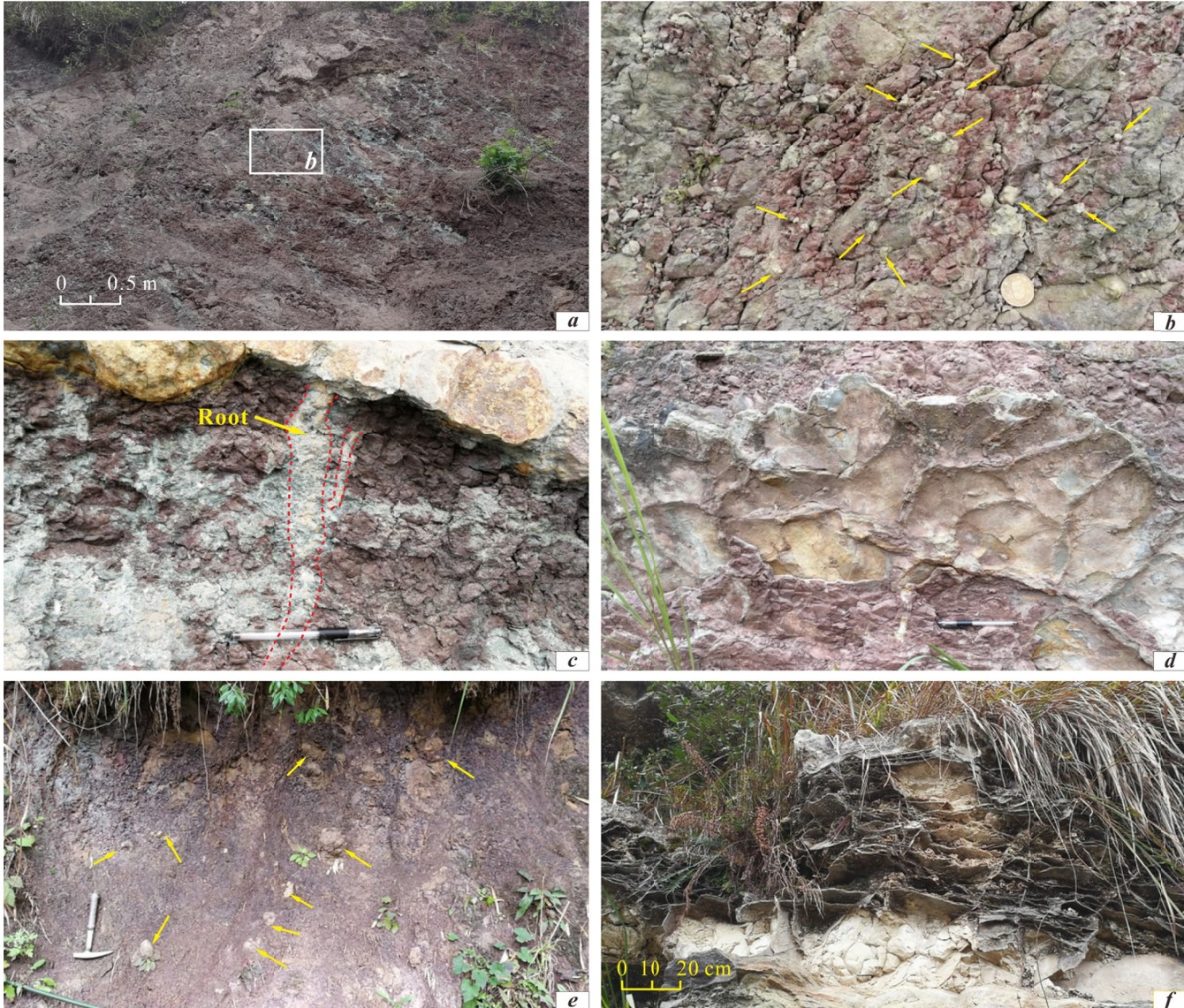

Figure 3    Field photographs of climate-sensitive sediments from the Lower Jurassic Ziliujing Fm in the GSB. *a*, Reddish purple calcisol with strong leaching structure. Lower Bed H8 of the upper Ma'anshan Member at Tanba village, Hechuan. *b*, Reddish purple calcisol showing the density and size of calcretes. The horizon and location same as *a*. Arrows point to calcretes. Coin 2.0 cm in diameter. *c,* Reddish purple calcisol with strong leaching structure and rhizoliths. Bed H13 of the top Ma'anshan Member at Maliuping, Hechuna. Pen 15 cm long. *d*, Mudcracks. Lower Bed H8 of the upper Ma'anshan Member at Maliuping, Hechuan. Pen 15 cm long. *e*, Brownish red calcisol with big calcretes (calcareous concretions). Arrows point to big calcretes. Calcisol horizon $J_1z$-10-01, Bed B10 of Ma'anshan Member at Shaping village, Ya'an. Hammer 34 cm long. *f*, Chicken-wire structure. Bed H12 of the Da'anzhai Member at Maliuping village, Hechuan.

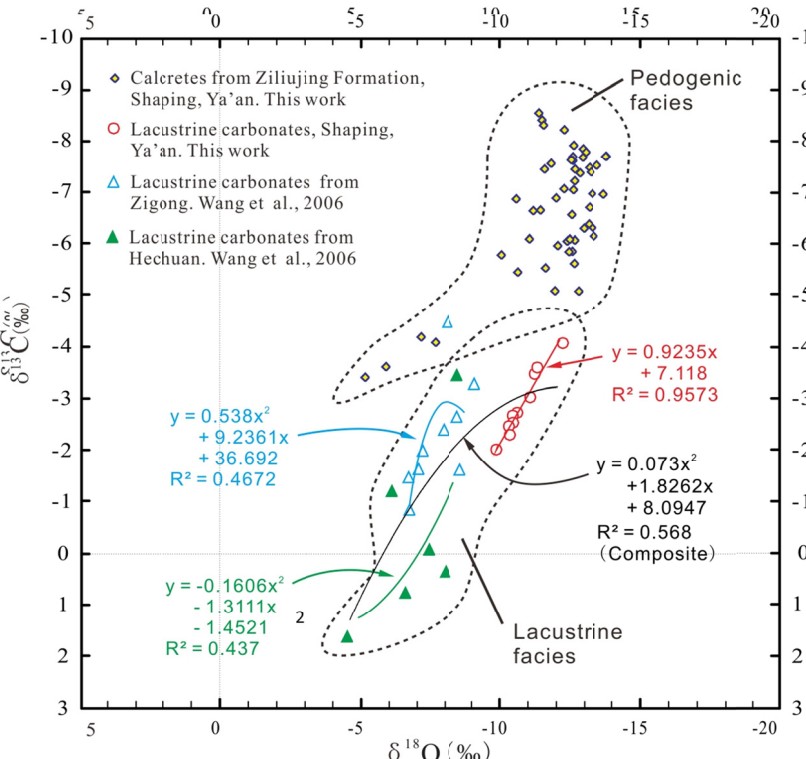

**Figure 4  Diagram showing the temporal and spatial variation of climate-sensitive sediments in GSB. Section loactions and data**
**sources refer to Table ~~S2~~S1.**


**Figure 5    Cross-plot and covariance of carbon and oxygen isotopic values of the Lower Jurassic pedogenic and lacustrine**
**carbonates from the GSB. Note, the pronounced covariance ($R^2$=0.957) between $\delta^{13}$C and $\delta^{18}$O from Shaping section, Ya'an,**
**indicating a compositional arid-evaporate and closed pattern lake; the moderate covariance ($R^2$=0.47 and 0.44) between $\delta^{13}$C and**
**$\delta^{18}$O from Zigong and Hechuan, indicating a (semi-) arid and semi-closed pattern lake.**




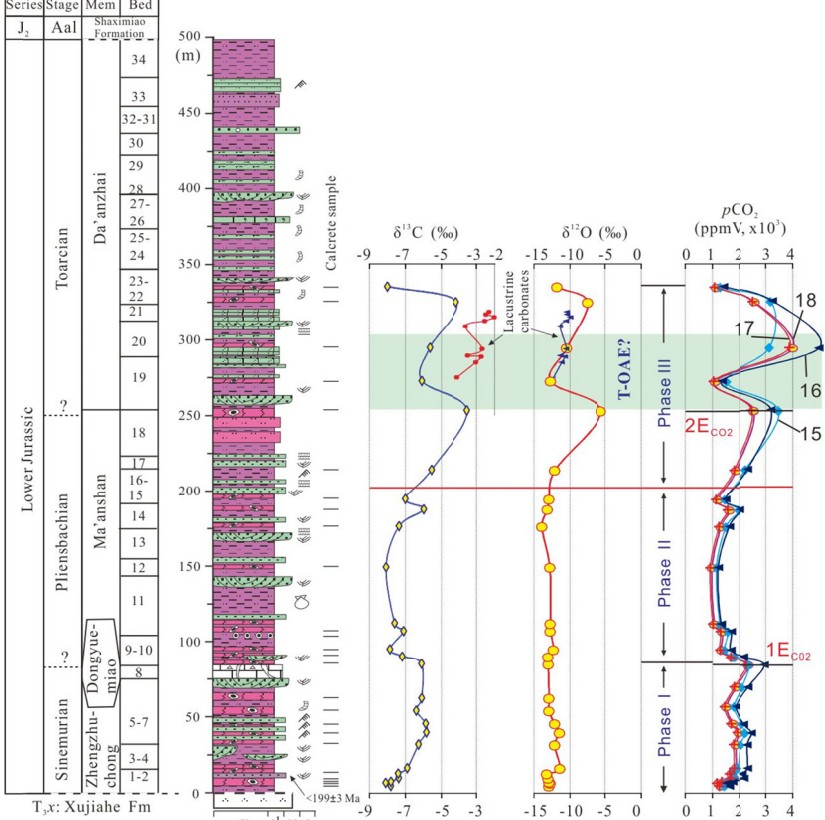

**Figure 6   Diagram of the Lower Jurassic strata and litholigcal log at the Shaping section, Ya'an with** ~~carbon-oxygen~~**carbon and**
**oxygen isotope values of pedogenic and lacustrine carbonates and $p$CO$_2$ cruve. Three phases and two events can be observed for**
**both stable isotope values of pedogenic carbonates and $p$CO$_2$ estimate. Legend of lithology in log refers to supplementary Figs. S1**
**and S2. T-OAE, Toarcian oceanic anoxic event. 1E$_{CO2}$ and 2E$_{CO2}$, rapid falling event of $p$CO$_2$. Numbers 15 to 18 are the curves of**
**$p$CO$_2$ in different parameters, and details refer to supplementary Table S4.**

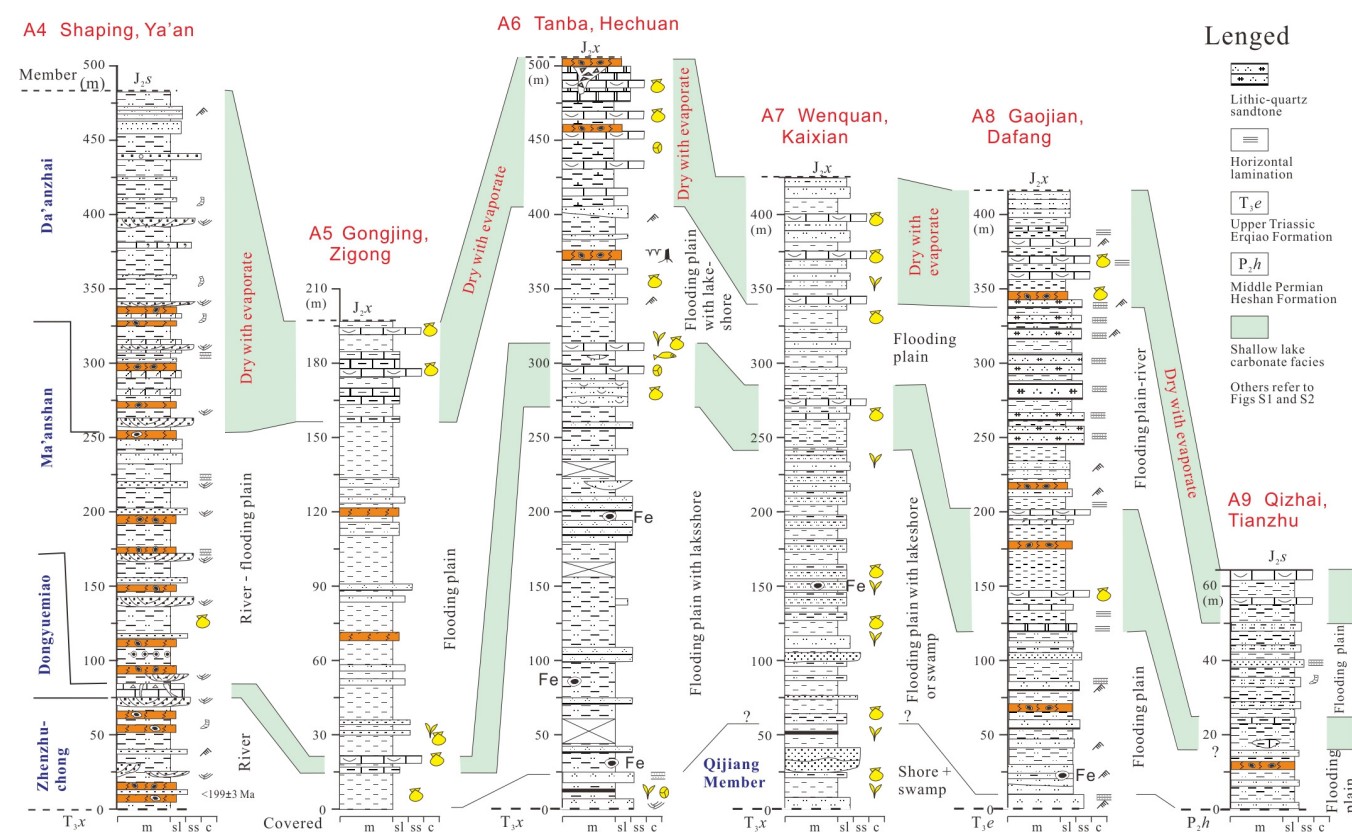

Figure 7 Stratigraphic correlation and depositional environment interpretation of the Lower Jurassic in the GSB. Data of sections refer to ~~f~~Figure~~.~~ 1 ~~and Table S2~~. Note, two lacustrine transgressive cycles are marked by correlative pale green areas.

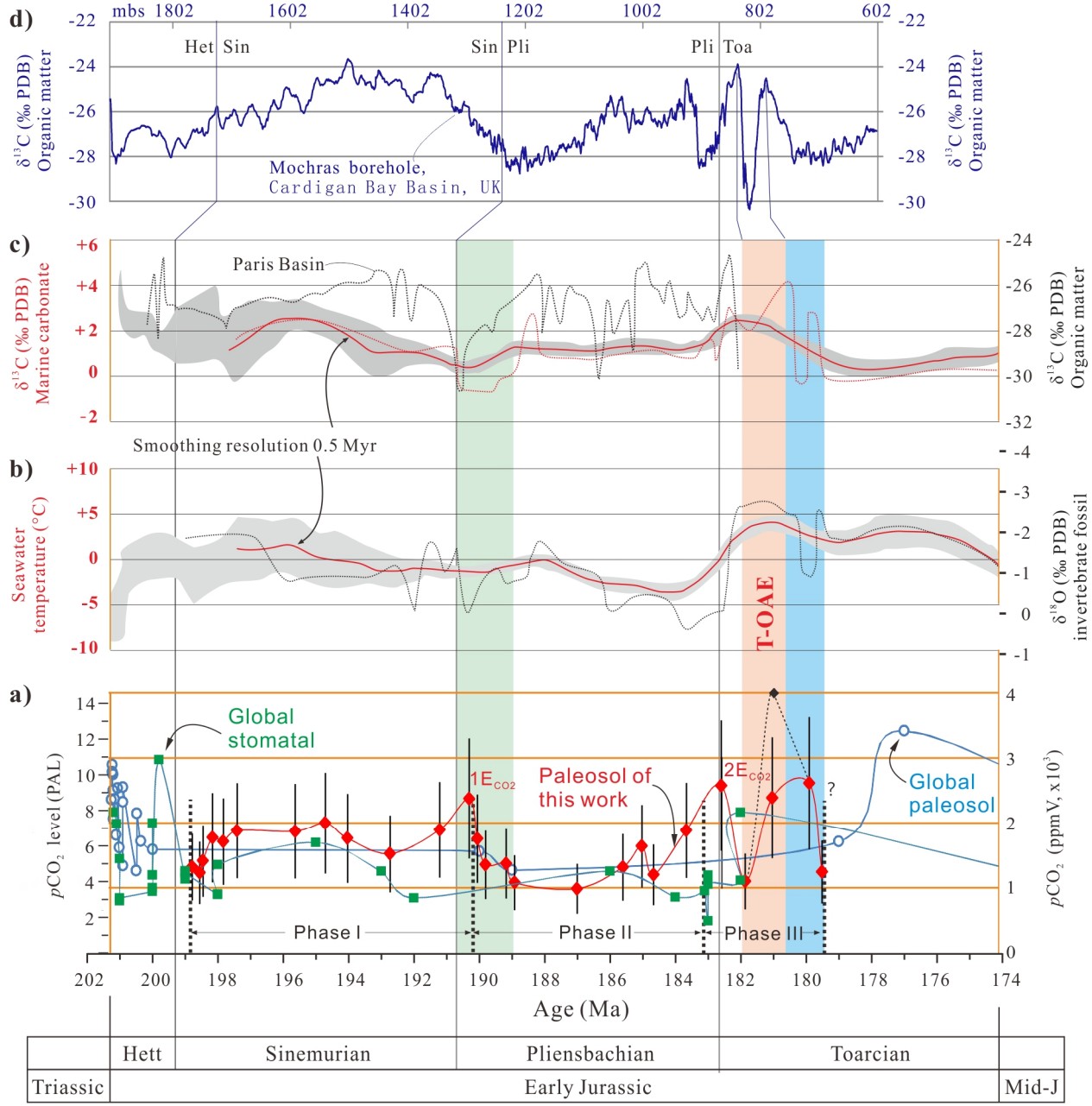

**Figure 8  Comparison among the Early Jurassic $pCO_2$, $\delta^{13}C$ of marine carbonates and organic matters, $\delta^{18}O$ of invertebrate fossils, and seawater temperature. Age model is from Cohen et al. (2013). ~~e~~a), $pCO_2$ values of this work and~~,~~ the composite $pCO_2$ by paleosol and stomatal index ~~collected from the published literatures refer t~~(~~o~~ supplementary Table ~~S5~~ S6 and ~~S6~~S7). Vertical bars are errors (1σ) of $pCO_2$ (Table S5). Errors are propagated using the Gaussian approach (Breecker and Retallack, 2014). Note:** 1) $pCO_2$ = 4027 ppmV (black solid diamond, sample J1z-20-01) if the $\delta^{13}C_r$ = -29.0 ‰ at 181 Ma from Xu et al. (2018) in case of other constant parameters; 2) the early published $pCO_2$ values from both carbon isotope of pedogenic carbonates and stomatal index of fossil plants (data refer to Table S~~5~~6 and ~~S6~~S7) were awfully rough dated with the average age of a lithostratigraphic formation or group, with which the uncertainty can be upto 10 Myr, leading to the difficulty of precise and accurate $pCO_2$ correlation in pace, frequency, and event~~in deep time~~. b), $\delta^{18}O$ and seawater temperature (black dot line) of marine invertebrate fossils compiled from ~~McArthur et al. (2000),~~ Rosales et al. (2001, 2004), Jenkyns et al. (2002), Bailey et al. (2003), van de Schootbrugge et al. (2005), Gómez et al. (2008), Metodiev and Koleva-Rekalova (2008), Suan et al. (2008), Korte et al. (2009), Dera et al. (2011), Gómez et al. (2015). ~~Smoothed δ^18O and seawater temperature (red curves) in b) and c) are after Dera et al. (2011).~~ c), red dot line $\delta^{13}C$ ~~(red dot line)~~ of marine carbonates and organic matters in western Tethys, composed from Jenkyns and Clayton (1986, 1997), Hesselbo et al. (2000), Dera et al. (2011), Arabas et al., 2017; black dot and solid line $\delta^{13}C$ ~~(black dot and solid line)~~ of organic matters from Paris Basin, France (Peti et al., 2017)~~and Cardigan Bay Basin, UK (Xu et al., 2018)~~. Smoothed $\delta^{18}O$ and

seawater temperature (red curves) in b) and c) are after Dera et al. (2011). d), δ¹³C of organic matters from North Atlantic. Composed from the Mochras borehole, Cardigan Bay Basin, UK (Xu et al., 2018; Storm et al., 2020), seven-point average smoothing against depth (mbs).

**Table**

**Table 1   Stratigraphic framework of the Lower Jurassic Ziliujing Fm in Sichuan and adjacent area (GSB), Southwest China**

| Epoch | Age | Formation | W Sichuan (Ya'an) | E Sichuan and Chongqing | S Sichuan and N Guizhou | N Sichuan |
|---|---|---|---|---|---|---|
| Middle Jurassic | Aalenian | Xintiangou Fm | Xintiangou Fm | Xintiangou Fm | Xintiangou Fm | Qianfuyan / Xintiangou Fm |
| Early Jurassic | Toarcian | Ziliujing Fm | Da'anzhai Mem (Bed 20-34) | Da'anzhai Mem | Da'anzhai Mem | Baitianba Fm |
| | Pliensbachian | | Ma'anshan Mem (Bed 9-18) | Ma'anshan Mem | Ma'anshan Mem | |
| | Sinemurian | | Dongyuemiao Mem (Bed 8) | Dongyuemiao Mem | Dongyuemiao Mem | |
| | | | Zhengzhuchong Mem (Bed 1-7) | Zhengzhuchong Mem | Zhengzhuchong Mem | |
| | Hettangian | | Hiatus | Qijiang Mem | Qijiang Mem | ? |
| Late Triassic | Rhaetian | Xujiahe Fm | Xujiahe Fm | Xujiahe Fm | Xujiahe Fm | Xujiahe Fm |

Notes:     Straigraphic classification and correlation were composed from Dong (1984); SBGM (1997), Wang et al. (2010), Wen and Zhao (2010), Xu et al (2017). Re-Os isotope age of the lower Da'anzhai Member is 180.3 ± 3.2 Ma in western Sichuan (Xu et al., 2017). Fm, Formation; Mem, Member.

**Supplementary data**

**Captions of supplemenatary figures**

**Figure S1   Lithological log of the Lower Jurassic Ziliujing Fm with depositional environment interpretations and sample positions at the Shaping section, Ya'an of Sichuan. Bed number and thickness are partly referred to Wen and Zhao (2010).**

**Figure S2   Lithological log of the Lower Jurassic Ziliujing Fm at the Tanba-Maliuping section, Hechuan of Chongqing with depositional environment interpretations and sample positions. Bed number and thickness are partly referred to Wang et al (2010).**

**Figure S3   Field photographs of the Lower Jurassic Ziliujing Fm lithofacies in the GSB. *a*, Well roundness and sorting gravels in the alluvial fan conglomerate. Basal and lower Baitianba Fm. Puji, Wangcang. Hammer 30 cm long. *b*, Large trough cross-bedding with scours in the point bar and channel sandstones. Upper Baitianba Fm; Puji, Wangcang. *c*, Calcisol developed within strong leaching overbank mudrocks on channelized sandstones. Middle of Bed B2, the Zhenzhuchong Member. Shaping section, Ya'an. *d*, Purple red mudrocks intercalated with thin siltstones in flood plain facies. Bed H7 of the Ma'anshan Member, Tanba section, Hechuan. *e*, Whitish medium-thick micritic dolomites in lacustrine facies. Bed H12 of the Da'anzhai Member. Maliuping section, Hechuan. Hammer 34 cm long. *f*, Greeinsh gray lacustrine muddy dolomites and dolomitic mudrocks**

associated with brownish / reddish purple mudrocks. Bed B21 of the Da'anzhai Member, Shaping section, Ya'an.
**Figure S4**  Microscopic photos showing lithological microfacies of the Lower Jurassic Ziliujing Fm. *a*, Fine lithic (quartz)
sandstone. Lithic-dominant fragments are mudrock. Sample $J_1z$-02-01b, Zhenzhuchong Member, Shaping section, Ya'an.
Plain-polarised light. *b*, Laminated muddy dolomite and dolomitic mudrocks. Sample $J_1z$-21S2B, Da'anzhai Member, Shaping
section, Ya'an. Plain-polarised light. *c*, Fine quartz arenite. Sample 18HC-02b3, Bed H2, Qijiang Member, Tanba section,
Hechuan. Cross-polarised light. *d*, Micritic dolomite. Sample 18HC-06b, Bed H12, Da'anzhai Member, Maliuping section,
Hechuan. Plain-polarised light. *e*, Coquina. Shell wall of bivalves were micritized. Mud and recrystalline calcites filled inter-shells
and intra-shells. Sample 18HC-04b, Base of Bed H12, Da'anzhai Member, Maliuping section, Hechuan. Cross-polarised light. *f*,
Relict of coquina. Shell wall of bivalves were parly micritized. Strongly recrystalline calcites replaced the fills and shells. Sample
18HC-05b, Bed H12, Da'anzhai Member, Maliuping section, Hechuan. Cross-polarised light.
**Figure S5**  Field photographs of the Lower Jurassic Ziliujing Fm lithofacies in the GSB. *a*, Lithofacies and stratigraphic sequence.
Beds B8 to B10 of the lower Ma'anshan and Dongyuemiao Members at Shaping, Ya'an. *b*, Karstified gravels within
the limestone. The horizon and location is same as *a*. Pen 15 cm long. *c*, Layered dolomites with Karstified cave gravels. Bed H12
of the Da'anzhai Member at Maliuping, Hechuan. *d*, Karstified cave gravels. The horizon and location is same as *c*.
Hammer 34 cm long.
**Figure S6**  Stratigraphic correlation of the Lower Jurassic Baitianba Fm in northern GSB. Locations and sources refer to Figure
1. Plant fossils and stratal thickness in the Shiguansi section, Wanyuan are cited from SBG (1980b).
**Captions of supplemenatary tables**
**Table S1    Occurrence list of the Early Jurassic climate-sensitive sediments in the GSB**
Table S2    Early Jurassic paleosols in Ya'an of Sichuan and Hechuan of Chongqing, Southwest China
Table S3    Carbon-oxygen isotope compositions of lacustrine carbonates from the Lower Jurassic Ziliujing Fm (Da'anzhai Mem)
in the GSB
Table S4    $p$CO$_2$ estimate by carbon isotope of pedogenic carbonates from the Lower Jurassic Ziliujing Fm at Shaping section,
Ya'an of Sichuan
~~Table S4    Occurrence list of the Early Jurassic climate-sensitive sediments in the GSB~~
Table S5    Calculation of Gaussian error propagation for the Early Jurassic $p$CO$_2$ estimate in the Sichuan paleobasin
Table S6    Global $p$CO$_2$ data of the Latest Triassic - Early Jurassic by stomatal method
Table S7    Global $p$CO$_2$ data of the Latest Triassic - Early Jurassic estimated by carbon isotope of pedogenic carbonates
**Captions of supplemenatary notes**
Note S1, Description and interpretation of sedimentary facies and its evolution

**Note S2, Notes of parameter usage and selection for the $p$CO$_2$ calculation**