# Peer review of "Early Jurassic climate and atmospheric CO2 concentration in the Sichuan paleobasin, Southwest China"

_Climate of the Past, 2020_

## Referee Comment (RC1) · Helmut Weissert (Referee) · 9 Jun 2020

General comments

In their CP manuscript, Li and colleagues trace the evolution of Early Jurassic climate and environment in lacustrine and pedogenic carbonates from the Sichuan Basin in SW China. They use "climate-sensitive sediments" and C- and O-isotope geochemistry of pedogenic carbonates as proxies for past climate and atmospheric pCO2. The data presented are new and the pCO2 calculations based on C-isotope composition of pedogenic carbonate will be of use in any future Jurassic climate reconstructions. Data presentation is fine, eg. fig 6 is very useful. Language needs considerable improve-

ment (see below). Im not entirely happy with the structuring of this paper. I recommend, to start the study with well-defined hypotheses, questions, best based on climate models (see below). I also recommend to restructure the discussion (see below, so it will be easier to follow for the reader. Even if the new pedogenic carbon isotope data are a valid contribution for paleoclimate science, I can, recommend publication of these data only after major revision of this paper.

Detailed comments:

1)Introduction The authors start their study with a rather general sentence: "The Jurassic was a typical greenhouse period" > what is a "typical greenhouse" ? The Jurassic was an interesting time because of major plate tectonic changes affecting global climate. The fragmentation of Pangea resulted in the collapse of a Monsoonal-type climate in the earliest Jurassic and by the end of Jurassic a more zonal climate was established. Jurassic was, no surprise, a time with cooler and warmer climate, possibly even with Ice Age episodes and the new data in this study confirm that changes in Early Jurassic $pCO_2$ were considerable. In their study, the authors compare data collected from the Sichuan Basin with data from the Colorado Plateau and from North China and northern Gondwanaland. As a reader I am, of course, interested in the paleogeography of the Early Jurassic and in the relative paleopositions of the study site and of other localities mentioned in the text. I recommend that the authors add a figure showing the plate tectonics of the Early Jurassic and showing the locations mentioned in their study. In addition to paleogeography, paleoclimate models could help to better understand climate trends in SW China and in North America, both discussed in this study. However, there are almost no models available on Early Jurassic climate. One of the few simulations providing information on precipitation pattern and aridity in the Early Jurassic can be found in Robertson et al. (Sedimentology, 2017). The authors may refer to this study. Models may help to better formulate hypotheses and questions addressed in this study. And, clear hypotheses and questions facilitate reading of the paper . » The authors may considerably improve their introductory part. Time window chosen: Your time window is defined by post-mass extinction time (Hettangian) at the base and the Toarcian OAE at the top. If you extend part of your discussion to the Mesozoic-Cenozoic (terrestrial proxies have begun to provide information. . .line 41) then you soon are moving on slippery grounds: Many, many studies on terrestrial climate exist for the Late Mesozoic and even more for the Cenozoic. Focus on your time window, even in your figures (fig 8).

2) Material and methods The methods you apply are up-to-date and they provide useful information on past climate. I recommend to add a paragraph on your selection procedure of CO2 concentration in soils. You have this discussion in your notes, that is fine. However, the S(z) value is a crucial value for your pCO2 estimates, please include your selection arguments in your main text. See also your line 162, where you may discuss the S(z) selection procedure if you did not do it in your method paragraph.

3) Discussion of your data In your discussion, you present first your sediment proxy data stage by stage, then, in a next chapter, you discuss your pCO2 data through the Early Jurassic. This structure of the text makes the reading of the discussion, at least for me, quite difficult. I prefer to see a climate discussion starting with the drivers of climate change, in this case changes in carbon cycling and in pCO2, which are both global signals (See, for example, the new and detailed C-isotope curve for the Hettangian-Pliensbachian in Storm et al.,PNAS, 2020). You can, if you take a general reference curve as a start, project your data into this curve and test, if pCO2 trends coincide, for example, with C-isotope trends, you can see if regional climate pattern (China, N-America etc) reflects some of the global trends. This may make the paper much easier to read and you can easily show the regional pattern as part of a global climate curve. You may refer in this discussion, if possible, to the few available model data. Of course, you also will use literature data you chose for your study and your figure 8 (e.g. Dera et al. 2011 and many others, as you cited correctly). A discussion starting with the global pattern also can make it easier to understand your comparison with regional data from N. America.
[Figure]

4) Details and corrections, including some comments on language Abstract ". . .terrestrial sediments show more complicated environment and climate. . ." I assume that you want to say that climate proxies in terrestrial sediments are more complex and more difficult to interpret than many marine proxies. Please do not write "carbon-oxygen isotopes", but carbon and oxygen isotopes. line 42 You are rather imprecise in your wording, when you write about a "negative feedback . . .. has been hypothesized to account for. . . in the carbon cycle" > what to you exactly mean? 105 . . .descriptions for sedimentary facies analysis were executed. . . 116 the description how to distinguish dolomite from a calcite the field is not really needed. It should be basic knowledge for students in geology. . . 216 . . .a distinct transfer of climate.. ??? 239 . . . a warm-humid climate followed the Late Triassic. . .. > Late Triassic climate 244-248 You compare North American climate with GSB climate throughout your study, this is ok, however, I like to know why you chose North America, what are hypotheses on climate similarities and differences between N. America and SW China. 248 » here it will be interesting to discuss climate pattern in the Early Jurassic, why is which regional climate similar/different from another regional climate? 305 Interesting is the observation that lake facies was widespread during a dry period in the Toarcian.You may further comment on this. 318 you use dolomite formation as a climate proxy, this is fine. You may refer to literature on dolomitization along the Persian Gulf ("Arabian Gulf") , eg. by McKenzie and others. 120 chichen-wire > chicken-wire (anhydrite) 253 That is the reddish rocks developed through the whole member. . .. , but it started. . . revise 257 . . .. Calcisols were also interpreted with the description of abundant calcretes. . .. (??) revise style 282 The Ma'anshan Member is likely the Pliensbachian. . .. 302 In other hand,. . . 321 . . .can serve the determination. . . I did not mark the many additional small language inconsistencies » please revise text carefully.

Figures I like your figure 6 which serves as a very good baseline for your discussion (you may also add data from Xu et al . from the same study area in figure 6 or elsewhere?)

Fig 8 You plot changes in temperature not temperature, please correct this in the figure. The global paleosol curve is, in this case not very helpful. I also wonder, if you should start your curve after the T-J boundary (?). The high-resolution data are not really discussed in your work and you did not include any T-J boundary data.

---

## Author Comment (AC1) · 28 Jun 2020

Dear professor Helmut Weissert,

We really appreciate the careful scrutiny that you took on our manuscript. The constructive comments and suggestions have pushed us to greatly improve the article. Below are replies to your comments.

1) Introduction The authors start their study with a rather general sentence: "The Jurassic was a typical greenhouse period", what is a "typical greenhouse" ? The Jurassic was an interesting time because of major plate tectonic changes affecting global climate. The fragmentation of Pangea resulted in the collapse of a Monsoonal-type climate in the earliest Jurassic and by the end of Jurassic a more zonal climate was established. Jurassic was, no surprise, a time with cooler and warmer climate, possibly even with Ice Age episodes and the new data in this study confirm that changes in Early Jurassic pCO2 were considerable. In their study, the authors compare data collected from the Sichuan Basin with data from the Colorado Plateau and from North China and northern Gondwanaland. As a reader I am, of course, interested in the paleogeography of the Early Jurassic and in the relative paleopositions of the study site and of other localities mentioned in the text. I recommend that the authors add a figure showing the plate tectonics of the Early Jurassic and showing the locations mentioned in their study. In addition to paleogeography, paleoclimate models could help to better understand climate trends in SW China and in North America, both discussed in this study. However, there are almost no models available on Early Jurassic climate. One of the few simulations providing information on precipitation pattern and aridity in the Early Jurassic can be found in Robertson et al. (Sedimentology, 2017). The authors may refer to this study. Models may help to better formulate hypotheses and questions addressed in this study. And, clear hypotheses and questions facilitate reading of the paper . Âż The authors may considerably improve their introductory part. Time window chosen: Your time window is defined by post-mass extinction time (Hettangian) at the base and the Toarcian OAE at the top. If you extend part of your discussion to the Mesozoic-Cenozoic (terrestrial proxies have begun to provide information: line 41) then you soon are moving on slippery grounds: Many, many studies on terrestrial climate exist for the Late Mesozoic and even more for the Cenozoic. Focus on your time window, even in your figures (fig 8).

We deleted the sentence "The Jurassic was a typical greenhouse period". This gets rid of greenhouse, but still expresses that the Jurassic was hotter than today. More works (almost rewriting) were made in the paragraph, with suggestions of the referee, to enhance expression of the Early Jurassic changeable and oscillating climate.
Following the suggestion, we have added a global map of climatic paleogeography, on which locations of the study area and correlative Colorado Plateau are marked. We replaced the previous Fig. 1 as Fig. 1b and the new paleogeographic map as Fig. 1a. Please see the new Fig.1. We laid the global Early-Middle Jurassic climate zones (Boucot et al., 2013) on the Early Jurassic ( 193 Ma, Sinemurian) paleogeographic map (Scotese, 2014), instead of a GCM map selection in the paper by Robinson (not Robertson) et al. (2017. Fig. 5, page 223, 64(4), Sedimentology). This is because oceans and continents (plates) are not distinct and climatic zones are not shown in the Early Jurassic GCM maps.

We further deleted the Gondwanaland term in the text, because it was a superland (composed of Antarctic, Africa, South America, India, Australia, New Zealand, etc.) of the South Hemisphere during the Paleozoic, and began to disassemble after the Pangea break-up since the Late Triassic, and it was not an independent supercontinent when the Pangea formed.

For the time window, we focused the interval of the Sinemurian to the early Toarcian in section 5.2 (pCO2 discussion) and figure 8 although we still keep the Hettangian in section 5.1 (climate evolution) as the Hettangian sediments can be observed in some investigated sections, and they show the climate transition from more humid to arid into the Jurassic.

2) Material and methods

The methods you apply are up-to-date and they provide useful information on past climate. I recommend to add a paragraph on your selection procedure of CO2 concentration in soils. You have this discussion in your notes, that is fine. However, the S(z) value is a crucial value for your pCO2 estimates, please include your selection arguments in your main text. See also your line 162, where you may discuss the S(z) selection procedure if you did not do it in your method paragraph.

We have deleted the supplementary data Note S2, and placed the main part in the section 3.3 (Calculation of atmospheric CO2 concentration). Then we added the notes for the procedure of CO2 concentration estimate and S(z) value selection in the revision. Totally 27 lines (line 155 to 181) were added.

3) Discussion of your data

In your discussion, you present first your sediment proxy data stage by stage, then, in a next chapter, you discuss your pCO2 data through the Early Jurassic. This structure of the text makes the reading of the discussion, at least for me, quite difficult. I prefer to see a climate discussion starting with the drivers of climate change, in this case changes in carbon cycling and in pCO2, which are both global signals (See, for example, the new and detailed C-isotope curve for the Hettangian-Pliensbachian in Storm et al., PNAS, 2020). You can, if you take a general reference curve as a start, project your data into this curve and test, if pCO2 trends coincide, for example, with C-isotope trends, you can see if regional climate pattern (China, N-America etc) reflects some of the global trends. This may make the paper much easier to read and you can easily show the regional pattern as part of a global climate curve. You may refer in this discussion, if possible, to the few available model data. Of course, you also will use literature data you chose for your study your figure 8 (e.g. Dera et al. 2011 and many others, as you cited correctly). A discussion starting with the global pattern also can make it easier to understand your comparison with regional data from N. America.

In the section 5 of the revision, we start with an introduction of the globally oscillated Early Jurassic climate with causes. Then we discuss the climatic change of the GSB based on sedimentary proxies and compare to other places of the world. In section 5.2, the pCO2 record and rapid change (event), and correlation to the global climate (sea water temperature) and carbon cycle are discussed. This contexture aims to test the hypotheses from the marine climate records.

In figure 8, we assembled the Early Jurassic carbon isotope ratio curve of organic matter (Fig. 8a) from the Mochras borehole, Cardigan Bay Basin, UK (Xu et al., 2018;

Storm et al., 2020) with seven-point average smoothing against depth (mbs). We did not combine the carbon isotope ratio curves of organic matter from the Paris Basin with that of Cardigan Bay Basin into one because the former is mainly against depth, and the latter against time.

4) Details and corrections, including some comments on language

Abstract : : :terrestrial sediments show more complicated environment and climate: : :" I assume that you want to say that climate proxies in terrestrial sediments are more complex and more difficult to interpret than many marine proxies. Please do not write "carbon-oxygen isotopes", but carbon and oxygen isotopes. line 42 You are rather imprecise in your wording, when you write about a "negative feedback : : :. has been hypothesized to account for: : : in the carbon cycle" > what to you exactly mean? 105 : : :descriptions for sedimentary facies analysis were executed: : : 116 the description how to distinguish dolomite from a calcite the field is not really needed. It should be basic knowledge for students in geology: : : 216 : : :a distinct transfer of climate.. ??? 239 : : : a warm-humid climate followed the Late Triassic: : :. > Late Triassic climate 244-248 You compare North American climate with GSB climate throughout your study, this is ok, however, I like to know why you chose North America, what are hypotheses on climate similarities and differences between N. America and SW China. 248 Âż here it will be interesting to discuss climate pattern in the Early Jurassic, why is which regional climate similar/ different from another regional climate? 305 Interesting is the observation that lake facies was widespread during a dry period in the Toarcian. You may further comment on this. 318 you use dolomite formation as a climate proxy, this is fine. You may refer to literature on dolomitization along the Persian Gulf ("Arabian Gulf") , eg. by McKenzie and others. 120 chichen-wire > chicken-wire (anhydrite) 253 That is the reddish rocks developed through the whole member: : :..., but it started: : : revise 257 : : :. Calcisols were also interpreted with the description of abundant calcretes: : :. (??) revise style 282 The Ma'anshan Member is likely the Pliensbachian: : :. 302 In other hand,: : : 321 : : :can serve the determination: : : I did not mark the

many additional small language inconsistencies Âż please revise text carefully.

In the abstract, we have deleted the sentence "Unlike marine archives, terrestrial sediments show more complicated and dynamic environment and climate", added a new sentence to introduce the climate of the Early Jurassic, and changed all the "carbon-oxygen" as "carbon and oxygen" in the text, tables, and figures.

Line 42. The questionable sentence was partly cited from the original paper (Xu et al., 2017). We deleted the sentence that is not distinctly relevant to the topic, and added another sentence to complement the subject of this paragraph.

Line 105. We revised the sentence as "We have made observations and descriptions of sedimentary characteristics for lithofacies analysis at six outcrop sections. . . . . .".

Line 116. Yes, we agree that the dolomite recognition is basic for any students in geology. We have deleted the sentence relevant (lines 115-123).

Line 120. We have corrected "chichen-wire" as the "chicken-wire".

Line 216. Taking the S(z)=2000 ppmV for pCO2 estimate originates from the calcisol determination and (semi-) arid climate indication (details are in the added parameter conditions in section 3.3). This climate condition is harmonized with the results indicated by climate-sensitive sediments, and may not be the representative of a climate transfer.

Line 239. Yes, the climate of the Hettangian (early Early Jurassic) is similar with that of the Late Triassic, but different from other ages of the Early Jurassic. The evidence of the Late Triassic warm-humid is provided in the first sentence of the first paragraph in section 5.1. And the warm-humid climate continued in the Hettangian age (verified in subsection 5.1.1). It is noted that we have stressed a total climate condition of the Early Jurassic as a (semi-) arid in the whole article even though the Hettangian was different in climate from the main Early Jurassic.

Line 244-248. Yes, we compared the climate between GSB and (North) America

through the section 5.1. From the comparison, we can see that: in the Hettangian, warm-humid climate in GSB was different from the arid climate in North America, but in the Sinemurian-Toarcian, both had the similar (semi-) arid climate. There are two reasons for us to compare the GSB with America in climate. One is that they were close in paleolatitude ( 15-30N. Fig. 1a) and similar climate zone (tropical) of the North Hemisphere in the Early Jurassic. The other one is that relatively complete climate-sensitive records of the Early Jurassic have only been published and are available in North America. In North (-west) China, the Early Jurassic succession is also complete, but the warm-humid climate is totally different from the GSB although we sometime compare them in the text. The comparison of the GSB and North America illustrates that the secular Early Jurassic (semi-) arid climate in the two regions could not have been responded to the (e.g., CAMP and Karro-Ferrar LIP) volcanism, which may have led to the global oscillation climate. This hypothesis is added in the last paragraph of section 5.1. The secular (semi-) arid climate could be related to the paleogeography. The aridity for the two regions does not obscure the total trend and eventful change of pCO2 demonstrated in section 5.2 as pCO2 is a global signal of climate.

Line 253. Yes, this sentence was confusing. We rewrote as: "Differences in the color appearance show that the reddish color started in the middle member in the central basin (Location A6. Fig. S2) but almost developed through the whole member in the western basin (Location A4. Fig. 6)" (new lines 272-274).

Line 257. We have revised the sentence to "We also interpret the reddish muddy sediments with abundant calcretes as the calcisol at sections of Dafang (Location A8. Zhang et al., 2016), Tianzhu (Location A9. Li and Chen, 2010), and Weiyuan (Location A10. SBG, 1980a) (new lines 276-278).

Line 282. The sentence was deleted in the new version as the age assignment is introduced in section 2. It is actually repetitious.

Line 302. We have composed this sentence with the first sentence in the paragraph.

Line 305. Yes, arid climate indication seems not consistent with the large lacustrine transgression. We then added some phrases in the next sentence (together composing a composite sentence), to show the possibility of arid records. Actually, the next couple of paragraphs are the process that verifies the arid climate by proxies of climate-sensitive sediments (lacustrine micritic dolomites, gypsum) and carbon and oxygen isotopes of lacustrine carbonates as well as (lacustrine or land side) reddish mudrocks.

Line 318. As we know, there are a number of hypotheses for the formation of dolomite mineral, such as primary authigenic origin, diagenetic replacement, microbial mediation. Yes, McKenzie J. and her team have much contributed to the microbial dolomite. We have referred to her papers in the revision, while we preferred to interpret the dolomites formed in an arid/evaporate climate condition especially when it is associated with gypsum (references added).

Line 321. We have considered the issue, and reorganized the sentence with others in the same paragraph. Please see the new version. Other similar issues were carefully checked and corrected in revision.

Figures I

like your figure 6 which serves as a very good baseline for your discussion (you may also add data from Xu et al. from the same study area in figure 6 or elsewhere?)

Adding the $\delta$13Com curve (Xu et al., 2017) in western GSB in figure 6 is a good idea. But the T-OAE $\delta$13Com curve (Xu et al., 2017) is too short (thin strata) to occur in the figure, i.e., the height of the curve is less than 1 cm if it is placed in the corresponding position. The other problem is that it is difficult to in age correlate between the two sections.

Fig 8

You plot changes in temperature not temperature, please correct this in the figure. The

global paleosol curve is, in this case not very helpful. I also wonder, if you should start your curve after the T-J boundary (?). The high-resolution data are not really discussed in your work and you did not include any T-J boundary data.

In figure 8, the seawater temperature was corrected!

Yes, the global paleosol curve is not very helpful for the observation and comparison of the Early Jurassic global climate, but we have to keep it in the figure as it shows the present situation of the global paleosol $pCO_2$ reconstruction except for this work.

Yes, we agree and delete the high-resolution latest Triassic data in the figure.

In addition, we have done lots of correction and amendment for the whole text, comparing to the previous version.

Thank you for time!

Yours sincerely,

Xianghui Li, Jingyu Wang, Troy Rasbury, et al.

---

## Author Comment (AC2) · 28 Jun 2020

Dear Prof. Helmut Weissert,

Please see the revised figures 1 and 8 for our reply to you comments.

Thank you!

Xianghui Li et al.

[Figure]

**Fig. 1.** Fig. 1

[Figure]

Fig. 2. Fig. 8

---

## Referee Comment (RC2) · Dan Breecker (Referee) · 6 Jul 2020

In this manuscript, Li and coauthors present new descriptions of Jurassic sediments and paleosols and new carbon and oxygen isotope data. They use these data to interpret an overall drying trend from Triassic into Jurassic, which is well supported, and they relate climate and environmental change to atmospheric pCO2.

My concerns are primarily related to the new pCO2 determinations made. First, luminescent calcite (lines 141 and 142) is probably not a good material to use for paleoCO2 determinations because luminescent pedogenic carbonate is thought to form under anoxic conditions, associated with water-saturation when there is a poor connection

between soil pore spaces and the atmosphere (Mintz, J. S., Driese, S. G., Breecker, D. O., & Ludvigson, G. A. (2011). Influence of changing hydrology on pedogenic calcite precipitation in Vertisols, Dance Bayou, Brazoria County, Texas, USA: implications for estimating paleoatmospheric pCO2.ÂăJournal of Sedimentary Research,Âă81(6), 394-400.). The paleosol carbonates studied here might be 'weakly' luminescent, but it is hard to tell without any quantification/standardization. It is also possible that there are other factors that influence luminescence. But all of this needs to be discussed so readers can evaluate the selection of materials. I will say, however, that the careful petrography and drilling of dense micritic zones is a plus.

I am concerned that the CO2 changes the authors interpret here may not be statistically significant changes. This is impossible to evaluate without uncertainty quantification. The authors do consider the effect of using different input values for the pCO2 calculation, but my guess is that they have nonetheless largely underestimated the error associated with their approach. For instance, the authors calculate d13Cr values from d13C values of OM measured in different locations (across the globe) from the carbonate nodules. What magnitude of uncertainty might this introduce? Furthermore, d13Ca is calculated from d13Cr. Given the effects of CO2 and water stress on d13C values of C3 plants , this approach is associated with substantial uncertainty that is not addressed in this manuscript. The authors recognize that there is uncertainty associated with the value of S(z). However, their consideration of S(z) = 2000 and 2500 ppmV is not an accurate representation of the uncertainty. I suggest error propagation that includes uncertainty associated with each input to the equation on line 159 and the results shown as error bars on each CO2 determination.

The descriptions of the sediments and paleosols reported here will be useful. I'm not sure I would call these Aridisols, though, because redoximorphic features are prominent (at least in some of the soils, e.g., Fig 3 a,b,c). Are you sure these are not Vertisols? Are there wedge-shaped peds? The authors mention abundant slickensides- a feature common in Vertisols.

---

## Author Comment (AC3) · 25 Jul 2020

Dear Prof. Dan Breecker,

We thank for your constructive comments and questions. Below are the replies.

——My concerns are primarily related to the new pCO2 determinations made. First, luminescent calcite (lines 141 and 142) is probably not a good material to use for paleoCO2 determinations because luminescent pedogenic carbonate is thought to form under anoxic conditions, associated with water-saturation when there is a poor connection between soil pore spaces and the atmosphere (Mintz, J. S., Driese, S. G.,

Breecker, D. O., & Ludvigson, G. A. (2011). Influence of changing hydrology on pedogenic calcite precipitation in Vertisols, Dance Bayou, Brazoria County, Texas, USA: implications for estimating paleoatmospheric pCO2 (Journal of Sedimentary Research, 81(6), 394-400.). The paleosol carbonates studied here might be 'weakly' luminescent, but it is hard to tell without any quantification/standardization. It is also possible that there are other factors that influence luminescence. But all of this needs to be discussed so readers can evaluate the selection of materials. I will say, however, that the careful petrography and drilling of dense micritic zones is a plus.

We thank for the valuable comment and providing the reference.

Mintz et al. (2011) provides a good example that hydrology influences the luminescence of calcretes, that we now cite and use to refine our discussion of the luminescent quality and justification for sampling the Jurassic calcretes we present in this paper.

All of the samples studied were screened for cathodoluminescence (CL), and only a few are shown in the manuscript. The samples shown have the brightest luminescence of all of the studied samples, and they were chosen because the quality of the images is much better. Most are calcites and dull to non-luminescent, with little ($\sim$ 5-10%) light orange or brownish red luminescence. The luminescence is almost certainly due to a relatively high Mn/Fe ratio, and we expect that seasonally the soils may have been water logged and disoxic.

More importantly, the key to distinguish the pedogenic calcretes from other geneses is the identification of both field occurrence and micro-texture. Our field and microscopic observations demonstrate that the calcretes have typical pedogenic features. The ginger-like calcretes are discrete within the Bk horizons, and do not form linear/tabular limestone. Slickensides and vertical rhizoliths can be often / sometimes seen in paleosols. Petrographically, we see that predominant micritic calcites occupy the dense areas of the calcretes (Fig. 2). Some calcretes have areas that were cracked and filled by secondary / diagenetic spar-calcites, that were avoided when micro-sampling.

Based on field and petrographic observations, we drilled powder samples for stable isotope analysis in dense micritic zones as the referee suggested. Thus, the carbon isotope value of carbonates can be used to estimate pCO2.

In the new figure 2, we inserted the scanned photos of the thin-sections and marked the cathodoluminescent and drilling dense areas and added plane light photos (Fig 2c and 2d), which roughly correspond to the CL image positions. We also added sentences in the text (3.2) to make notes on the observation results of petrography and CL images (new lines 137-145).

——I am concerned that the CO2 changes the authors interpret here may not be statistically significant changes. This is impossible to evaluate without uncertainty quantification. The authors do consider the effect of using different input values for the pCO2 calculation, but my guess is that they have nonetheless largely underestimated the error associated with their approach. For instance, the authors calculate d13Cr values from d13C values of OM measured in different locations (across the globe) from the carbonate nodules. What magnitude of uncertainty might this introduce? Furthermore, d13Ca is calculated from d13Cr. Given the effects of CO2 and water stress on d13C values of C3 plants, this approach is associated with substantial uncertainty that is not addressed in this manuscript. The authors recognize that there is uncertainty associated with the value of S(z). However, their consideration of S(z) = 2000 and 2500 ppmV is not an accurate representation of the uncertainty. I suggest error propagation that includes uncertainty associated with each input to the equation on line 159 and the results shown as error bars on each CO2 determination.

Following the suggestion, we have made the error propagation for the pCO2 results with procedures and formulas by Breecker (2013) and Breecker and Retallack (2014), and also considered parameters from Zhang et al. (2018). We have added the errors in figure 8d and a paragraph of uncertainty assessment in the end of the subsection 4.3 (new lines 244-250).

It is noted that parameters of temperature, $\delta 13Cr$, $\delta 13Ca$, $\delta 13Cs$, and S(z), remain the same as the calculation of pCO2 in the supplementary Table S4 in order to be consistent. We did not use variable temperatures and S(z) because the clumped isotopes analyses is currently not available in China. Additionally, and the depth (m) to the Bk is not known due to the disappearance or erosion of the top boundaries of the observed paleosols, thus making our S(z) values minimum estimates. $2°C$, 0.15%, 0.1%, and 0.1% are selected for the standard errors of temperature, $\delta 13Cc$, $\delta 13Com$, $\delta 13Ca$, and 788 ppmV is adopted for the standard error of soil carbonate transfer function with S(z) as suggested by Breecker and Retallack (2014). For details please see Table S5.

Results of error propagation show that the largest source of the uncertainty is the S(z) standard error 766 ppmV of modern soil carbonate (Breecker and Retallack, 2014). The second largest source of error is the S(z) value selection. Details of these errors are now discussed in a new paragraph in text (new lines 254-250, subsection 4.3). Other errors such as those for temperature, $\delta 13Cr$, $\delta 13Ca$, $\delta 13Cs$, exert far less to the uncertainty of pCO2 estimates.

——The descriptions of the sediments and paleosols reported here will be useful. I'm not sure I would call these Aridisols, though, because redoximorphic features are prominent (at least in some of the soils, e.g., Fig 3 a,b,c). Are you sure these are not Vertisols? Are there wedge-shaped peds? The authors mention abundant slickensides- a feature common in Vertisols.

We thank the reviewer for asking this question. There is much confusion from the literature with multiple classifications, and discussions of features such as slickensides that do seem to suggest more moisture. The aridisol (calcisol and gypsisol) and vertisol are are distinguished with clay-heave structure by Retallack (1993, 1998, 2001). For calcisols, a high ratio of clay can produce clay-heave and slickenside structures due to hydrological changes. Additionally, some angular, subangular, and platy peds are common in the calcisols, but few are wedge-shaped. We concur with the reviewer that the Jurassic soils we have studied are likely vertisols.

Please also see the two figures herein named of Figs. 1 and 2, which are actually Figs. 2 and 8 in text, respectively.

Thank you for attention!

Correspondence: Xianghui Li
* * *
[Figure]

Fig. 1.

[Figure]

**Fig. 2.**

---

## Author Response (AR2)

Dear Prof. Appy Sluijs,

We thank Referee #2 (Prof. Dan Breecker) for re-revision of our manuscript. We appreciate your edition of our manuscript numbered as cp-2020-35!

According to suggestions by Referee #2, we have revised Table S4 and try to re-calculate $p$CO$_2$ and to propagate errors with different way of $\delta^{13}C_a$ determination. Below are notes on how we did with questions one by one.

The authors have largely addressed my comments. One thing remains to be done in my opinion and that is to tidy up supplementary table S4 to make it most useful to readers. This table may be more heavily used than anything else in the manuscript. Here are my suggestions:

Columns 7 and 8 are not described, it is not 'd$^{12}$O' (columns 8 and 9).

We have added the description for columns 7 and 8, and also for column 2 in Table S4.

Column 13- provide the references for d13Cr here in the supplementary table

References have been provided in column 13, and corresponding references are increased below the table.

You did the error propagation, show the uncertainty on atmospheric CO2 in this table. Definitions of columns 4 and 5 are confusing- please explain what these are more clearly.

We revised notes of columns 4 and 5 in Table S4. Below are the revised:

4 Horizon (cm) of calcrete sample to the base of individual (calcisol) BK

5 Location (cm) of calcrete sample at the depth of the whole section (log)

Below the description of column 18, the table reads "Crossed data are invalable. " what does this mean?

We are sorry. It is a mistake. It could be caused by editing. We have deleted this note.

If you can determine d13Ca form marine carbonates, the uncertainty in CO2 would be lowered. You use d13C values of OM to get d13Ca, but we know that in climates where soil carbonates form vegetation is of water-stressed, which affects the carbon isotope fractionation between plant and atmospheric CO2. There is also subsntatial evidence, although there are arguments in the literature, that CO2 concentrations affects this fractionation. I think it is acceptable to move forward as is because your approach is clearly described, but if it is possible to use marine carbonates instead (or as another point of comparison) I would urge you to do so.

Up to date, global carbon isotopes of the Early Jurassic organic matters are well recovered and constructed, but those of coeval carbonates are not good as the organic matters (comp. Fig. 8a and 8b). The carbon isotope data of coeval carbonates are mainly derived from different basins in western Tethys, showing distinct differences of carbon isotopes. More

importantly, there are no secular records of surface water carbon isotope reported, which can represent the balance of carbon fractionation between sea-water and atmosphere. Though the carbon isotope of belemnite and ammonite carbonates is available in a few intervals, the isotope data could not reflect results of carbon fractionation in a stable surface water regime.

Supposed that the carbon isotope of belemnite to somewhat is eligible to the fractionation balance between sea-water and atmosphere, we tried to use the data from Cleveland Basin, UK. (Korte and Hesselbo, 2011) and from Pieniny Klippen Belt, Carpathians (Arabas et al., 2017) for the $pCO_2$ calculation and error propagation of the late Sinemurian-earliest Toarcian (~196-179 Ma). Due to few data of the early Sinemurian (~199-196 Ma) carbon isotope of planktonic fossil carbonate, we temporally use the carbon isotope of benthic oyster fossils from SW Britain (Korte et al., 2009) for $pCO_2$ calculation and error propagation. Results are attached as the supplementary Table S8, if required.

As shown in Table S8, both $pCO_2$ and errors are lowered, ~400 ppmV and 200 ppmV lower than those by $\delta^{13}C_a$ transferred by organic matter $\delta^{13}C_{om}$. However, the uncertainty is almost same each other. Please compare the results: mean values of $pCO_2$, error, and uncertainty: 1274 ppmV, 495 ppmV, 38.84% by carbonate (Table S8) and 1661 ppmV, 647 ppmV, 38.97% by organic matter (Table S5). And so quite similar are the uncertainty for each sample and its tendency (comp. Table S5 and S8).

We agree with Referee #2 that by the determination or transfer of $\delta^{13}C_a$ from marine carbonates, the uncertainty in $pCO_2$ could be lowered. In other hand, this relies on the conditions of parameters for the $pCO_2$ calculation and error propagation, particularly for the usage of S(z), paleosol depth of the BK, etc. We think there could be three main factors that causes the result of the same uncertainty using different $\delta^{13}C_a$ determination methods. The first factor is the aforementioned parameter condition; the second one is that $\delta^{13}C_c$ of planktonic fossil carbonates are not available for the Early Jurassic interval, which are the real representative of balanced carbon isotope fractionation between sea-water and atmosphere; the third one is the standard error of the Quaternary $S_{(z)}$. As discussed in text, the third one is still the largest source of the uncertainty.

For the reasons above, we don't think it is necessary to add the detailed results and discussions in text and Fig. 8. In the newly-uploaded manuscript, we only added one sentence explaining the similar uncertainty.

Thank you for time!

Yours sincerely,

Correspondence: Xianghui Li

---

## Author Response (AR3)

Dear Prof. Appy Sluijs,

Thank you for careful check of the manuscript!

Below are notes on the two issues.

First, in your table S4, columns 8 and 9 are headed d12Oc. This should be d18Oc.

We are sorry for the mistake! We have corrected in the revised Table S4.

Secondly, with Dr. Breecker, I think there needs to be clear what the CO2 errors derive from. Ideally, in addition to Table S4, you provide an indication which assumptions provide the basis for lines 15-18 in (the caption of) Figure 6 of the main paper.

For this suggestion, we added an error curve in Fig. 6, which are produced between $pCO_2$ results of column 16 ($S_{(z)}$=2500 ppmV) and 18 ($S_{(z)}$=2000 ppmV) in Table S4. Additional detail is also added in the Fig. 6 caption, where the largest uncertainty $S_{(z)}$ is further noted. The new error curve in Fig. 6 and extra details for the caption are added as below.

Discrepancies (errors) are produced from the $pCO_2$ subtraction of column 16 ($S_{(z)}$=2500 ppmV) from column 18 ($S_{(z)}$=2000 ppmV) in Table S4, indicating that the largest uncertainty for the estimate of $pCO_2$ is the $S_{(z)}$. The highest difference of $pCO_2$ is 965 (3463-2498) ppmV (Sample J1Z-18-01 at depth 182.6 m), the lowest is 245 (1226-981) ppmV (Sampe J1Z-12-01 at depth 148.9 m), and the mean is ~ 360 ppmV.

Thank you for time!

Yours sincerely,

Correspondence: Xianghui Li